# Anchored Diffusion Language Model

**Litu Rout**    **Constantine Caramanis**    **Sanjay Shakkottai**

The University of Texas at Austin

`{litu.rout, constantine, sanjay.shakkottai}@utexas.edu`

## Abstract

Diffusion Language Models (DLMs) promise parallel generation and bidirectional context, yet they underperform autoregressive (AR) models in both *likelihood modeling* and *generated text quality*. We identify that this performance gap arises when important tokens (e.g., key words or low-frequency words that anchor a sentence) are masked early in the forward process, limiting contextual information for accurate reconstruction. To address this, we introduce the *Anchored Diffusion Language Model (ADLM)*, a novel two-stage framework that first predicts distributions over important tokens via an anchor network, and then predicts the likelihoods of missing tokens conditioned on the anchored predictions. ADLM significantly improves test perplexity on LM1B and OpenWebText, achieving up to 25.4% gains over prior DLMs, and narrows the gap with strong AR baselines. It also achieves state-of-the-art performance in zero-shot generalization across seven benchmarks and surpasses AR models in MAUVE score, which marks the first time a DLM generates better human-like text than an AR model. Theoretically, we derive an Anchored Negative Evidence Lower Bound (ANELBO) objective and show that anchoring improves sample complexity and likelihood modeling. Beyond diffusion, anchoring boosts performance in AR models and enhances reasoning in math and logic tasks, outperforming existing chain-of-thought approaches. Please see our project page: https://anchored-diffusion-llm.github.io/ for code and demo.

## 1  Introduction

Large autoregressive language models (LLMs) have achieved remarkable success in next-token prediction, powering high quality text generation and emergent reasoning capabilities in AI systems. By generating tokens sequentially, autoregressive (AR) models like GPT (Brown et al., 2020), Gemini (Team-Gemini et al., 2023), LLaMA (Touvron et al., 2023), and Claude (Anthropic, 2024) condition on a growing prefix and excel at fitting the distribution of the next token. However, their sequential generation process makes it challenging to solve complex reasoning tasks, since the model does not see the entire sequence all at once.

An alternative paradigm has recently emerged in the form of *Diffusion Language Models* (DLMs), which perform masked-token prediction via iterative refinement. Inspired by diffusion models for continuous data (Sohl-Dickstein et al., 2015), these approaches corrupt text (e.g., by masking (Lou et al., 2024; Shi et al., 2024; Sahoo et al., 2024; Ou et al., 2025) or random flipping (Austin et al., 2021; Lou et al., 2024; Liu et al., 2025)) and train a model to denoise or reconstruct the original sequence over multiple steps (Austin et al., 2021; Li et al., 2022a; Lou et al., 2024; Sahoo et al., 2024; Shi et al., 2024; Ou et al., 2025). DLMs generate the entire sequence in parallel, allowing bidirectional attention for better context and potential gains in controllable generation, complex reasoning, and fast sampling. Despite their promise, masked diffusion models still lag behind AR models in *modeling the likelihood of missing tokens* and *generated text quality*. Even with modern training improvements, DLMs often achieve worse (higher) perplexity than AR transformers on standard benchmarks (Lou et al., 2024; Sahoo et al., 2024; Shi et al., 2024; Arriola et al., 2025).

We identify a key limitation in existing DLMs: when important tokens (e.g., low-frequency or semantically important words) are masked early in the forward process, the model lacks sufficient context to accurately reconstruct the original sequence. To address this, we propose the **Anchored Diffusion Language Model (ADLM)**(§3), which modifies the noising process such that important

Figure 1: **Anchored Diffusion Language Model (ADLM).** During inference, ADLM unmasks over two stages. (1) The *Anchor Transformer* outputs a sequence of **anchor predictions**; for any masked token, the corresponding anchor prediction is a distribution with support skewed towards over important tokens. (2) The *Denoising Transformer* conditions on these anchor predictions to unmask the remaining tokens. Anchor predictions help unmask important tokens early by reducing anchor-conditional entropy-guiding toward more accurate denoising.

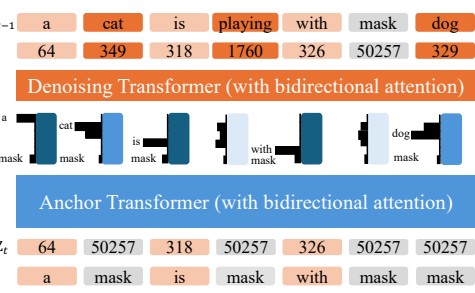

tokens remain unmasked longer in the forward process—equivalently, they are revealed earlier in the reverse process. We theoretically show that ADLM reveals important tokens early during unmasking and improves sample complexity within directed graphical models (§4).

As illustrated in Figure 1, ADLM introduces a two-level anchored denoising process. A key idea is that of anchors – *tokens whose inclusion as conditioning variables yields a substantial reduction in the conditional entropy of the remaining tokens* (e.g., knowing 'cat' and 'dog' reduces the entropy at position 4 and 6). In the first stage, an *Anchor Transformer* outputs a sequence of **anchor predictions**, where each anchor prediction for masked tokens is a mixture distribution that is skewed toward anchor tokens. In the second stage, a *Denoising Transformer* conditions on these soft anchor predictions to unmask the next set of tokens. We derive an *Anchored Negative Evidence Lower Bound* (ANELBO) objective to jointly train both networks within a single end-to-end differentiable framework. For anchoring, the loss is imposed only on the important (aka anchor) tokens. As a result, three things occur during inference (see Figure 1): (1) anchored predictions for already unmasked tokens peak around their true values (e.g., 'a' in position 1, 'is' in position 3 and 'with' in position 5); (2) anchor predictions corresponding to important tokens have a skewed distribution focusing on important words (e.g., 'cat' and 'dog' in positions 2 and 7); (3) anchor predictions for all other tokens yield fairly flat, high entropy distributions. The denoiser's bidirectional attention across all anchor predictions ensures that these predictions influence the denoising of all unmasked tokens.

Notably, the denoiser's prediction for 'playing' now benefits from anchors like 'cat' and 'dog' (masked tokens in input $Z_t$ but whose likelihoods are high in the anchor prediction sequence); this is unlike standard masked DLMs (Sahoo et al., 2024; Shi et al., 2024) where 'playing' relies only on already unmasked tokens ('a', 'is', 'with'). At the output, positions corresponding to anchor tokens have higher likelihood for anchor words, increasing their chance of being decoded earlier. Thus, during inference, the sequence of anchor predictions creates a **latent reasoning space** consisting of a sparse collection of anchors through which ADLM infers the full sequence more coherently and efficiently. Interestingly, these ideas extend to AR models where we demonstrate that anchoring enables a "**look-ahead**" planning behavior, which is in contrast to the commonly observed left-to-right decoding bias of standard chain-of-thought (CoT) reasoning (see Appendix C.2.5 for details).

Anchoring significantly improves both in-distribution and out-of-distribution (OOD) performance in generative modeling (§5.1), and also enhances the reasoning capabilities of AR models (§5.2). On the LM1B (Chelba et al., 2013) benchmark, ADLM achieves a test perplexity improvement of 9.54% over MDLM (Sahoo et al., 2024) and 25.4% over SEDD (Lou et al., 2024). On OpenWebText (OWT) (Gokaslan & Cohen, 2019), ADLM reaches a perplexity of 20.14 with 524B tokens, outperforming MDLM by 12.3%, matching the hybrid (AR+Diffusion) baseline BD3LM ($L' = 4$) (Arriola et al., 2025), which achieves 20.73. In terms of generation quality, ADLM achieves a GPT-2 Large perplexity of 26.8—surpassing MDLM by 39% and SEDD by 48%—and is thus the first diffusion language model to exceed AR in MAUVE score (measures human-like text quality) (Pillutla et al., 2021) using the remasking sampler. Furthermore, ADLM achieves state-of-the-art zero-shot perplexities on 6 out of 7 language modeling benchmarks and outperforms AR baselines on long-context and domain-specific datasets such as Lambada, PubMed, and ArXiv, demonstrating its strong language understanding and generalization capabilities. We also show that when anchoring is integrated into an AR model, it shows stronger logical consistency and planning capabilities in text generation and complex reasoning at GPT-2 scale.

**Contributions.** This paper makes the following contributions:

• We propose ADLM, a novel two-stage diffusion language model that improves the prediction of masked tokens through anchor-guided denoising (§3). We derive an anchored evidence lower

bound to train ADLM in an end-to-end fashion, proving improved sample complexity and better likelihood modeling in a DAG model (§4).

- ADLM achieves lower test perplexities than prior DLMs on LM1B and OWT, narrowing the gap with AR models (§5). Anchoring generalizes better in zero-shot evaluation, improving perplexity on OOD tasks such as PubMed and ArXiv, outperforming both MDLM and AR baselines (§5.1).
- We demonstrate the benefits of anchoring using two different samplers: (a) locked-in (Sahoo et al., 2024) and (b) remasking (Wang et al., 2025a) samplers. With remasking sampler, ADLM outperforms AR models in human-like text generation measured by MAUVE score (§5.1).
- Beyond diffusion, we integrate our anchoring mechanism into AR models, which leads to a novel reasoner that supplements conventional chain-of-thought. Our results show improvements in next-token prediction and supervised fine-tuning on Math (GSM8K (Cobbe et al., 2021)) and logical reasoning (ProntoQA (Saparov & He, 2023) and ProsQA (Hao et al., 2024) (§5.2)) tasks.

## 2 Background

Consider a discrete state space $\mathcal{S} = \mathcal{V}^L$, where $\mathcal{V} = \{1, \cdots, K-1, K\}$ denotes the set of discrete alphabets or tokens augmented with an extra $(K)$-th letter representing a dummy token called 'mask'. Further, $L$ is the dimension of each sequence $x = (x^1, x^2, \cdots, x^L) \in \mathcal{S}$ for $x^l \in \mathcal{V}, l \in [L]$. We represent each sequence as a collection of one-hot encodings as: $\mathbf{x} = (\mathbf{x}^1, \mathbf{x}^2, \cdots, \mathbf{x}^L)$, where $\sum_{j=1}^{K} \mathbf{x}^l(j) = 1, \mathbf{x}^l[j] \geq 0$ and $\mathbf{x}^l[x^l] = 1$. In case of a mask, we denote the corresponding one-hot encoded vector as $\mathbf{m} = (0, 0, \cdots, 0, 1)^T$. Let $X$ denote a random variable taking values in $\mathcal{S}$. Given a finite set of samples from an unknown data distribution $q(\cdot)$ supported on $\mathcal{S}$, the objective in generative modeling is to generate new samples from this distribution.

### 2.1 Auto-Regressive Models

Autoregressive models encompass widely used approaches in discrete generative modeling. These methods typically train a neural network to approximate the distribution of the next token conditioned on all previous tokens. This corresponds to the *causal factorization* of the joint data distribution $q(\cdot)$ by modeling the causal relationships in $X \sim q$ as follows (Jelinek, 1980; Bengio et al., 2003): $q(\mathbf{x}) = q(\mathbf{x}^1) \prod_{l=2}^{L} q(\mathbf{x}^l | \mathbf{x}^{1:l-1})$, where $\mathbf{x}^{1:l-1} := \mathbf{x}^1, \mathbf{x}^2, \cdots, \mathbf{x}^{l-1}$. A neural network parameterized by $p_\theta$ is trained to approximate these factors. The training objective for the neural network is designed to maximize the likelihood of a given finite set of sequences, which is equivalent to minimizing the negative log-likelihood: $\mathcal{L}_{\text{AR}}(\theta) = -\mathbb{E}_{X \sim q} [\log p_\theta(X)] = -\mathbb{E}_{X \sim q} \left[ \sum_{l=2}^{L} \log p_\theta(X^l | X^{1:l-1}) \right]$. This objective encourages the model to learn the conditional distributions, enabling autoregressive sampling from the learned distribution $p_\theta(\cdot)$.

### 2.2 Diffusion Language Models

Let $T$ represent the finite number of time steps used in a diffusion model. We denote by $t(i) = \frac{i}{T}$ and $s(i) = \frac{i-1}{T}$, where $i \in \{1, 2, \cdots, T\}$. For brevity, we drop the index $i$ from $t$ and $s$. In D3PM (Austin et al., 2021), the conditional of the forward process at time $t$ is given by

$$q(\mathbf{z}_t | \mathbf{x}) = \prod_{l=1}^{L} q(\mathbf{z}_t^l | \mathbf{x}), \quad q(\mathbf{z}_t^l | \mathbf{x}) = \text{Cat}\left(\mathbf{z}_t^l; \alpha_t \mathbf{x}^l + (1 - \alpha_t)\mathbf{m}\right), \quad l \in \{1, 2, \cdots, L\}, \quad (1)$$

which has a transition probability $q(\mathbf{z}_t^l | \mathbf{z}_s^l) = \text{Cat}(\mathbf{z}_t^l; \frac{\alpha_t}{\alpha_s} \mathbf{z}_s^l + (1 - \frac{\alpha_t}{\alpha_s})\mathbf{m})$ (see Appendix A.2 for details). The masking schedule $\alpha_t \in [0, 1]$ is predefined as a monotonically decreasing function of $t$ with $\alpha_0 = 1$ and $\alpha_1 = 0$. The corresponding reverse posterior becomes:

$$q(\mathbf{z}_s^l | \mathbf{z}_t^l, \mathbf{x}^l) = \begin{cases} \text{Cat}(\mathbf{z}_s^l; \mathbf{z}_t^l), & \mathbf{z}_t^l \neq \mathbf{m} \\ \text{Cat}\left(\mathbf{z}_s^l; \frac{\alpha_s - \alpha_t}{1 - \alpha_t} \mathbf{x}^l + \frac{1 - \alpha_s}{1 - \alpha_t}\mathbf{m}\right), & \mathbf{z}_t^l = \mathbf{m}. \end{cases} \quad (2)$$

This reverse posterior is useful because as we see in (3), it helps parameterize the generative model to have a similar form. The reverse process of D3PM (Austin et al., 2021) defines a $\theta-$parameterized joint distribution over sequences given by $p_\theta(\mathbf{x}, \mathbf{z}_{0:1})$. It follows a Markovian structure with transition probability $p_\theta(\mathbf{z}_s | \mathbf{z}_t) = \prod_{l=1}^{L} p_\theta(\mathbf{z}_s^l | \mathbf{z}_t)$. Intuitively, given a noisy latent $\mathbf{z}_t$, the model predicts a clean token and then re-noises it forward according to the forward dynamics defined in (1).

Recall that $\mathbf{x}$ denotes a sequence of $K$-dimensional one-hot encoded tokens, i.e., $\mathbf{x} = (\mathbf{x}^l)_{l=1}^L$. We slightly overload notation and use $\mathbf{x}_\theta = (\mathbf{x}_\theta^l)_{l=1}^L$ to represent a sequence of $\theta$-parameterized vectors

on the $K$-simplex. Each $\mathbf{x}_\theta^l$ defines a distribution over the vocabulary, where one-hot vectors $\mathbf{x}^l$ correspond to a token and lie at the corners of the simplex. Thus, we can interpret $\mathbf{x}_\theta$ as a mixture distribution over tokens, henceforth referred to as the *predicted mixture token*. With this notation, the probability of generating $\mathbf{x}^l$ given $\mathbf{z}_t$ can be compactly expressed as $p_\theta(\mathbf{x}^l|\mathbf{z}_t) = \langle \mathbf{x}_\theta^l(\mathbf{z}_t), \mathbf{x}^l \rangle$.

The (general) discrete-diffusion setting in D3PM (Austin et al., 2021) has subsequently been specialized to masking-based diffusion in MDLM (Sahoo et al., 2024) and MD4 (Shi et al., 2024). Their specialization has two properties: (i) *zero-masking*, where the predicted mixture has no support on the 'mask' letter, i.e., $\langle \mathbf{x}_\theta(\mathbf{z}_t)^l, \mathbf{m} \rangle = 0$, and (ii) *carry-over unmasking*, where for an already unmasked token (i.e., $\mathbf{z}_t^l \neq \mathbf{m}$), it continues to remain the same, meaning $\langle \mathbf{x}_\theta^l(\mathbf{z}_t), \mathbf{z}_t^l \rangle = 1$. Thus, for each token $l \in [L]$, this parameterization of the learned transition kernel leads to the following representation:

$$p_\theta(\mathbf{z}_s^l|\mathbf{z}_t) = q(\mathbf{z}_s^l|\mathbf{z}_t^l, \mathbf{x}_\theta^l(\mathbf{z}_t)) = \begin{cases} \text{Cat}(\mathbf{z}_s^l; \mathbf{z}_t^l), & \mathbf{z}_t^l \neq \mathbf{m} \\ \text{Cat}\left(\mathbf{z}_s^l; \frac{\alpha_s - \alpha_t}{1 - \alpha_t}\mathbf{x}_\theta^l(\mathbf{z}_t) + \frac{1 - \alpha_s}{1 - \alpha_t}\mathbf{m}\right), & \mathbf{z}_t^l = \mathbf{m}. \end{cases} \quad (3)$$

The denoising network is trained using Negative ELBO (NELBO) (Sohl-Dickstein et al., 2015; Austin et al., 2021) $\mathcal{L}_{\text{NELBO}}(\mathbf{x}; \theta) :=$

$$\mathbb{E}_{Z_0 \sim q(\cdot|\mathbf{x})}\left[ -\log p_\theta(\mathbf{x}|Z_0) \right] + \sum_{i=1}^{T} \mathbb{E}_{Z_{t(i)} \sim q(\cdot|\mathbf{x})}\left[ \frac{\alpha_{t(i)} - \alpha_{s(i)}}{1 - \alpha_{t(i)}} \sum_{l=1}^{L} \log\langle \mathbf{x}_\theta^l(Z_{t(i)}), \mathbf{x}^l \rangle \right]. \quad (4)$$

## 3 Anchored Diffusion Language Models

Our key idea is to anchor the denoising process using important tokens we call the *anchor tokens* (or [ANT] in short). These are tokens that, if revealed, make it much easier to generate the remaining tokens. As an example, if the underlying data distribution could be represented as a 2-depth tree (with each node on the tree being a token), knowledge of the value of the root node (anchor token) would lead to easier decoding of the leaves. As another example, in a sentence, knowledge of the verb or noun (anchor token) is likely more useful than the articles (e.g., 'a', 'an', 'the') or conjunction words.

Anchoring addresses the critical challenge posed by random masking in DLMs (Austin et al., 2021; Sahoo et al., 2024; Lou et al., 2024; Shi et al., 2024; Wang et al., 2025a), where important tokens in a sequence $\mathbf{x}$ may be masked in $\mathbf{z}_t$, making it difficult to estimate the missing likelihoods. To overcome this challenge, we split the denoising process into two steps. First, we use an *anchor network* to predict the probability mixture over important tokens for each position $l \in [L]$. Next, we employ a *denoising network* to aggregate these anchor predictions and compute likelihoods for the masked tokens. We call this approach **Anchored Diffusion Language Model (ADLM)**.

**ADLM Parameterization.** The forward process in ADLM follows the standard absorbing discrete diffusion formulation (1), with the inference posterior given in (2). To improve denoising, we introduce a new anchored parameterization of the reverse process. We propose to break the one-step denoising process, widely used in practice (Austin et al., 2021; Sahoo et al., 2024; Shi et al., 2024; Wang et al., 2025a; Ou et al., 2025; Nie et al., 2025a,b), into a two-stage anchored denoising framework. This allows latent reasoning over important tokens during pretraining.

Since the reverse process is Markovian, the joint probability distribution factorizes as: $p_\theta(\mathbf{x}, \mathbf{z}_{0:1}) = p_\theta(\mathbf{z}_1)p_\theta(\mathbf{x}|\mathbf{z}_0)\prod_{i=1}^{T} p_\theta(\mathbf{z}_{s(i)}|\mathbf{z}_{t(i)})$. We represent each learned transition $p_\theta(\mathbf{z}_{s(i)}|\mathbf{z}_{t(i)})$ by the composite of two functions, and this learned function (that maps $(\mathbf{z}_{s(i)}, \mathbf{z}_{t(i)}) \rightarrow [0, 1]$) is reparameterized through the pair $(\psi, \varphi)$ as: $p_\theta(\mathbf{z}_{s(i)}|\mathbf{z}_{t(i)}) := q(\mathbf{z}_{s(i)}|\mathbf{z}_{t(i)}, \mathbf{x}_\psi(\mathbf{y}_\varphi(\mathbf{z}_{t(i)})))$. Here, $\mathbf{y}_\varphi$ denotes the *anchor network*, which predicts a mixture distribution over important tokens from the masked input $\mathbf{z}_t$, and $\mathbf{x}_\psi(\mathbf{y}_\varphi(\mathbf{z}_t))$ denotes the *anchor-guided denoising network*, which predicts likelihoods of missing tokens conditioned on the important token mixture. We analyze the benefits in §4 theoretically, showing that anchoring reduces the training difficulty in DLMs by focusing optimization on important tokens. There are two key components in ADLM:

**(1) Anchor Transition.** Let $\mathcal{A}(\cdot)$ be an operator that takes a sequence $\mathbf{x} = (\mathbf{x}^l)_{l=1}^{L}$ as input and outputs an important token mixture $\mathbf{y} = (\mathbf{y}^l)_{l=1}^{L} = \mathcal{A}(\mathbf{x})$. We define the anchor transition as:

$$r(\mathbf{y}_s^l|\mathbf{z}_t^l, \mathbf{y}_\varphi(\mathbf{z}_t)) = \begin{cases} \text{Cat}(\mathbf{y}_s^l; (1 - \sigma_t)\mathbf{z}_t^l + \sigma_t\mathbf{m}), & \mathbf{z}_t^l \neq \mathbf{m}, \\ \text{Cat}(\mathbf{y}_s^l; \frac{\alpha_s - (1 - \sigma_t)\alpha_t}{1 - \alpha_t}\mathbf{y}_\varphi^l(\mathbf{z}_t) + \frac{1 - \alpha_s - \alpha_t\sigma_t}{1 - \alpha_t}\mathbf{m}), & \mathbf{z}_t^l = \mathbf{m}, \end{cases} \quad (5)$$

where $\mathbf{y}^l = \mathcal{A}(\mathbf{x}^l = \mathbf{z}_t^l)$. In other words, when a token $\mathbf{z}_t$ is already unmasked, the model preserves it as an important anchor token ([ANT]) with probability $(1 - \sigma_t)$, but can also re-mask it with

probability $\sigma_t$ (typically small). Conversely, when $\mathbf{z}_t$ is masked, the anchoring network $\mathbf{y}_\varphi(\cdot)$ predicts an important token mixture with probability $\frac{\alpha_s - (1-\sigma_t)\alpha_t}{1-\alpha_t}$, and keeps it masked with probability $\frac{1-\alpha_s-\alpha_t\sigma_t}{1-\alpha_t}$. This aims to reconstruct the important tokens earlier during sampling.

**(2) Inference Posterior.** Anchoring introduces an implicit reasoning mechanism into DLM pre-training. Once the model is trained, we modify the standard inference posterior to incorporate anchor-guided denoising. Since ADLM is trained to reason through anchor tokens internally, we do not explicitly decode the anchor tokens during inference. Thus, our inference posterior is given by:

$$q(\mathbf{z}_s^l|\mathbf{z}_t^l, \mathbf{x}_\psi^l(\mathbf{y}_\varphi(\mathbf{z}_t))) = \begin{cases} \mathrm{Cat}(\mathbf{z}_s^l; (1-\sigma_t)\mathbf{x}^l + \sigma_t\mathbf{m}), & \mathbf{z}_t^l \neq \mathbf{m}, \\ \mathrm{Cat}(\mathbf{z}_s^l; \frac{\alpha_s-(1-\sigma_t)\alpha_t}{1-\alpha_t}\mathbf{x}_\psi^l(\mathbf{y}_\varphi(\mathbf{z}_t)) + \frac{1-\alpha_s-\alpha_t\sigma_t}{1-\alpha_t}\mathbf{m}), & \mathbf{z}_t^l = \mathbf{m}, \end{cases} \quad (6)$$

where $\sigma_t$ controls the remasking probability at each timestep (Wang et al., 2025a). If a token $\mathbf{z}_t^l$ is already unmasked, the denoiser network $\mathbf{x}_\psi(\mathbf{y}_\varphi(\mathbf{z}_t))$ carries it over to the next time step with probability $(1-\sigma_t)$, while still allowing a small probability $\sigma_t$ of masking for correction. When $\mathbf{z}_t^l$ is masked, the inference posterior interpolates between predicting the missing token via the anchored logits and remasking it, with weights determined by $\sigma_t$ and the forward process parameters $(\alpha_t, \alpha_s)$.

Important tokens, once masked, lead to information loss that affects reconstruction by standard denoising network. By predicting these important tokens early via the anchor network, ADLM: (1) introduces intermediate latent reasoning, (2) maintains stronger context throughout the denoising trajectory, and (3) enables high quality sequence generation. In practice, even a lightweight denoising network (e.g., using half the number of layers compared to the anchor network) significantly improves overall likelihood modeling when guided by anchored predictions.

**Training Objective.** Given a sequence $\mathbf{x}$ and important token mixture $\mathbf{y}$, we optimize the parameters ($\psi$ and $\varphi$) of ADLM using *Anchored Negative Evidence Lower Bound (ANELBO)* (see **Theorem 4.1**):

$$\mathcal{L}_{\mathrm{ANELBO}}(\mathbf{x}, \mathbf{y}; \varphi, \psi) = \mathbb{E}_{Z_0 \sim q(\cdot|\mathbf{x})}[-\log p_\psi(\mathbf{x}|\mathbf{y}_\varphi(Z_0))] + \quad (7)$$

$$\sum_{i=1}^{T} \mathbb{E}_{Z_{t(i)} \sim q(\cdot|\mathbf{x})}\left[\frac{(1-\sigma_{t(i)})\alpha_{t(i)} - \alpha_{s(i)}}{1-\alpha_{t(i)}} \sum_{l=1}^{L} \log\langle \mathbf{x}_\psi^l(\mathbf{y}_\varphi(Z_{t(i)})), \mathbf{x}^l\rangle + \gamma \log\langle \mathbf{y}_\varphi^l(Z_{t(i)}), \mathbf{y}^l\rangle\right],$$

where $\gamma$ controls anchor strength. For $\sigma_{t(i)} = 0 = \gamma$, we recover the standard MDLM (4).

**Anchor Token Selection.** We study three different anchor token selection mechanisms: **(i) Relative frequency** for LM1B and OWT in generative modeling tasks using DLMs (§5.1), where we identify important tokens using a frequency-based criterion inspired by TF-IDF. For each token $\mathbf{x}^l$ in a sequence $\mathbf{x}$, we compute its relative frequency as $\mu(\mathbf{x}^l) = \frac{1}{L}\sum_{j=1}^{L}\mathbf{1}_{\{\mathbf{x}^j=\mathbf{x}^l\}}$. Tokens with $\mu(\mathbf{x}^l) \leq \tau$ are considered important and contribute to the anchoring loss. **(ii) Digit extraction** for GSM8K in math reasoning tasks using AR models, where we use the numerical digits as the anchors, and this enables out-of-order reasoning; and **(iii) Extraction of verbs and nouns** (excluding articles such as 'a', 'an', 'the') for logical reasoning tasks in AR models with symbolic reasoning traces from ProntoQA and ProsQA. We defer further details on anchored AR models to §5.2.

# 4 Theoretical Results

## 4.1 Anchored Negative Evidence Lower Bound

Recall that each latent variable $Z_t$ is a corrupted version of the original sequence $\mathbf{x}$, and $\mathbf{y} = \mathcal{A}(\mathbf{x})$ is a mixture of important tokens. We define the anchored transition function as:

$$r(\mathbf{y}_s^l|\mathbf{z}_t^l, \mathbf{y}^l) := \begin{cases} \mathrm{Cat}(\mathbf{y}_s^l; (1-\sigma_t)\mathbf{y}^l + \sigma_t\mathbf{m}), & \mathbf{z}_t^l \neq \mathbf{m}, \\ \mathrm{Cat}(\mathbf{y}_s^l; \frac{\alpha_s-(1-\sigma_t)\alpha_t}{1-\alpha_t}\mathbf{y}^l + \frac{1-\alpha_s-\alpha_t\sigma_t}{1-\alpha_t}\mathbf{m}), & \mathbf{z}_t^l = \mathbf{m}, \end{cases} \quad (8)$$

where $\mathbf{z}_1^l = \mathbf{y}_1^l = \mathbf{m}$ and $\alpha_t, \sigma_t$ are time-dependent coefficients derived from the corruption schedule. This motivates our choice of anchored transition (5) in ADLM parameterization (§3). To align the model's anchored predictions with this target transition, we define the *anchor loss*:

$$\mathcal{L}_{\mathrm{Anchor}}(\mathbf{x}; \varphi) := \mathbb{E}_{q(Z_{0:1}|\mathbf{x})}\left[\sum_{i=0}^{T} \mathrm{D}_{\mathrm{KL}}(r(Y_{s(i)}|Z_{t(i)}, \mathbf{y}) \parallel r_\varphi(Y_{s(i)}|Z_{t(i)})),\right], \quad \mathbf{y} = \mathcal{A}(\mathbf{x}), \quad (9)$$

where $r_\varphi$ is a learned parametric anchor transition function. We now derive the ANELBO objective, which integrates the anchor network within a denoising model $\mathbf{x}_\psi(\mathbf{y}_\varphi(\cdot))$. The resulting bound regularizes the denoising process using structured guidance from the anchor predictions.

**Theorem 4.1** (Anchored Negative Evidence Lower Bound). *Suppose the inference posterior is parameterized as in (6). Denote by $\theta$ the collection of parameters of the anchor and denoiser networks, i.e., $\theta = [\psi, \varphi]$. Given a sequence $\mathbf{x} = (\mathbf{x}^l)_{l=1}^L$, let the important token mixture $\mathbf{y} = (\mathbf{y}^l)_{l=1}^L = \mathcal{A}(\mathbf{x})$ be obtained through the operator $\mathcal{A}(\cdot)$. Then, the anchored negative log-likelihood is bounded by:* $-\log p_\theta(\mathbf{x}) + \gamma \mathcal{L}_{\text{Anchor}}(\mathbf{x}; \varphi) \leq \mathcal{L}_{\text{ANELBO}}(\mathbf{x}; \psi, \varphi)$, *where*

$$\mathcal{L}_{\text{ANELBO}}(\mathbf{x}; \psi, \varphi) := \mathbb{E}_{Z_0 \sim q(\cdot|\mathbf{x})} \left[ -\log p_\psi(\mathbf{x}|\mathbf{y}_\varphi(Z_0)) \right]$$

$$+ \sum_{i=1}^T \mathbb{E}_{Z_{t(i)} \sim q(\cdot|\mathbf{x})} \left[ \lambda_{t(i)} \sum_{l=1}^L \left( \log\langle \mathbf{x}_\psi^l(\mathbf{y}_\varphi(Z_{t(i)})), \mathbf{x}^l \rangle + \gamma \log\langle \mathbf{y}_\varphi^l(Z_{t(i)}), \mathbf{y}^l \rangle \right) \right],$$

*with weight $\lambda_{t(i)} = \frac{(1-\sigma_{t(i)})\alpha_{t(i)} - \alpha_{s(i)}}{1-\alpha_{t(i)}}$ and $\gamma > 0$.*

**Remark 4.2.** *We choose a constant $\gamma$ to simplify the notation. Our derivation also applies to a time dependent $\gamma_t$. This only changes the contribution of the anchor loss in $\mathcal{L}_{\text{ANELBO}}(\mathbf{x}; \psi, \varphi)$.*

**Implications.** The ANELBO objective highlights two important aspects induced by anchoring:

- The first term $\log\langle \mathbf{x}_\psi^l(\mathbf{y}_\varphi(Z_{t(i)})), \mathbf{x}^l \rangle$ encourages the denoising network to model the likelihoods of missing tokens, conditioned on the output of the anchor network.
- The second term $\log\langle \mathbf{y}_\varphi^l(Z_{t(i)}), \mathbf{y}^l \rangle$ directly supervises the anchor network, encouraging it to predict important tokens early during sampling.

To summarize, anchoring improves likelihood because the denoiser does not waste capacity modeling high-entropy distributions over missing key words, having already resolved them via anchors.

### 4.2 Anchored Graphical Model Analysis

The core training objective in both AR and DLMs is maximum likelihood estimation (MLE). MLE has a rich foundation in graphical models (Koller & Friedman, 2009), providing a principled way to understand expressiveness, tractability, and sample complexity. We reinterpret AR and DLM training as learning in directed graphical models (DAGs) and formally analyze our anchoring mechanism. While rooted in classical theory, we demonstrate that anchoring yields practical benefits in both large-scale pretraining (§5.1) and supervised fine-tuning (§5.2) tasks.

**Assumption 4.3.** *Suppose the following properties hold: (i) Each conditional distribution $p(\mathbf{x}^l|\cdot)$ is modeled as a categorical distribution. (ii) The model is parameterized by Conditional Probability Tables (CPTs); that is, a distinct parameter is assigned to each configuration of the conditioning set. (iii) Anchor sets $\pi_l \subset \{1, \ldots, L\} \setminus \{l\}$ are fixed and of bounded size $|\pi_l| \leq d$, with $d \ll L$.*

**Proposition 4.4** (Reduced Sample Complexity via Anchoring). *Suppose Assumption 4.3 holds. The sample complexity of MLE is given as follows: (i) Standard AR: Each token $\mathbf{x}^l$ is conditioned on all previous tokens $\mathbf{x}^{1:l-1}$. The total number of parameters is $\mathcal{O}(K^L)$, resulting in a sample complexity of $\mathcal{O}(K^L)$. (ii) Standard DLM: Each masked token $\mathbf{x}^l$ is predicted conditioning on all other tokens $\mathbf{x} \setminus \mathbf{x}^l$. The per-token parameter count is $\mathcal{O}(K^L)$, leading to a total sample complexity of $\mathcal{O}(LK^L)$. (iii) Anchored AR: Each token $\mathbf{x}^l$ is conditioned only on a fixed-size anchor set $\mathbf{x}^{\pi_l}$. The number of parameters per conditional is $\mathcal{O}(K^{d+1})$, giving a total sample complexity $\mathcal{O}(LK^{d+1})$. (iv) Anchored DLM: Each masked token $\mathbf{x}^l$ is predicted using only anchor tokens $\mathbf{x}^{\pi_l} \setminus \{\mathbf{x}^l\}$. The per-token parameter count becomes $\mathcal{O}(K^{d+1})$, resulting in a total sample complexity of $\mathcal{O}(LK^{d+1})$.*

**Implications.** Assuming the existence of important tokens (anchors) in a sequence, anchoring achieves exponential reductions in sample complexity: $\mathcal{O}(K^L)$ to $\mathcal{O}(LK^{d+1})$. We provide additional theoretical results and discussion in §A. We defer discussion on sample complexity to §A.3.1 and §A.3.2. We discuss an expectation-maximization (EM) (Dempster et al., 1977) interpretation of our anchored training procedure in §A.3.3. Finally, we provide an example in §A.3.4 that shows improved likelihood of unmasking anchor tokens early in the denoising process.

## 5 Experiments

Our experiments are designed to evaluate two main aspects of language modeling: (1) likelihood modeling and (2) generated text quality. Prior work has shown that improved perplexity during pretraining often correlates with better downstream performance (Austin et al., 2021; Lou et al., 2024; Shi et al., 2024; Sahoo et al., 2024; Wang et al., 2025a; Arriola et al., 2025). Since our focus is on pretraining, we primarily evaluate models in terms of their test/validation perplexities, as well as

Table 1: Test perplexities (PPL↓) on LM1B and OWT. [†]Reported in (Sahoo et al., 2024). **Bold**: Best diffusion method. We retrain AR and MDLM to match performance reported in original papers. Our method outperforms previous diffusion language models using the same number of training tokens.

| (a) LM1B | | |
|---|---|---|
| Model | PPL ($\downarrow$) | Tokens |
| *Autoregressive* | | |
| Transformer-X Base (Dai et al., 2019) | 23.5 | - |
| OmniNet$_T$ (Tay et al., 2021) | 21.5 | - |
| Transformer[†] (Sahoo et al., 2024) | 22.32 | 33B |
| Transformer (retrained) | 21.55 | 65B |
| | | |
| *Diffusion* | | |
| BERT-Mouth (Wang & Cho, 2019) | 142.89 | - |
| D3PM (absorb)[†] (Austin et al., 2021) | 76.90 | - |
| Diffusion-LM (Li et al., 2022b) | 118.62 | - |
| DiffusionBert[†] (He et al., 2023) | 63.78 | - |
| SEDD (Lou et al., 2024) | 32.79 | 33B |
| MDLM (Sahoo et al., 2024) | 27.04 | 33B |
| MDLM (retrained) | 27.07 | 33B |
| UDLM (uniform) (Schiff et al., 2025) | 31.30 | 33B |
| DUO (uniform) (Sahoo et al., 2025) | 29.90 | 33B |
| ADLM (ours) | **26.40** | 33B |
| | | |
| MDLM (retrained) | 25.49 | 65B |
| ADLM (ours) | **24.46** | 65B |

| (b) OWT | | |
|---|---|---|
| Model | PPL ($\downarrow$) | Tokens |
| *Autoregressive* | | |
| AR (retrained) | 17.94 | 110B |
| AR[†] (Sahoo et al., 2024) | 17.54 | 262B |
| AR (retrained) | 17.26 | 524B |
| *Diffusion* | | |
| MDLM (retrained) | 24.04 | 110B |
| ADLM (ours) | **21.66** | 110B |
| | | |
| SEDD[†] (Lou et al., 2024) | 24.10 | 262B |
| MDLM[†] (Sahoo et al., 2024) | 23.21 | 262B |
| MDLM (retrained) | 23.17 | 262B |
| GIDD (von Rütte et al., 2025) | 22.29 | 262B |
| UDLM (uniform) (Schiff et al., 2025) | 27.40 | 262B |
| DUO (uniform) (Sahoo et al., 2025) | 25.20 | 262B |
| ADLM*(ours) | **21.79** | 262B |
| ADLM (ours) | **20.62** | 262B |
| | | |
| MDLM (Sahoo et al., 2024) | 22.98 | 524B |
| MD4 (Shi et al., 2024) | 22.13 | 524B |
| GenMD4 (Shi et al., 2024) | 21.80 | 524B |
| BD3LM ($L' = 4$) (Arriola et al., 2025) | 20.73 | 524B |
| ADLM (ours) | **20.14** | 524B |

zero-shot generalization. Additionally, we measure generation quality using GPT-2 Large perplexity, entropy, and MAUVE (Pillutla et al., 2021) scores. While perplexity (PPL) captures the likelihood, MAUVE score measures divergence between neural text and human text.

## 5.1 Diffusion Language Models

**Setup.** We evaluate ADLM on two benchmarks: One Billion Words (LM1B) (Chelba et al., 2013) and OpenWebText (OWT) (Gokaslan & Cohen, 2019). For LM1B, we use a context length of 128 with the BERT-base-uncased tokenizer and evaluate on the standard test split. For OWT, we use the GPT-2 tokenizer (Radford et al., 2019). Our anchor network adopts the transformer architecture from SEDD (Lou et al., 2024), based on the Diffusion Transformer (DiT) (Peebles & Xie, 2023) with rotary positional embeddings (Su et al., 2024). The denoiser network uses the same base architecture but with half the number of transformer layers.

We experiment with two diffusion samplers: (a) the *locked-in sampler* from MDLM (Sahoo et al., 2024), which fixes previously unmasked tokens by setting $\sigma_t = 0$, and (b) the *remasking sampler* from ReMDM (Wang et al., 2025a), which allows re-masking with a small $\sigma_t \neq 0$. For fair comparison, we adopt the exact sampler configurations used in the respective baseline implementations.

**Baselines.** We compare against the following baselines: (1) the Autoregressive (AR) architecture from (Sahoo et al., 2024) trained with next-token prediction; (2) SEDD (Lou et al., 2024): A score entropy discrete DLM; (3) MDLM (Sahoo et al., 2024): A masked diffusion language model; (4) MD4 (Shi et al., 2024): A masked diffusion model for discrete data; (5) BD3LM (Arriola et al., 2025): A hybrid approach combining AR and diffusion components; (6) ReMDM (Wang et al., 2025a): MDLM with re-masking sampler; (7) GIDD (von Rütte et al., 2025): A DLM that interpolates between masking and uniform noising; and flow matching methods, such as (8) DFM (Gat et al., 2024) and (9) Forward-Backward (FB) (Campbell et al., 2022) samplers. We follow the implementation of these baselines from MDLM (Sahoo et al., 2024) and ReMDM (Wang et al., 2025a) repository. We describe each baseline and provide links to the source code in §C.

### 5.1.1 Improved Likelihood Modeling and Generated Text Quality

We first evaluate ADLM on likelihood modeling and generated text quality using the *locked-in* sampler from MDLM (Sahoo et al., 2024). Based on our ablation study (deferred to §C), we adopt $\gamma = $ 3e-3 and $\tau = 5$ as our default configuration for anchoring (§3).

**Results on LM1B.** Table 1 (a) shows that ADLM outperforms previous diffusion models such as SEDD, MDLM, and DiffusionBERT. At 33B tokens, ADLM achieves a test PPL of 26.40, improving over MDLM (27.04). Scaling to 65B tokens further reduces PPL to 24.46, approaching AR models like our retrained Transformer (21.55).

**Results on OWT.** Table 1 (b) presents the test PPL of our proposed method, ADLM, across three training regimes: 110B, 262B, and 524B tokens. At each scale, ADLM consistently outperforms diffusion-based baselines such as MDLM and GIDD, as well as the hybrid (AR+Diffusion) BD3LM. Notably, at 262B tokens, ADLM achieves a PPL of 20.62, narrowing the gap with the AR models, which reach a PPL of 17.54. ADLM* uses our multi-stage design (anchor and denoiser with $\gamma = 0$) to train MDLM that improves its PPL from 23.17 to 21.79. It demonstrates that anchoring is not just adding extra capacity, but truly guiding efficient learning. At 524B tokens, ADLM further improves to 20.14, approaching the AR performance (17.26). These results verify the effectiveness of ADLM in bridging the gap between DLMs and AR approaches, without relying on AR components.

**Generated Text Quality:** While test PPL evaluates the ability to predict missing tokens, it does not necessarily reflect the quality of generated text. Following common practice, we assess generated text quality using GPT-2 Large. Table 2 shows GPT-2 Large PPL scores on OWT for models trained on 524B tokens. Our method, ADLM, achieves the lowest PPL among DLMs, outperforming prior approaches such as MDLM and SEDD. ADLM surpasses the hybrid BD3LM at both $L' = 8$ and $L' = 16$ block lengths. Since smaller block lengths make BD3LM behave more like an AR model ($L' = 1$ is equivalent to pure AR), larger block lengths represent the diffusion regime more accurately. Thus, higher $L'$ provides a fair evaluation against other DLMs. We refer to Table 7 in Appendix C.1.3 for a more extensive study.

Table 2: GPT2-Large perplexities (PPL; ↓) on OWT (524B tokens). ADLM uses the remasking sampler with 1000 steps.

|  | PPL ($\downarrow$) |
|---|---|
| AR (Sahoo et al., 2024) | 14.10 |
| BD3LM (Arriola et al., 2025) |  |
| $\quad L' = 16$ | 33.4 |
| $\quad L' = 8$ | 30.4 |
| $\quad L' = 4$ | 25.7 |
| SEDD (Lou et al., 2024) | 52.0 |
| MDLM (Sahoo et al., 2024) | 44.2 |
| ADLM (ours) | **26.8** |

**Zero-shot Perplexity Evaluation.** We train ADLM on OWT and evaluate its zero-shot generalization across seven diverse benchmarks using validation perplexity. As shown in Table 4, ADLM consistently outperforms prior diffusion-based models such as SEDD and MDLM on 6 of 7 tasks, and matches performance on the remaining WikiText benchmark. It also surpasses the hybrid BD3LM (with block length $L' = 4$) on five benchmarks. These results suggest that ADLM learns robust representations because *the notion of importance captured by the anchor network generalizes even when the distribution shifts.*

Notably, ADLM outperforms the AR baseline on three challenging datasets: (1) Lambada, which tests long-range contextual understanding, and (2) PubMed and (3) ArXiv, which evaluate scientific language modeling. These results indicate that ADLM not only narrows the performance gap with AR models but can exceed them on tasks requiring long-context reasoning and specialized knowledge. Importantly, ADLM achieves these gains with the same number of neural function evaluations (NFEs) as AR models, demonstrating both efficiency and strong out-of-distribution generalization.

**Remasking Sampler.** Now, we evaluate ADLM using the remasking sampler (Wang et al., 2025a), with results shown in Table 3. This flexibility enables more expressive and diverse sampling. Our pre-training method, ADLM, when combined with remasking sampler, achieves state-of-the-art performance across multiple metrics. Importantly, it becomes the first DLM to outperform the AR model in MAUVE score, particularly at $T = 2048$ and $T = 4096$. Anchoring provides up to 4x improvement in inference speed compared to ReMDM as evident from Table 3 (highlighted in green).

In addition to MAUVE, we report GPT-2 Large perplexity and entropy to assess generation quality and diversity with increasing number of sampling steps. While test PPL can be artificially lowered by repeating high-likelihood phrases, such models typically exhibit low entropy and fail to capture the richness of natural language. Our method maintains high entropy—closely matching the data distribution—while achieving low PPL and high MAUVE scores. This balance indicates that ADLM not only generates high quality human-like text but also preserves the diversity.

## 5.2 Auto-Regressive Models

We observe that the benefits of anchoring extend beyond the diffusion setting. Specifically, anchoring can be incorporated into auto-regressive generative models as discussed in Appendix C.2, as well as for supervised finetuning (SFT) of pretrained auto-regressive models as discussed in Appendix C.2.5.

Below, we discuss the second setting of supervised finetuning of pretrained auto-regressive models. Our key observation is that inserting [ANT] after questions and before the start of (reason, answer) tokens enhances reasoning capabilities of AR models (see also Appendix C.2.5).

Table 3: Evaluation of sample quality using the ADLM with remasking sampler (Wang et al., 2025a) on OWT. ADLM† outperforms state-of-the-art masked diffusion and flow-matching methods. For $T = 2048$ and $T = 4096$, ADLM surpasses AR in MAUVE score (measures human-like text).

| Method | MAUVE (↑) | | | Gen PPL. (↓) | | | Entropy (↑) | | |
|---|---|---|---|---|---|---|---|---|---|
| Data | 1.00 | | | 14.8 | | | 5.44 | | |
| AR *(T=1024)*† | 0.760 | | | 12.1 | | | 5.22 | | |
| | *T=1024* | *T=2048* | *T=4096* | *T=1024* | *T=2048* | *T=4096* | *T=1024* | *T=2048* | *T=4096* |
| SEDD (absorb) | 0.008 | 0.008 | 0.009 | 104.7 | 103.2 | 102.5 | 5.62 | 5.61 | 5.61 |
| MDLM | 0.042 | 0.037 | 0.035 | 51.3 | 51.3 | 50.9 | 5.46 | 5.46 | 5.45 |
| MDLM+FB | 0.133 | 0.197 | 0.243 | 33.8 | 28.6 | 22.8 | 5.35 | 5.28 | 5.18 |
| MDLM+DFM | 0.254 | 0.294 | 0.269 | 21.7 | 21.0 | 20.7 | 5.20 | 5.19 | 5.17 |
| ReMDM | 0.403 | 0.610 | 0.656 | 28.6 | 22.8 | 17.6 | 5.38 | 5.30 | 5.20 |
| ADLM (ours) | **0.699** | **0.788** | **0.791** | 25.4 | 20.3 | 15.9 | 5.35 | 5.28 | 5.19 |
| | *T=128* | *T=256* | *T=512* | *T=128* | *T=256* | *T=512* | *T=128* | *T=256* | *T=512* |
| SEDD (absorb) | 0.007 | 0.007 | 0.008 | 119.2 | 110.1 | 107.2 | 5.65 | 5.63 | 5.62 |
| MDLM | 0.015 | 0.023 | 0.031 | 61.5 | 55.8 | 53.0 | 5.52 | 5.49 | 5.48 |
| MDLM+FB | 0.064 | 0.084 | 0.100 | 42.8 | 39.6 | 37.1 | 5.44 | 5.41 | 5.38 |
| MDLM+DFM | 0.041 | 0.144 | 0.211 | 37.9 | 26.5 | 23.3 | 5.31 | 5.26 | 5.23 |
| ReMDM | 0.057 | 0.216 | 0.350 | 42.5 | 30.5 | 21.1 | 5.43 | 5.34 | 5.21 |
| ADLM (ours) | **0.140** | **0.349** | **0.573** | 52.5 | 39.85 | 31.6 | 5.52 | 5.46 | 5.40 |

Table 4: Zero-shot validation perplexities (↓) of models trained on 524B tokens from OWT. ADLM achieves a new state-of-the-art among diffusion language models and outperforms autoregressive (AR) models on three benchmarks: Lambda, PubMed, and ArXiv. All models use 1024 NFEs.

| | Lambada | PTB | Wikitext | LM1B | AG News | PubMed | ArXiv |
|---|---|---|---|---|---|---|---|
| AR | 51.28 | 82.05 | 25.75 | 51.25 | 52.09 | 49.01 | 41.73 |
| BD3-LM ($L' = 4$) | 50.03 | 96.81 | 31.31 | 60.88 | 61.67 | 42.52 | 39.20 |
| SEDD | 49.86 | 100.09 | 34.28 | 68.20 | 62.09 | 44.53 | 38.38 |
| MD4 | 48.43 | 98.16 | 32.42 | - | - | - | - |
| MDLM | 47.52 | 95.26 | 32.83 | 67.01 | 61.15 | 41.89 | 37.37 |
| UDLM (uniform) | 53.57 | 112.82 | 39.42 | 77.59 | 80.96 | 50.98 | 44.08 |
| DUO (uniform) | 49.78 | 89.35 | 33.57 | 73.86 | 67.81 | 44.48 | 40.39 |
| ADLM (ours) (262B) | 44.93 | 98.16 | 32.45 | 65.59 | 57.10 | 38.29 | 35.08 |
| ADLM (ours) (524B) | **44.32** | **95.37** | **31.94** | **64.43** | **55.72** | **37.56** | **33.69** |

**Setup.** We use a pretrained GPT-2 (Radford et al., 2019) model as the base architecture. We finetune the base model on math and logical reasoning tasks using standard SFT on reasoning traces and answers. We evaluate on three benchmarks: (1) GSM8K (Cobbe et al., 2021)–grade-school math problems with arithmetic reasoning, (2) ProntoQA (Saparov & He, 2023)–rule-based logical reasoning, and (3) ProsQA (Hao et al., 2024)– planning with structured reasoning over graph-based inference traces. Our experimental setup follows the fine-tuning protocols outlined in prior work (Hao et al., 2024), enabling direct comparison with established baselines.

**Baselines.** We compare against a range of latent reasoning and chain-of-thought (CoT) methods. These include standard CoT finetuning (Wei et al., 2022), improved variants such as iCoT (Deng et al., 2024), and multi-stage fine-tuning approaches like COCONUT (Hao et al., 2024). We compared with two additional baselines: **No-CoT**, which trains models directly on question-answer pairs without intermediate reasoning traces, and **Pause Token** (Goyal et al., 2024), which inserts special pause tokens between the question and answer to encourage thinking. These methods are finetuned using the same base model: GPT-2 (openai-community/gpt2) with identical parameter count. We also include a recent work BoLT (Ruan et al., 2025) that reasons to learn from latent thoughts.

### 5.2.1 Improved Reasoning using Anchored Chain-of-Thought

Inspired by recent work on chain-of-thought prompting (Wei et al., 2022; Hao et al., 2024; Goyal et al., 2024), we investigate whether anchoring improves the reasoning ability of AR models in both math and logic domains. To operationalize our anchoring mechanism in AR models, we insert [ANT] after question and before (reason, answer) tokens, and then use standard SFT with important tokens from reasoning traces as lables for these [ANT] tokens. We defer implementation details to Appendix C. We refer to this variant as **Anchored Chain-of-Thought (ACoT)**, illustrated in Figure 2. Here, [BoA]/[EoA] denote the beginning and end of anchors, while [BoR]/[EoR] mark the start and end of reasoning tokens. Quantitative results are presented in Table 5.

Figure 2: **Anchored Chain-of-Thought (ACoT).** During training, anchor embeddings are passed through the language model head to compute cross-entropy loss on anchor tokens. At inference, the model reasons directly in the latent anchor space without decoding these embeddings, enabling future-aware, out-of-order reasoning since anchors need not correspond to immediate next tokens. Example trace from GSM8K dataset.

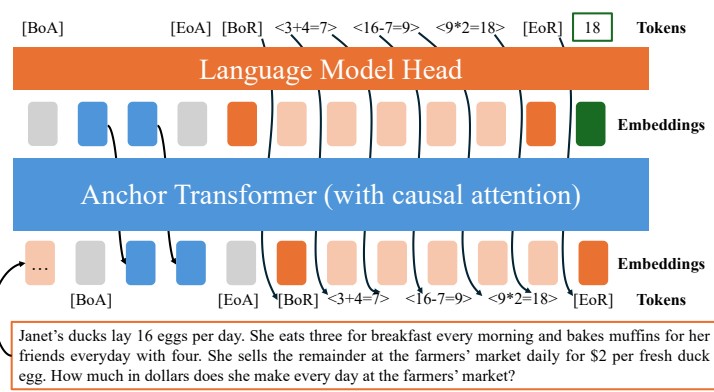

**Results on Math.** On GSM8K, ACoT achieves an accuracy of 45.2%, outperforming compared baselines, including standard CoT (42.9%) and multi-stage finetuning approaches like COCONUT (34.1%). Anchoring improves decoding by treating an ordered subset of reasoning trace tokens, after filtering out punctuation and arithmetic operators $(+, -, \times, \div)$, as anchoring tokens. This guides the ACoT model through important tokens before generating reasoning traces and the final answer.

**Results on Logic.** We anchor using valid nodes from the reasoning traces after pruning conjunctions, articles or adjectives, such as 'every', is', and 'a'. As recommended in COCONUT, we progressively increase the number of [ANT] tokens. For ProsQA, we gradually remove the reasoning steps while inserting [ANT] tokens, which helps enhance logical reasoning (Hao et al., 2024). On a relatively easier benchmark ProntoQA, ACoT achieves 100% accuracy, matching or exceeding prior approaches. On the more challenging ProsQA benchmark, ACoT reaches 97.3%, improving over COCONUT (97.0%) and surpassing CoT variants except iCoT.

Table 5: **Accuracy (%) on Math and Logical Reasoning.** ACoT improves the performance of prior (continuous) latent reasoning methods despite using the same multi-stage training setup as COCONUT. † reported in COCONUT.

| Method | GSM8K | ProntoQA | ProsQA |
|---|---|---|---|
| No-CoT[†] | 16.5 | 93.8 | 76.7 |
| Pause Token[†] | 16.4 | 77.7 | 75.9 |
| CoT[†] | 42.9 | 98.8 | 77.5 |
| iCoT | 30.0 | 99.8 | **98.2** |
| COCONUT[†] | 34.1 | 99.8 | 97.0 |
| *- Pause*[†] | 24.1 | **100** | 96.6 |
| BoLT | 33.6 | – | – |
| **ACoT (ours)** | **45.2** | **100** | 97.3 |

While standard multi-stage training such as in COCONUT leads to an improvement from 98.8% (CoT) to 99.8%, our anchoring mechanism further enhances performance. By incorporating explicit supervision through [ANT], ACoT achieves a perfect accuracy of 100% on this relatively easier benchmark, demonstrating the effectiveness of anchoring in logical reasoning. This study also opens up other intriguing questions and potential directions. Words corresponding to verbs and nouns relate to the concepts of *terminals and syntactic categories* in the Chomsky Hierarchy (formal language theory). We defer additional results and discussion to Appendix C.

# 6 Conclusion

We introduce Anchored Diffusion Language Model (ADLM), a two-stage generative framework that improves diffusion language modeling by leveraging anchor tokens (e.g., low-frequency or important key words). We provide theoretical justification along with strong empirical evidence supporting our results. Our method bridges the gap between diffusion and AR models in likelihood modeling and generated text quality. ADLM significantly reduces test PPL on LM1B and OWT, outperforming previous DLMs in 6 out of 7 zero-shot benchmarks, and, for the first time, enables a diffusion model to outperform AR models in MAUVE score that measures human-like text generation quality. Beyond diffusion, we demonstrate that anchoring is broadly applicable and improves reasoning in AR models. Our Anchored Chain-of-Thought (ACoT) method improves performance on math and logic benchmarks, outperforming existing approaches. These results highlight the impact of anchoring as a general-purpose framework for language modeling and complex reasoning.

**Limitation.** While anchoring yields consistent gain, the definition of token importance is task-specific. We use low frequency tokens or key words as proxies for importance, which may not generalize. Future work may explore adaptive or LLM-guided anchoring for efficient planning and reasoning.

## Acknowledgments

This research has been supported by NSF Grants 2019844 and 2112471, the UT Austin Machine Learning Lab, and computing support on the Vista GPU Cluster through the Center for Generative AI (CGAI) and the Texas Advanced Computing Center (TACC) at UT Austin.

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

# A   Additional Theoretical Results and Proofs

This appendix provides complete theoretical results that were either stated without proof or omitted from the main text due to space constraints. For completeness, we restate key theorems and provide detailed proofs, along with additional theoretical insights. In §A.1, we present the full proof of **Theorem 4.1**. In §A.2, we derive the transition kernel for masked DLM for completeness. Finally, in §A.3, we provide a detailed discussion on the statistical benefits of anchoring in both diffusion and autoregressive models, with an emphasis on sample complexity and likelihood modeling.

## A.1   Proof of Theorem 4.1

We follow the standard discrete diffusion analysis for the NELBO objective (Sohl-Dickstein et al., 2015), but incorporate our anchored parameterization into the divergence computation.

**Theorem A.1** (Anchored Negative Evidence Lower Bound). *Suppose the forward process follows (1), and the inference posterior is parameterized by anchored denoising as in (6). Denote by $\theta$ the collection of parameters of the anchor and denoiser networks, i.e., $\theta = [\psi, \varphi]$. Let $\mathcal{A}(\cdot)$ be an operator that takes a sequence $\mathbf{x} = (\mathbf{x}^l)_{l=1}^L$ as input and returns an important token mixture $\mathbf{y} = (\mathbf{y}^l)_{l=1}^L = \mathcal{A}(\mathbf{x})$ as output. Then, the anchored negative log-likelihood is bounded by:*

$$-\log p_\theta(\mathbf{x}) + \gamma \mathcal{L}_{\mathrm{Anchor}}(\mathbf{x}; \varphi) \leq \mathcal{L}_{\mathrm{ANELBO}}(\mathbf{x}; \psi, \varphi), \quad \text{where}$$

$$\mathcal{L}_{\mathrm{ANELBO}}(\mathbf{x}; \psi, \varphi) := \mathbb{E}_{Z_0 \sim q(\cdot|\mathbf{x})}\left[-\log p_\psi(\mathbf{x}|\mathbf{y}_\varphi(Z_0))\right]$$

$$+ \sum_{i=1}^T \mathbb{E}_{Z_{t(i)} \sim q(\cdot|\mathbf{x})}\left[\lambda_{t(i)} \sum_{l=1}^L \left(\log\langle \mathbf{x}_\psi^l(\mathbf{y}_\varphi(Z_{t(i)})), \mathbf{x}^l\rangle + \gamma \log\langle \mathbf{y}_\varphi^l(Z_{t(i)}), \mathbf{y}^l\rangle\right)\right],$$

*with weight $\lambda_{t(i)} = \frac{(1-\sigma_{t(i)})\alpha_{t(i)} - \alpha_{s(i)}}{1 - \alpha_{t(i)}}$ and $\gamma > 0$.*

*Proof.* We first derive the bound for sequence length $L = 1$; the extension to $L > 1$ is straightforward following the standard analysis (Sohl-Dickstein et al., 2015).

We start from the standard negative log-likelihood:

$$-\log p_\theta(\mathbf{x}) = -\log \int p_\theta(\mathbf{x}, Z_0, \ldots, Z_1)\, d(Z_0, \ldots, Z_1)$$

$$= -\log \int \frac{p_\theta(\mathbf{x}, Z_{0:1})}{q(Z_{0:1}|\mathbf{x})} q(Z_{0:1}|\mathbf{x})\, d(Z_{0:1}).$$

Applying Jensen's inequality yields:

$$-\log p_\theta(\mathbf{x}) \leq \mathbb{E}_{q(Z_{0:1}|\mathbf{x})}\left[-\log p_\theta(\mathbf{x}|Z_0) + \log\frac{q(Z_1|\mathbf{x})}{p_\theta(Z_1)} + \sum_{i=1}^T \log\frac{q(Z_{s(i)}|Z_{t(i)}, \mathbf{x})}{p_\theta(Z_{s(i)}|Z_{t(i)})}\right] := \mathcal{L}_{\mathrm{NELBO}}(\mathbf{x}; \theta)$$

Thus, the NELBO decomposes into:

$$\mathcal{L}_{\mathrm{NELBO}}(\mathbf{x}; \theta) = \mathbb{E}_{q(Z_{0:1}|\mathbf{x})}\left[-\log p_\theta(\mathbf{x}|Z_0) + D_{\mathrm{KL}}\big(q(Z_1|\mathbf{x}) \,\|\, p_\theta(Z_1)\big) + \sum_{i=1}^T D_{\mathrm{KL}}\big(q(Z_{s(i)}|Z_{t(i)}, \mathbf{x}) \,\|\, p_\theta(Z_{s(i)}|Z_{t(i)})\big)\right]$$

Combining the NELBO decomposition with our anchored loss (9) gives:

$$\mathcal{L}_{\mathrm{ANELBO}}(\mathbf{x}; \psi, \varphi) = \mathcal{L}_{\mathrm{NELBO}}(\mathbf{x}; \psi, \varphi) + \gamma \mathcal{L}_{\mathrm{Anchor}}(\mathbf{x}; \varphi)$$

$$= \mathbb{E}_{q(Z_{0:1}|\mathbf{x})}\left[-\log p_\theta(\mathbf{x}|Z_0) + D_{\mathrm{KL}}(q(Z_1|\mathbf{x})\|p_\theta(Z_1)) + \sum_{i=1}^T D_{\mathrm{KL}}(q(Z_{s(i)}|Z_{t(i)}, \mathbf{x})\|p_\theta(Z_{s(i)}|Z_{t(i)}))\right]$$

$$+ \gamma \mathbb{E}_{q(Z_{0:1}|\mathbf{x})}\left[\sum_{i=1}^T D_{\mathrm{KL}}(r(Y_{s(i)}|Z_{t(i)}, \mathbf{y})\|r_\varphi(Y_{s(i)}|Z_{t(i)}))\right]$$

$$= \mathbb{E}_{q(Z_{0:1}|\mathbf{x})}\left[-\log p_\theta(\mathbf{x}|Z_0)\right] + \mathbb{E}_{q(Z_{0:1}|\mathbf{x})}\left[D_{\mathrm{KL}}(q(Z_1|\mathbf{x})\|p_\theta(Z_1))\right]$$

$$+ \mathbb{E}_{q(Z_{0:1}|\mathbf{x})}\left[\sum_{i=1}^T D_{\mathrm{KL}}(q(Z_{s(i)}|Z_{t(i)}, \mathbf{x})\|p_\theta(Z_{s(i)}|Z_{t(i)})) + \gamma D_{\mathrm{KL}}(r(Y_{s(i)}|Z_{t(i)}, \mathbf{y})\|r_\varphi(Y_{s(i)}|Z_{t(i)}))\right]$$

The three terms in the above expression have natural interpretations:

- The first term captures the reconstruction loss at the final step of the reverse process.
- The second term measures the error due to mismatch between the stationary distribution of the forward process and the initial distribution of the reverse process. This vanishes when the reverse process is initialized with a sequence of all masks.
- The third term aggregates the KL divergences across diffusion steps, and encodes the difficulty of denoising masked tokens. Our anchor network aims to reduce this difficulty by enabling early decoding of important tokens.

We now focus on analyzing the third term defined as:

$$\mathcal{L}_{\text{diffusion}}(\mathbf{x}; \psi, \varphi) := \sum_{i=1}^{T} D_{\text{KL}}(q(Z_{s(i)}|Z_{t(i)}, \mathbf{x}) \| p_\theta(Z_{s(i)}|Z_{t(i)})) + \gamma D_{\text{KL}}(r(Y_{s(i)}|Z_{t(i)}, \mathbf{y}) \| r_\varphi(Y_{s(i)}|Z_{t(i)}))$$

Since $\mathbf{y} = \mathcal{A}(\mathbf{x})$, each KL divergence can be computed by splitting into two cases, depending on whether the token $Z_{t(i)}$ is already unmasked.

**Case 1:** $Z_{t(i)} \neq \mathbf{m}$ **(unmasked).** In this case, the diffusion loss for the $i^{th}$ KL-Divergence term:

$$\mathcal{L}_{\text{diffusion}}^{i}(\mathbf{x}; \psi, \varphi) = \mathbb{E}_{q(Z_{s(i)}|Z_{t(i)}, \mathbf{x})} \left[ \log \left( \frac{q(Z_{s(i)}|Z_{t(i)}, \mathbf{x})}{p_\theta(Z_{s(i)}|Z_{t(i)})} \right) \right] + \gamma \mathbb{E}_{r(Y_{s(i)}|Z_{t(i)}, \mathbf{y})} \left[ \log \left( \frac{r(Y_{s(i)}|Z_{t(i)}, \mathbf{y})}{r_\varphi(Y_{s(i)}|Z_{t(i)})} \right) \right]$$

$$= q(Z_{s(i)} = \mathbf{m}|Z_{t(i)} \neq \mathbf{m}, \mathbf{x}) \log \left( \frac{q(Z_{s(i)} = \mathbf{m}|Z_{t(i)} \neq \mathbf{m}, \mathbf{x})}{p_\theta(Z_{s(i)} = \mathbf{m}|Z_{t(i)} \neq \mathbf{m})} \right)$$

$$+ q(Z_{s(i)} = \mathbf{x}|Z_{t(i)} \neq \mathbf{m}, \mathbf{x}) \log \left( \frac{q(Z_{s(i)} = \mathbf{x}|Z_{t(i)} \neq \mathbf{m}, \mathbf{x})}{p_\theta(Z_{s(i)} = \mathbf{x}|Z_{t(i)} \neq \mathbf{m})} \right)$$

$$+ \gamma r(Y_{s(i)} = \mathbf{m}|Z_{t(i)} \neq \mathbf{m}, \mathbf{y}) \log \left( \frac{r(Y_{s(i)} = \mathbf{m}|Z_{t(i)} \neq \mathbf{m}, \mathbf{y})}{r_\varphi(Y_{s(i)} = \mathbf{m}|Z_{t(i)} \neq \mathbf{m})} \right)$$

$$+ \gamma r(Y_{s(i)} = \mathbf{y}|Z_{t(i)} \neq \mathbf{m}, \mathbf{y}) \log \left( \frac{r(Y_{s(i)} = \mathbf{y}|Z_{t(i)} \neq \mathbf{m}, \mathbf{y})}{r_\varphi(Y_{s(i)} = \mathbf{y}|Z_{t(i)} \neq \mathbf{m})} \right)$$

$$= q(Z_{s(i)} = \mathbf{m}|Z_{t(i)} \neq \mathbf{m}, \mathbf{x}) \log \left( \frac{q(Z_{s(i)} = \mathbf{m}|Z_{t(i)} \neq \mathbf{m}, \mathbf{x})}{q(Z_{s(i)} = \mathbf{m}|Z_{t(i)} \neq \mathbf{m}, \mathbf{x}_\psi(\mathbf{y}_\varphi(Z_{t(i)})))} \right)$$

$$+ q(Z_{s(i)} = \mathbf{x}|Z_{t(i)} \neq \mathbf{m}, \mathbf{x}) \log \left( \frac{q(Z_{s(i)} = \mathbf{x}|Z_{t(i)} \neq \mathbf{m}, \mathbf{x})}{q(Z_{s(i)} = \mathbf{x}|Z_{t(i)} \neq \mathbf{m}, \mathbf{x}_\psi(\mathbf{y}_\varphi(Z_{t(i)})))} \right)$$

$$+ \gamma r(Y_{s(i)} = \mathbf{m}|Z_{t(i)} \neq \mathbf{m}, \mathbf{y}) \log \left( \frac{r(Y_{s(i)} = \mathbf{m}|Z_{t(i)} \neq \mathbf{m}, \mathbf{y})}{r(Y_{s(i)} = \mathbf{m}|Y_{t(i)} \neq \mathbf{m}, \mathbf{y}_\varphi(Z_{t(i)}))} \right)$$

$$+ \gamma r(Y_{s(i)} = \mathbf{y}|Z_{t(i)} \neq \mathbf{m}, \mathbf{y}) \log \left( \frac{r(Y_{s(i)} = \mathbf{y}|Z_{t(i)} \neq \mathbf{m}, \mathbf{y})}{r(Y_{s(i)} = \mathbf{y}|Z_{t(i)} \neq \mathbf{m}, \mathbf{y}_\varphi(Z_{t(i)}))} \right)$$

$$= (1 + \gamma)\sigma_{t(i)} \log \left( \frac{\sigma_{t(i)}}{\sigma_{t(i)}} \right) + (1 + \gamma)(1 - \sigma_{t(i)}) \log \left( \frac{(1 - \sigma_{t(i)})}{(1 - \sigma_{t(i)})} \right) = 0.$$

**Implications.** This result demonstrates that when the generative model's reverse transition aligns with the inference posterior for unmasked tokens, the diffusion loss becomes zero. This validates the effectiveness of our two-stage parameterization. The key implications are:

- **Unbiased Learning:** Anchoring introduces no additional bias when its distribution matches the inference posterior (6).
- **Tight Variational Bound:** The ANELBO objective (7) remains a tight bound on the data likelihood, ensuring the theoretical soundness of our formulation.

**Case 2:** $Z_t = \mathbf{m}$ **(masked).** Following the anchored denoising formulation, we obtain:

$$\mathcal{L}_{\text{diffusion}}^i(\mathbf{x};\psi,\varphi) = \mathbb{E}_{q(Z_{s(i)}|Z_{t(i)},\mathbf{x})}\left[\log\left(\frac{q(Z_{s(i)}|Z_{t(i)},\mathbf{x})}{p_\theta(Z_{s(i)}|Z_{t(i)})}\right)\right] + \gamma\mathbb{E}_{r(Y_{s(i)}|Z_{t(i)},\mathbf{y})}\left[\log\left(\frac{r(Y_{s(i)}|Z_{t(i)},\mathbf{y})}{r_\varphi(Y_{s(i)}|Z_{t(i)})}\right)\right]$$

$$= q(Z_{s(i)}=\mathbf{m}|Z_{t(i)}=\mathbf{m},\mathbf{x})\log\left(\frac{q(Z_{s(i)}=\mathbf{m}|Z_{t(i)}=\mathbf{m},\mathbf{x})}{p_\theta(Z_{s(i)}=\mathbf{m}|Z_{t(i)}=\mathbf{m})}\right)$$

$$+ q(Z_{s(i)}=\mathbf{x}|Z_{t(i)}=\mathbf{m},\mathbf{x})\log\left(\frac{q(Z_{s(i)}=\mathbf{x}|Z_{t(i)}=\mathbf{m},\mathbf{x})}{p_\theta(Z_{s(i)}=\mathbf{x}|Z_{t(i)}=\mathbf{m})}\right)$$

$$+ \gamma r(Y_{s(i)}=\mathbf{m}|Z_{t(i)}=\mathbf{m},\mathbf{y})\log\left(\frac{r(Y_{s(i)}=\mathbf{m}|Z_{t(i)}=\mathbf{m},\mathbf{y})}{r_\varphi(Y_{s(i)}=\mathbf{m}|Z_{t(i)}=\mathbf{m})}\right)$$

$$+ \gamma r(Y_{s(i)}=\mathbf{y}|Z_{t(i)}=\mathbf{m},\mathbf{y})\log\left(\frac{r(Y_{s(i)}=\mathbf{y}|Z_{t(i)}=\mathbf{m},\mathbf{y})}{r_\varphi(Y_{s(i)}=\mathbf{y}|Z_{t(i)}=\mathbf{m})}\right)$$

$$= q(Z_{s(i)}=\mathbf{m}|Z_{t(i)}=\mathbf{m},\mathbf{x})\log\left(\frac{q(Z_{s(i)}=\mathbf{m}|Z_{t(i)}=\mathbf{m},\mathbf{x})}{q(Z_{s(i)}=\mathbf{m}|Z_{t(i)}=\mathbf{m},\mathbf{x}_\psi(\mathbf{y}_\varphi(Z_{t(i)})))}\right)$$

$$+ q(Z_{s(i)}=\mathbf{x}|Z_{t(i)}=\mathbf{m},\mathbf{x})\log\left(\frac{q(Z_{s(i)}=\mathbf{x}|Z_{t(i)}=\mathbf{m},\mathbf{x})}{q(Z_{s(i)}=\mathbf{x}|Z_{t(i)}=\mathbf{m},\mathbf{x}_\psi(\mathbf{y}_\varphi(Z_{t(i)})))}\right)$$

$$+ \gamma r(Y_{s(i)}=\mathbf{m}|Z_{t(i)}=\mathbf{m},\mathbf{y})\log\left(\frac{r(Y_{s(i)}=\mathbf{m}|Z_{t(i)}=\mathbf{m},\mathbf{y})}{r(Y_{s(i)}=\mathbf{m}|Z_{t(i)}=\mathbf{m},\mathbf{y}_\varphi(Z_{t(i)}))}\right)$$

$$+ \gamma r(Y_{s(i)}=\mathbf{y}|Z_{t(i)}=\mathbf{m},\mathbf{y})\log\left(\frac{r(Y_{s(i)}=\mathbf{y}|Z_{t(i)}=\mathbf{m},\mathbf{y})}{r(Y_{s(i)}=\mathbf{y}|Z_{t(i)}=\mathbf{m},\mathbf{y}_\varphi(Z_{t(i)}))}\right)$$

$$= \left(\frac{1-\alpha_s-\sigma_t\alpha_t}{1-\alpha_t}\right)\log\left[\frac{\left(\frac{1-\alpha_s-\sigma_t\alpha_t}{1-\alpha_t}\right)}{\left(\frac{1-\alpha_s-\sigma_t\alpha_t}{1-\alpha_t}\right)}\right] + \left(\frac{\alpha_s-\alpha_t+\alpha_t\sigma_t}{1-\alpha_t}\right)\log\left[\frac{\left(\frac{\alpha_s-\alpha_t+\alpha_t\sigma_t}{1-\alpha_t}\right)}{\left(\frac{\alpha_s-\alpha_t+\alpha_t\sigma_t}{1-\alpha_t}\right)\langle\mathbf{x}_\psi(\mathbf{y}_\varphi(Z_t))),\mathbf{x}\rangle}\right]$$

$$+ \gamma\left(\frac{1-\alpha_s-\sigma_t\alpha_t}{1-\alpha_t}\right)\log\left[\frac{\left(\frac{1-\alpha_s-\sigma_t\alpha_t}{1-\alpha_t}\right)}{\left(\frac{1-\alpha_s-\sigma_t\alpha_t}{1-\alpha_t}\right)}\right] + \gamma\left(\frac{\alpha_s-\alpha_t+\alpha_t\sigma_t}{1-\alpha_t}\right)\log\left[\frac{\left(\frac{\alpha_s-\alpha_t+\alpha_t\sigma_t}{1-\alpha_t}\right)}{\left(\frac{\alpha_s-\alpha_t+\alpha_t\sigma_t}{1-\alpha_t}\right)\langle\mathbf{y}_\varphi(Z_t))),\mathbf{y}\rangle}\right]$$

$$= \left(\frac{(1-\sigma_{t(i)})\alpha_{t(i)}-\alpha_{s(i)}}{1-\alpha_{t(i)}}\right)\left[\log\langle\mathbf{x}_\psi(\mathbf{y}_\varphi(Z_{t(i)})),\mathbf{x}\rangle + \gamma\log\langle\mathbf{y}_\varphi(Z_{t(i)}),\mathbf{y}\rangle\right]$$

Combining Case 1 and Case 2, we conclude the proof. $\square$

**Summary.** The complete derivation of the ANELBO (7) formally establishes the theoretical soundness of our two-stage ADLM parameterization. It confirms that anchoring introduces no additional bias and remains a tight bound on the data manifold. When the anchor and denoising networks are properly aligned with the inference posterior, the KL terms decompose nicely under our parameterization. This leads to a variational bound that reflects both token-level reconstruction and anchor-level guidance, enabling effective learning in large-scale diffusion language models.

## A.2 Derivation of Absorbing Transition Kernel

This derivation is a special case of the D3PM framework (Austin et al., 2021) applied to masked diffusion language modeling. We include a simplified proof here for completeness.

Recall from §2.2 that the forward noising process is defined as:

$$q(\mathbf{z}_t|\mathbf{x}) \coloneqq \mathrm{Cat}\left(\mathbf{z}_t;\alpha_t\mathbf{x}+(1-\alpha_t)\mathbf{m}\right), \quad t\in\left\{\frac{1}{T},\frac{2}{T},\ldots,1\right\}, \tag{10}$$

where $\mathbf{m}$ denotes the mask token distribution and $\alpha_t \in [0, 1]$ controls the corruption level at time $t$.

We aim to derive the transition kernel:

$$q(\mathbf{z}_t|\mathbf{z}_s) = \text{Cat}\left(\mathbf{z}_t; \alpha_{t|s}\mathbf{z}_s + (1 - \alpha_{t|s})\mathbf{m}\right), \tag{11}$$

where $\alpha_{t|s} := \alpha_t/\alpha_s$ for $t > s$.

**Law of Total Probability.** We begin by marginalizing over $Z_s$:

$$\begin{aligned}
q(\mathbf{z}_t|\mathbf{x}) &= \sum_{\mathbf{z}_s} q(\mathbf{z}_t|\mathbf{z}_s, \mathbf{x}) \, q(\mathbf{z}_s|\mathbf{x}) \\
&= q(\mathbf{z}_t|\mathbf{z}_s = \mathbf{m}) \, q(\mathbf{z}_s = \mathbf{m}|\mathbf{x}) + q(\mathbf{z}_t|\mathbf{z}_s = \mathbf{x}) \, q(\mathbf{z}_s = \mathbf{x}|\mathbf{x}).
\end{aligned}$$

**Simplifying the Components.** From the forward process in Eq. (10), we know:

$$\begin{aligned}
q(\mathbf{z}_s = \mathbf{x}|\mathbf{x}) &= \alpha_s, \\
q(\mathbf{z}_s = \mathbf{m}|\mathbf{x}) &= 1 - \alpha_s.
\end{aligned}$$

Let $\alpha := q(\mathbf{z}_t = \mathbf{x}|\mathbf{z}_s = \mathbf{x})$, and note that since $\mathbf{m}$ is an absorbing state, we have:

$$\begin{aligned}
q(\mathbf{z}_t = \mathbf{x}|\mathbf{z}_s = \mathbf{m}) &= 0, \\
q(\mathbf{z}_t = \mathbf{m}|\mathbf{z}_s = \mathbf{m}) &= 1.
\end{aligned}$$

Then, the marginal probability of $\mathbf{z}_t = \mathbf{m}$ given $\mathbf{x}$ becomes:

$$\begin{aligned}
q(\mathbf{z}_t = \mathbf{m}|\mathbf{x}) &= q(\mathbf{z}_t = \mathbf{m}|\mathbf{z}_s = \mathbf{m}) \, q(\mathbf{z}_s = \mathbf{m}|\mathbf{x}) + q(\mathbf{z}_t = \mathbf{m}|\mathbf{z}_s = \mathbf{x}) \, q(\mathbf{z}_s = \mathbf{x}|\mathbf{x}) \\
&= (1 - \alpha_s) + (1 - \alpha) \, \alpha_s \\
&= 1 - \alpha\alpha_s.
\end{aligned}$$

From Eq. (10), we also know:

$$q(\mathbf{z}_t = \mathbf{m}|\mathbf{x}) = 1 - \alpha_t.$$

Equating the two expressions:

$$1 - \alpha\alpha_s = 1 - \alpha_t \quad \Rightarrow \quad \alpha = \frac{\alpha_t}{\alpha_s} = \alpha_{t|s}.$$

Thus, we have shown:

$$q(\mathbf{z}_t|\mathbf{z}_s = \mathbf{z}_s) = \text{Cat}\left(\mathbf{z}_t; \alpha_{t|s}\mathbf{z}_s + (1 - \alpha_{t|s})\mathbf{m}\right),$$

which completes the derivation of the absorbing transition kernel.

### A.3 Anchored Graphical Model Analysis

The foundational principle behind both AR and DLM pre-training is Maximum Likelihood Estimation (MLE), which optimizes their respective log-likelihood objectives. MLE has been extensively studied in the context of graphical models (Koller & Friedman, 2009), offering a principled framework to analyze expressiveness, tractability, and sample complexity. In this section, we recast AR and diffusion training as instances of learning in Directed Graphical Models (DAGs) and use this perspective to formally analyze our anchoring mechanism. We show that anchoring—conditioning only on a small, important subset of tokens—leads to significant reduction in parameters and sample complexity. While related ideas are well-established in probabilistic modeling, we demonstrate their effectiveness in large-scale language model pre-training (§5.1.1) and fine-tuning (§5.2.1) tasks.

**Setup.** Given a sample $\mathbf{x} = \left(\mathbf{x}^1, \cdots, \mathbf{x}^L\right)$, we consider a DAG denoted by $\mathcal{G} = (L, E)$, where $L = \{1, 2, \ldots, L\}$ denotes the set of nodes (each corresponding to a token position in the sequence), and $E$ denotes the set of directed edges representing conditional dependencies. Each node $\mathbf{x}^l$ takes a discrete value from a vocabulary $\mathcal{V}$ of size $K$, so $\mathbf{x}^l \in \mathcal{V}$. Let $\pi_l \subseteq \{1, \ldots, L\} \setminus \{l\}$ denote the set of parent indices of node $l$, and define $\mathbf{x}^{\pi_l} = \{\mathbf{x}^j : j \in \pi_l\}$ to be the corresponding parent tokens.

In the following, we analyze the sample complexity of learning graphical models for language, comparing standard approaches with our proposed anchored models. Our focus is to demonstrate that anchoring—by conditioning on a small, important subset of tokens—can dramatically reduce the number of parameters and training samples required.

**Assumption A.2.** *Suppose the following properties hold.*

- *Each conditional distribution $p(\mathbf{x}^l|\cdot)$ is modeled as a categorical distribution.*
- *The model is parameterized by Conditional Probability Tables (CPTs); that is, a distinct parameter is assigned to each possible configuration of the conditioning set.*
- *Anchor sets $\pi_l \subset \{1,\ldots,L\} \setminus \{l\}$ are fixed and of bounded size $|\pi_l| \leq d$, with $d \ll L$.*

**Proposition A.3** (Reduced Sample Complexity via Anchoring). *Let $\mathbf{x} = (\mathbf{x}^1,\ldots,\mathbf{x}^L)$ be a sequence of discrete random variables, each taking values in a finite vocabulary $\mathcal{V}$ of size $K$, i.e., $\mathbf{x}^l \in \mathcal{V}$ with $|\mathcal{V}| = K$. Suppose we are given $N$ i.i.d. samples $\{\mathbf{x}_i\}_{i=1}^N \sim q$, and **Assumption A.2** holds. Then the sample complexity of MLE under different modeling paradigms is as follows:*

1. ***Standard Autoregressive Modeling:*** *Since each token $\mathbf{x}^l$ is conditioned on all previous tokens $\mathbf{x}^{1:l-1}$, the total number of parameters is $\mathcal{O}(K^L)$, resulting in a sample complexity of $\mathcal{O}(K^L)$.*
2. ***Standard Diffusion Modeling:*** *Each masked token $\mathbf{x}^l$ is conditioned on all other tokens $\mathbf{x} \setminus \mathbf{x}^l$. The per-token parameter count is $\mathcal{O}(K^L)$, leading to total sample complexity of $\mathcal{O}(LK^L)$.*
3. ***Anchored Autoregressive Modeling (A2R):*** *Since each token $\mathbf{x}^l$ is conditioned only on a fixed-size anchor set $\mathbf{x}^{\pi_l}$, the number of parameters per conditional is $\mathcal{O}(K^{d+1})$, giving total sample complexity $\mathcal{O}(LK^{d+1})$.*
4. ***Anchored Diffusion Language Modeling (ADLM):*** *Each masked token $\mathbf{x}^l$ is predicted using only anchor tokens $\mathbf{x}^{\pi_l} \setminus \{\mathbf{x}^l\}$. The per-token parameter count becomes $\mathcal{O}(K^{d+1})$, resulting in a total sample complexity of $\mathcal{O}(LK^{d+1})$.*

**Implications.** Assuming the existence of important tokens in an anchor set of fixed cardinality $d$, anchored modeling achieves exponential reductions in sample complexity without sacrificing model expressiveness or decoding fidelity. A2R improves upon standard AR models by reducing the sample complexity from $\mathcal{O}(K^L)$ to $\mathcal{O}(LK^{d+1})$. ADLM offers an analogous benefit, reducing the sample complexity from $\mathcal{O}(LK^L)$ to $\mathcal{O}(LK^{d+1})$. These results highlight the theoretical advantage of anchoring in high-dimensional structured prediction settings, particularly for language modeling.

**Remark A.4.** *Consider a sequence of length $L = 1024$ and vocabulary size $K = 50257$ (as used in MDLM training) for a line network $\mathbf{x}^1 \to \mathbf{x}^2 \to \cdots \to \mathbf{x}^L$. Under standard autoregressive modeling, the total number of parameters in CPTs required to model the full joint distribution is $\mathcal{O}(K^L) = \mathcal{O}(50257^{1024})$, which is computationally intractable to estimate.*

*In contrast, under the anchored autoregressive model (A2R) with a small anchor set size, e.g., $d = 1$ for the line network, the total number of parameters reduces to $\mathcal{O}(LK^2) = \mathcal{O}(1024 \times 50257^2)$, which is within the scale of modern large language models.*

*Similarly, for diffusion models, anchoring reduces the per-token parameter complexity from $K^L = 50257^{1024}$ to $K^{d+1} = 50257^2$, leading to a sample complexity of $\mathcal{O}(LK^{d+1}) = \mathcal{O}(1024 \times 50257^2)$.*

*These exponential savings illustrate that anchored modeling makes otherwise intractable parameter estimation feasible in large-scale (diffusion/AR) language modeling.*

We provide detailed derivations and discussion of these results in the subsequent sections A.3.1 and A.3.2. Our anchored training procedure also has an interpretation of expectation-maximization (EM) (Dempster et al., 1977), which we discuss in A.3.3. In A.3.4, we provide an example to show how anchoring helps improve the likelihood of decoding important tokens.

### A.3.1 Sample Complexity in Standard Training

Assume we are given $N$ i.i.d. samples $\{\mathbf{x}_i\}_{i=1}^N$ drawn from an unknown distribution $q$ over sequences $\mathbf{x}_i \in \mathcal{V}^L$, where $\mathcal{V}$ is a vocabulary of size $K$. Our goal is to estimate the parameters of a conditional probability model $p(\mathbf{x}|\theta)$, where $\theta = [\psi, \varphi]$, using MLE. The standard MLE objective is defined as:

$$\mathcal{L}_{\mathrm{MLE}}(\psi, \varphi) = \sum_{i=1}^N \log p(\mathbf{x}_i|\psi, \varphi).$$

Since the structure of the underlying graphical model $\mathcal{G}$ is typically unknown, a common modeling assumption is to use a fully autoregressive factorization of the joint distribution. This leads to the

following objective:

$$\mathcal{L}_{\mathrm{AR}}(\psi, \varphi) = \sum_{i=1}^{N} \left[ \log p(\mathbf{x}_i^1 | \psi, \varphi) + \sum_{l=2}^{L} \log p(\mathbf{x}_i^l | \mathbf{x}_i^{1:l-1}, \psi, \varphi) \right],$$

which models the joint probability by conditioning each token on all previous tokens.

Alternatively, in masked diffusion language modeling, the likelihood of the missing token is computed by conditioning on all tokens except the masked token:

$$\mathcal{L}_{\mathrm{Diffusion}}(\psi, \varphi) = \sum_{i=1}^{N} \left[ \sum_{l=1}^{L} \log p(\mathbf{x}_i^l | \mathbf{x}_i \setminus \mathbf{x}_i^l, \psi, \varphi) \cdot \mathbf{1}_{\{\mathbf{x}_i^l = \mathbf{m}\}} \right],$$

where $\mathbf{m}$ denotes a masked token and $\mathbf{x}_i \setminus \mathbf{x}_i^l$ is the set of all other tokens in the sequence.

In both formulations, each conditional probability is modeled as a categorical distribution over $\mathcal{V}$ and is thus associated with a CPT. Learning such a model amounts to estimating these CPTs. The number of parameters required for each CPT depends exponentially on the size of its conditioning set. For example:

- To model $q(\mathbf{x}^1)$, we need $K$ parameters to define $p(\mathbf{x}^1 | \theta)$.
- For $q(\mathbf{x}^2 | \mathbf{x}^1)$, we require $K^2$ parameters, one categorical distribution for each value of $\mathbf{x}^1$.
- In general, for $q(\mathbf{x}^l | \mathbf{x}^{1:l-1})$, the CPT size is $K^l$, as we require a distribution over $K$ values for each of the $K^{l-1}$ configurations of the conditioning context $\mathbf{x}^{1:l-1}$.

Summing over all positions yields the total number of parameters in the model:

$$\sum_{l=1}^{L} K^l = \frac{K(K^L - 1)}{K - 1}.$$

In this tabular setting, it is well known that the sample complexity of MLE grows at least linearly with the number of parameters in order to guarantee accurate estimation. Thus, the sample complexity of learning such a model is $\mathcal{O}(K^L)$, which becomes infeasible even for modest values of $L$ and $K$. For example, with $L = 1024$ and $K = 50257$ (as in GPT-2's vocabulary size), the number of parameters is on the order of $50257^{1024}$, which is computationally intractable.

This exponential order highlights the need for structure-aware modeling techniques, such as *anchoring*, which constrain the dependency structure and significantly reduce the number of learnable parameters. In the following sections, we show how anchoring enables more sample-efficient learning by focusing on a subset of important tokens that govern the generative structure of the data.

### A.3.2 Reduced Sample Complexity via Anchored Modeling

A key motivation for our approach is the observation that a sentence can often be accurately decoded given a small set of important tokens. We propose to leverage this property by computing the likelihood of missing tokens while conditioning only on a carefully selected subset of important tokens, referred to as *anchor tokens*, rather than the full context. This design leads to a substantial reduction in the sample complexity required for maximum likelihood estimation.

Identifying the most informative tokens is an exciting problem that has been studied extensively in the NLP literature (Linzen et al., 2016; Li et al., 2016; Sundararajan et al., 2017; Clark et al., 2019; Tenney et al., 2019; Khandelwal et al., 2020). In this work, we adopt a simple yet effective information-theoretic strategy: tokens with low marginal frequency in the given sample tend to carry more information (§3). Hence, we treat such low-frequency tokens as candidates for anchoring in language modeling tasks. Our empirical results support this strategy across two commonly used generative modeling benchmarks: (1) LM1B (Chelba et al., 2013) and (2) OWT (Gokaslan & Cohen, 2019), and seven downstream evaluation benchmarks (§5). We further demonstrate the benefits of anchoring in AR models by exploring alternate anchoring strategies:

- In logical reasoning benchmarks such as ProntoQA (Saparov & He, 2023) and ProsQA (Hao et al., 2024), root nodes of reasoning traces reliably serve as effective anchors.

- In mathematical reasoning benchmarks like GSM8K, we find that early steps in reasoning traces—excluding arithmetic operators (such as $+$, $-$, $\times$, and $\div$)—contain the important information needed to derive the correct answer.

**Anchored Autoregressive Modeling.** Incorporating anchored tokens into our graphical model framework yields a more compact parameterization of the conditional likelihood. Specifically, we introduce *Anchored Autoregressive* (A2R) modeling, which modifies the conditioning structure in the likelihood. The training objective becomes:

$$\mathcal{L}_{\mathrm{A2R}}(\psi) = -\sum_{i=1}^{N} \left[ \log p(\mathbf{x}_i^1|\psi) + \sum_{l=2}^{L} \log p(\mathbf{x}_i^l|\mathbf{x}_i^{\pi_l}, \psi) \right],$$

where $\mathbf{x}_i^{\pi_l}$ denotes the set of parent nodes treated as anchor tokens.

This formulation significantly reduces the number of parameters and thus the sample complexity. Consider a line graph where each token depends only on its immediate predecessor: $\mathbf{x}^1 \rightarrow \mathbf{x}^2 \rightarrow \cdots \rightarrow \mathbf{x}^L$. In standard AR modeling, $\mathbf{x}^L$ is conditioned on all preceding tokens, requiring $K^L$ parameters and yielding a sample complexity of $\mathcal{O}(K^L)$. In contrast, A2R limits dependencies to single-token anchors (i.e., $|\pi_l| = 1$), so each conditional requires only $\mathcal{O}(K^2)$ parameters. The total sample complexity becomes $\mathcal{O}(LK^2)$, a drastic reduction from $\mathcal{O}(K^L)$. For example, with sequence length $L = 1024$ and vocabulary size $K = 50257$ (as in GPT-2), standard modeling scales as $\mathcal{O}(50257^{1024})$, whereas A2R scales as $\mathcal{O}(1024 \times 50257^2)$.

**Anchored Diffusion Language Modeling.** The anchoring mechanism generalizes to the diffusion setting via our ADLM parameterization (§3). The corresponding training objective is:

$$\mathcal{L}_{\mathrm{ADLM}}(\psi) = -\sum_{i=1}^{N} \left[ \sum_{l=1}^{L} \log p(\mathbf{x}_i^l|\mathbf{x}_i^{\pi_l} \setminus \mathbf{x}_i^l, \psi) \cdot \mathbf{1}_{\{\mathbf{x}_i^l = \mathbf{m}\}} \right],$$

where $\mathbf{m}$ is the 'mask' token, and $\mathbf{1}_{\{\mathbf{x}_i^l = \mathbf{m}\}}$ ensures that only masked tokens contribute to the overall loss. As in A2R, anchoring reduces the size of the conditioning context, leading to significantly lower sample complexity in estimating the parameters of the denoising model.

**Learning Anchors via KL-Divergence.** While anchored modeling reduces sample complexity for the decoder, learning the anchor network could be challenging because the number of possible token subsets grows exponentially with sequence length. To address this, we use a simple strategy—such as selecting low-frequency tokens—for anchoring, which are easy to compute from the input sequence.

We introduce a regularization term based on KL-divergence that encourages the learned anchor distribution $r_\varphi(\mathbf{y}|\mathbf{x})$ to align with the chosen anchor distribution $r(\mathbf{y}|\mathbf{x})$. Since the model doesn't have access to the actual parents during inference, we replace $\mathbf{x}_i^{\pi_l}$ with predicted anchor tokens—$\mathbf{y}_\varphi(\mathbf{x}_i^{1:l-1})$ for AR and $\mathbf{y}_\varphi(\mathbf{x}_i \setminus \mathbf{x}_i^l)$ for diffusion—leading to the following regularized objective:

$$\arg\min_{\psi,\varphi} \left[ \mathcal{L}_{\mathrm{A2R/ADLM}}(\psi, \varphi) + \gamma \mathrm{D}_{\mathrm{KL}}\left( r(\mathbf{y}|\mathbf{x}) \| r_\varphi(\mathbf{y}|\mathbf{x}) \right) \right],$$

where $\gamma > 0$ controls the anchor strength. This objective encourages the model to first decode the important tokens before reconstructing the rest of the sequence.

**Summary.** By leveraging inductive biases about the distribution of important tokens within sequences, our anchoring approach achieves substantial improvements in sample complexity. Crucially, it avoids the combinatorial explosion of full-context modeling while delivering strong performance on generative modeling and complex reasoning tasks. Importantly, the proposed method is theoretically grounded, computationally tractable, and readily scalable to modern language models.

We note that exact inference in graphical models is NP-hard in the worst case (Koller & Friedman, 2009). However, many real-world scenarios do not exhibit worst-case behavior. As a result, such problems can often be effectively tackled using approximate inference techniques that operate over functional representations, rather than tabular. While the tabular form remains sufficient to highlight the importance of anchoring, more expressive functional representations can further complement and enhance its effectiveness.

### A.3.3 Interpretation Through Expectation-Maximization

Our anchored training procedure can be naturally interpreted through the lens of the EM algorithm (Dempster et al., 1977), a classical framework for MLE in models with latent variables. In our setting, the anchor tokens $\mathbf{y}$ act as latent variables. The anchor network (parameterized by $\varphi$) estimates a soft distribution over important tokens from the observed sequence $\mathbf{x}$ (E-step), and the denoising model (parameterized by $\psi$) uses this distribution to reconstruct the full sequence (M-step). While classical EM alternates between these steps, this is computationally expensive for large language models due to the doubled training time. Instead, we unify both steps into a single ANELBO training objective, which allows efficient end-to-end training (approximately two months to train ADLM). Although our implementation does not explicitly perform separate E and M steps during training, the EM interpretation offers valuable theoretical insight for future research.

We now formalize this interpretation. Consider a parameterized model $p(\mathbf{x}|\theta)$ with parameters $\theta = [\psi, \varphi]$, where $\psi$ governs the denoising (generative) model and $\varphi$ governs the anchor network. Recall from anchor transition function in §3 that $r(\mathbf{y}|\mathbf{x}, \varphi) = r(\mathbf{y}|\mathbf{x})$ when $\mathbf{z}_t \neq \mathbf{m}$. Therefore, minimizing the ANELBO objective is equivalent to maximizing the log likelihood in the fully-observed DAG[1]. Thus, the task simplifies to maximizing the marginal likelihood of observed data:

$$\psi^*, \varphi^* = \arg\min_{\psi,\varphi} \mathcal{L}(\mathbf{x}; \psi, \varphi) = \arg\max_{\psi,\varphi} \log p(\mathbf{x}|\psi, \varphi). \tag{12}$$

In practice, this objective is generally non-convex and does not have a closed-form solution. However, meaningful convergence analysis can be carried out under simplifying assumptions (Neal & Hinton, 1998; Koller & Friedman, 2009; Kwon et al., 2024). In the following, we assume a specific update rule analogous to EM and show that it leads to monotonic improvement of the ANELBO objective.

**Assumption A.5.** *Suppose the parameter updates follow the anchored EM update rule:*

$$\psi_{i+1}, \varphi_{i+1} = \arg\min_{\psi,\varphi} - \sum_{\mathbf{y} \in \mathcal{V}} r(\mathbf{y}|\mathbf{x}, \varphi_i) \log p(\mathbf{x}, \mathbf{y}|\psi, \varphi),$$

*where $r(\mathbf{y}|\mathbf{x}, \varphi_i)$ is the anchor distribution predicted by the anchor network at the current iterate $i$, and $p(\mathbf{x}, \mathbf{y}|\psi, \varphi)$ is the joint likelihood of observed variables and anchor tokens.*

**Theorem A.6** (Monotonic Improvement of Anchored Likelihood)**.** *Suppose **Assumption A.5** holds. Let the anchor distribution be parameterized as $p(\mathbf{y}|\mathbf{x}, \psi, \varphi) = r(\mathbf{y}|\mathbf{x}, \varphi)$. Then the ANELBO objective undergoes monotonic improvement:*

$$\mathcal{L}(\mathbf{x}; \psi_{i+1}, \varphi_{i+1}) \leq \mathcal{L}(\mathbf{x}; \psi_i, \varphi_i).$$

**Implications.** Theorem A.6 guarantees that the anchored EM procedure produces non-increasing negative log-likelihood at each iteration, thereby ensuring stability and convergence to a first-order stationary point. Notably, this convergence behavior emerges even though the anchor tokens are unobserved (Kwon et al., 2024); they are estimated in the E-step and subsequently used in a regularized M-step to update both the denoiser and anchor network parameters. In practice, we approximate these two steps using a single gradient update for efficient scaling.

*Proof.* **Optimal distribution over anchors.** We begin by conditioning the standard negative log-likelihood on the anchor tokens $\mathbf{y} \in \mathcal{V}$. By introducing an arbitrary distribution $r(\mathbf{y})$ over the anchors,

---

[1]We note that our analysis can be easily extended to the case where some tokens have been masked. This follows from conditioning on the latent variables similar to diffusion models (Sohl-Dickstein et al., 2015). We also refer to our derivation in Appendix A.1 for extending this analysis to latent variables.

we apply Jensen's inequality to obtain an upper bound:

$$\mathcal{L}(\mathbf{x}; \psi, \varphi) = -\log p(\mathbf{x}|\psi, \varphi) = -\log \sum_{\mathbf{y} \in \mathcal{V}} p(\mathbf{x}, \mathbf{y}|\psi, \varphi)$$

$$= -\log \sum_{\mathbf{y} \in \mathcal{V}} \left( \frac{r(\mathbf{y})}{r(\mathbf{y})} \right) p(\mathbf{x}, \mathbf{y}|\psi, \varphi)$$

$$\leq -\sum_{\mathbf{y} \in \mathcal{V}} r(\mathbf{y}) \log \left( \frac{p(\mathbf{x}, \mathbf{y}|\psi, \varphi)}{r(\mathbf{y})} \right) \quad \text{(Jensen's inequality)}$$

$$= -\left( \underbrace{\sum_{\mathbf{y} \in \mathcal{V}} r(\mathbf{y}) \log p(\mathbf{x}, \mathbf{y}|\psi, \varphi)}_{\text{Expected Energy}} - \underbrace{\sum_{\mathbf{y} \in \mathcal{V}} r(\mathbf{y}) \log r(\mathbf{y})}_{\text{Entropy}} \right)$$

$$:= \mathcal{F}(r, [\psi, \varphi]).$$

The upper bound $\mathcal{F}(r, [\psi, \varphi])$ is referred to as the *free energy*. The free energy is minimized in two steps. First, we minimize with respect to $r$ for a fixed $\psi, \varphi$, and then use the optimal $r^*$ to optimize $\psi, \varphi$. We now simplify this expression to identify the optimal choice of $r$:

$$\mathcal{F}(r, [\psi, \varphi]) = -\sum_{\mathbf{y} \in \mathcal{V}} r(\mathbf{y}) \log p(\mathbf{x}, \mathbf{y}|\psi, \varphi) + \sum_{\mathbf{y} \in \mathcal{V}} r(\mathbf{y}) \log r(\mathbf{y})$$

$$= -\sum_{\mathbf{y} \in \mathcal{V}} r(\mathbf{y}) \log p(\mathbf{y}|\mathbf{x}, \psi, \varphi) - \sum_{\mathbf{y} \in \mathcal{V}} r(\mathbf{y}) \log p(\mathbf{x}|\psi, \varphi) + \sum_{\mathbf{y} \in \mathcal{V}} r(\mathbf{y}) \log r(\mathbf{y})$$

$$= -\sum_{\mathbf{y} \in \mathcal{V}} r(\mathbf{y}) \log p(\mathbf{x}|\psi, \varphi) + \sum_{\mathbf{y} \in \mathcal{V}} r(\mathbf{y}) \log \left( \frac{r(\mathbf{y})}{p(\mathbf{y}|\mathbf{x}, \psi, \varphi)} \right)$$

$$= -\sum_{\mathbf{y} \in \mathcal{V}} r(\mathbf{y}) \log p(\mathbf{x}|\psi, \varphi) + \mathrm{D}_{\mathrm{KL}} \left( r(\mathbf{y}) || p(\mathbf{y}|\mathbf{x}, \psi, \varphi) \right)$$

$$= -\log p(\mathbf{x}|\psi, \varphi) + \mathrm{D}_{\mathrm{KL}} \left( r(\mathbf{y}) || r(\mathbf{y}|\mathbf{x}, \varphi) \right) \quad \text{(since } p(\mathbf{y}|\mathbf{x}, \psi, \varphi) = r(\mathbf{y}|\mathbf{x}, \varphi)).$$

The inequality becomes an equality when $\mathrm{D}_{\mathrm{KL}} \left( r(\mathbf{y}) || r(\mathbf{y}|\mathbf{x}, \varphi) \right) = 0$. Therefore, the minimum is attained when $r^*(\mathbf{y}) = r(\mathbf{y}|\mathbf{x}, \varphi)$ for a fixed $[\psi, \varphi]$, motivating our choice of anchor transitions in §3.

Next, we show that the anchored log likelihood improves monotonically under the anchored EM procedure using the optimal distribution $r^*(\mathbf{y})$.

**Decomposition of the log-likelihood.** Using the identity

$$p(\mathbf{x}, \mathbf{y}|\psi, \varphi) = p(\mathbf{y}|\mathbf{x}, \psi, \varphi)\, p(\mathbf{x}|\psi, \varphi),$$

we can write:

$$\mathcal{F}(r^*, [\psi, \varphi]) = -\log p(\mathbf{x}|\psi, \varphi) = -\log p(\mathbf{x}, \mathbf{y}|\psi, \varphi) + \log p(\mathbf{y}|\mathbf{x}, \psi, \varphi).$$

**Expectation over $r^*(\mathbf{y})$.** Multiplying both sides by $r(\mathbf{y}|\mathbf{x}, \varphi_i)$ and summing over $\mathbf{y} \in \mathcal{V}$:

$$-\sum_{\mathbf{y} \in \mathcal{V}} r(\mathbf{y}|\mathbf{x}, \varphi_i) \log p(\mathbf{x}|\psi, \varphi) = -\sum_{\mathbf{y} \in \mathcal{V}} r(\mathbf{y}|\mathbf{x}, \varphi_i) \log p(\mathbf{x}, \mathbf{y}|\psi, \varphi) + \sum_{\mathbf{y} \in \mathcal{V}} r(\mathbf{y}|\mathbf{x}, \varphi_i) \log p(\mathbf{y}|\mathbf{x}, \psi, \varphi).$$

Since $\log p(\mathbf{x}|\psi, \varphi)$ is constant with respect to $\mathbf{y}$, we simplify:

$$-\log p(\mathbf{x}|\psi, \varphi) = -\sum_{\mathbf{y} \in \mathcal{V}} r(\mathbf{y}|\mathbf{x}, \varphi_i) \log p(\mathbf{x}, \mathbf{y}|\psi, \varphi) + \sum_{\mathbf{y} \in \mathcal{V}} r(\mathbf{y}|\mathbf{x}, \varphi_i) \log p(\mathbf{y}|\mathbf{x}, \psi, \varphi).$$

**Difference of log-likelihoods.** We now compute the difference between two consecutive iterations:

$$
-\log p(\mathbf{x}|\psi_{i+1}, \varphi_{i+1}) + \log p(\mathbf{x}|\psi_i, \varphi_i)
$$

$$
= -\sum_{\mathbf{y}} r(\mathbf{y}|\mathbf{x}, \varphi_i) \log p(\mathbf{x}, \mathbf{y}|\psi_{i+1}, \varphi_{i+1}) + \sum_{\mathbf{y}} r(\mathbf{y}|\mathbf{x}, \varphi_i) \log p(\mathbf{y}|\mathbf{x}, \psi_{i+1}, \varphi_{i+1})
$$

$$
+ \sum_{\mathbf{y}} r(\mathbf{y}|\mathbf{x}, \varphi_i) \log p(\mathbf{x}, \mathbf{y}|\psi_i, \varphi_i) - \sum_{\mathbf{y}} r(\mathbf{y}|\mathbf{x}, \varphi_i) \log p(\mathbf{y}|\mathbf{x}, \psi_i, \varphi_i)
$$

$$
= \left[ -\sum_{\mathbf{y}} r(\mathbf{y}|\mathbf{x}, \varphi_i) \log p(\mathbf{x}, \mathbf{y}|\psi_{i+1}, \varphi_{i+1}) + \sum_{\mathbf{y}} r(\mathbf{y}|\mathbf{x}, \varphi_i) \log p(\mathbf{x}, \mathbf{y}|\psi_i, \varphi_i) \right]
$$

$$
+ \sum_{\mathbf{y}} r(\mathbf{y}|\mathbf{x}, \varphi_i) \log \left( \frac{p(\mathbf{y}|\mathbf{x}, \psi_{i+1}, \varphi_{i+1})}{p(\mathbf{y}|\mathbf{x}, \psi_i, \varphi_i)} \right).
$$

**Applying Jensen's Inequality.** Using Jensen's inequality on the last term:

$$
\sum_{\mathbf{y}} r(\mathbf{y}|\mathbf{x}, \varphi_i) \log \left( \frac{p(\mathbf{y}|\mathbf{x}, \psi_{i+1}, \varphi_{i+1})}{p(\mathbf{y}|\mathbf{x}, \psi_i, \varphi_i)} \right) \leq \log \sum_{\mathbf{y}} r(\mathbf{y}|\mathbf{x}, \varphi_i) \left( \frac{p(\mathbf{y}|\mathbf{x}, \psi_{i+1}, \varphi_{i+1})}{p(\mathbf{y}|\mathbf{x}, \psi_i, \varphi_i)} \right)
$$

$$
= \log \sum_{\mathbf{y}} p(\mathbf{y}|\mathbf{x}, \psi_{i+1}, \varphi_{i+1})
$$

$$
= \log 1 = 0,
$$

where we use the fact that $r(\mathbf{y}|\mathbf{x}, \varphi_i) = p(\mathbf{y}|\mathbf{x}, \psi_i, \varphi_i)$ and that $p(\cdot|\mathbf{x}, \cdot, \cdot)$ is a valid probability distribution.

**Monotonicity.** Thus, we conclude:

$$
-\log p(\mathbf{x}|\psi_{i+1}, \varphi_{i+1}) + \log p(\mathbf{x}|\psi_i, \varphi_i) \leq -\sum_{\mathbf{y}} r(\mathbf{y}|\mathbf{x}, \varphi_i) \log p(\mathbf{x}, \mathbf{y}|\psi_{i+1}, \varphi_{i+1})
$$

$$
+ \sum_{\mathbf{y}} r(\mathbf{y}|\mathbf{x}, \varphi_i) \log p(\mathbf{x}, \mathbf{y}|\psi_i, \varphi_i)
$$

$$
\leq 0,
$$

where the final inequality holds because $(\psi_{i+1}, \varphi_{i+1})$ is chosen to minimize the expected negative log-joint likelihood under $r(\mathbf{y}|\mathbf{x}, \varphi_i)$.

Therefore,

$$
\mathcal{L}(\mathbf{x}; \psi_{i+1}, \varphi_{i+1}) \leq \mathcal{L}(\mathbf{x}; \psi_i, \varphi_i),
$$

which proves monotonic improvement of the ANELBO objective. $\qquad\square$

### A.3.4 Improved Likelihood Modeling During Inference

**Example.** To illustrate how anchoring improves likelihood modeling during inference, we consider a DAG model with $L = 3$ nodes: $\mathbf{x} = (\mathbf{x}^1, \mathbf{x}^2, \mathbf{x}^3)$. Each token takes values from a discrete vocabulary $\mathcal{V} = \{0, 1, m\}$ of size $K = 3$, where $m$ denotes the 'mask' token. We discretize time into $T = 3$ steps, with $t \in \{0, \frac{1}{3}, \frac{2}{3}, 1\}$. Each token $\mathbf{x}^l$ is represented as a one-hot vector, i.e., a corner point of the 3-dimensional probability simplex.

The full state space is $\mathcal{S} = \mathcal{V}^L$ and contains $3^3 = 27$ possible states. Let $X$ be a random variable taking values in $\mathcal{S}$ (such as $X = x = (1, 0, m) \in \mathcal{S}$), which is represented by a mixture distribution:

$$
\mathbf{x} = (\mathbf{x}^1, \mathbf{x}^2, \mathbf{x}^3) = \begin{bmatrix} 0 & 1 & 0 \\ 1 & 0 & 0 \\ 0 & 0 & 1 \end{bmatrix}
$$

We define a DAG structure where $\mathbf{x}^2$ is the parent node, and $\mathbf{x}^1$ and $\mathbf{x}^3$ are its children:

$$
q(\mathbf{x}) = q(\mathbf{x}^2) \, q(\mathbf{x}^1|\mathbf{x}^2) \, q(\mathbf{x}^3|\mathbf{x}^2)
$$

Suppose the data distribution $q$ is supported on two states:

$$x_1 = (1, 0, 0) \quad \text{with probability } 0.9, \quad \text{and} \quad x_2 = (0, 1, 1) \quad \text{with probability } 0.1.$$

This structure can be interpreted as a logical circuit: if $x^2 = 1$, it activates the bulb on the right ($x^3 = 1$); otherwise, if $x^2 = 0$, it activates the bulb on the left ($x^1 = 1$). In this circuit, $x^2$ determines the entire configuration of the sequence and thus acts as an important node.

It is easy to verify that the token $x^2$ is the most informative variable for modeling the joint likelihood. When $x^2 = 0$, which occurs with high probability in $q$, the entire sequence is determined. Thus, the *anchor* token is $x^2 = 0$ and it is represented as a column vector: $\mathbf{x}^2 = [1, 0, 0]^\top$.

This example is designed to illustrate how identifying and conditioning on such informative tokens (anchors) can improve the estimation of masked token likelihoods. Anchoring thus provides, as we discuss next, a principled mechanism for improving inference quality in discrete diffusion models.

**Forward Process.** In our discrete diffusion framework, the forward process gradually corrupts an input sequence $\mathbf{x} = (\mathbf{x}^1, \mathbf{x}^2, \mathbf{x}^3)$ by replacing tokens with a special *mask* token $\mathbf{m}$, according to a fixed noise schedule. We choose the forward noise schedule to be:

$$\alpha_{t(i)} = 1 - t(i) = 1 - \frac{i}{3}$$

This defines the amount of information retained from the original input $\mathbf{x}$ at time step $t$. The forward transition at each step is defined such that the token $\mathbf{x}^l$ is preserved with probability $\alpha_{t(i)}$ and replaced with the mask token $\mathbf{m}$ with probability $(1 - \alpha_{t(i)})$. For brevity, we drop $i$ from the noise schedule and denote by $\alpha_t$. The conditional distribution of the forward process is given by:

$$q(\mathbf{z}_t|\mathbf{x}) = \prod_{l=1}^{3} q(\mathbf{z}_t^l|\mathbf{x}), \quad q(\mathbf{z}_t^l|\mathbf{x}) = \text{Cat}\Big(\mathbf{z}_t^l; \alpha_t \mathbf{x}^l + (1 - \alpha_t)\mathbf{m}\Big), \quad l \in \{1, 2, 3\} \qquad (13)$$

Here, $\mathbf{z}_t^l$ denotes the corrupted version of token $\mathbf{x}^l$ at time $t$, and $\text{Cat}(\cdot)$ denotes the categorical distribution over the vocabulary $\mathcal{V} = \{0, 1, m\}$. The vector $\mathbf{x}^l \in \mathbb{R}^3$ is a one-hot column vector corresponding to the original token $x^l$, and $\mathbf{m} = [0, 0, 1]^\top$ is the one-hot vector for the mask token.

To make this concrete, consider the example input $x_1 = (1, 0, 0)$ from our earlier setup. This corresponds to the one-hot matrix:

$$\mathbf{x}_1 = \begin{bmatrix} 0 & 1 & 1 \\ 1 & 0 & 0 \\ 0 & 0 & 0 \end{bmatrix}.$$

At time $t = \frac{1}{3}$ (i.e., $i = 1$), we have $\alpha_t = 1 - \frac{1}{3} = \frac{2}{3}$. Substituting into the forward conditional gives:

$$q(\mathbf{z}_{1/3}|\mathbf{x}_1) = \begin{bmatrix} 0 & \frac{2}{3} & \frac{2}{3} \\ \frac{2}{3} & 0 & 0 \\ \frac{1}{3} & \frac{1}{3} & \frac{1}{3} \end{bmatrix}.$$

Each column of this matrix represents the categorical distribution over $\mathcal{V} = \{0, 1, m\}$ for the corresponding position in the sequence. For example:

- The first column corresponds to $\mathbf{z}_{1/3}^1 \sim \text{Cat}(0, \frac{2}{3}, \frac{1}{3})$
- The second column corresponds to $\mathbf{z}_{1/3}^2 \sim \text{Cat}(\frac{2}{3}, 0, \frac{1}{3})$
- The third column corresponds to $\mathbf{z}_{1/3}^3 \sim \text{Cat}(0, 0, 1)$

Similarly, we compute the following conditionals:

$$q(\mathbf{z}_{2/3}|\mathbf{x}_1) = \begin{bmatrix} 0 & \frac{1}{3} & \frac{1}{3} \\ \frac{1}{3} & 0 & 0 \\ \frac{2}{3} & \frac{2}{3} & \frac{2}{3} \end{bmatrix}, \quad q(\mathbf{z}_1|\mathbf{x}_1) = \begin{bmatrix} 0 & 0 & 0 \\ 0 & 0 & 0 \\ 1 & 1 & 1 \end{bmatrix}.$$

This formulation captures the probabilistic corruption of each token independently according to the noise schedule, blending its original identity with increasing uncertainty (i.e., masking) over time. It provides a concrete foundation for analyzing how anchoring improves denoising, especially when key tokens (like $\mathbf{x}^2$ in our setup) are retained or inferred with higher confidence.

To illustrate the benefit of anchoring during inference, we analyze a single reverse step in the diffusion process: transitioning from $\mathbf{z}_{2/3}$ to $\mathbf{z}_{1/3}$. Suppose we observe $\mathbf{z}_{2/3} = (\mathbf{m}, \mathbf{m}, \mathbf{m}) \sim q(\cdot|\mathbf{x}_1)$, i.e., the sequence is fully masked at this step. Ideally, we would prefer to unmask $\mathbf{z}_{2/3}^2 = [0, 0, 1]^\top$ to $\mathbf{z}_{1/3}^2 = [1, 0, 0]^\top$, since this corresponds to the parent node $\mathbf{x}^2 = \mathbf{0} = [1, 0, 0]^\top$, which is the most informative token in the sequence. Decoding this parent token early makes it significantly easier to infer the full sequence, and conditioning on it reduces the sample complexity of estimating the model's conditional probability tables from exponential to polynomial, as discussed in §A.3.2.

We now compute the likelihood of correctly recovering the important token under both the standard reverse process and our anchored reverse process. Finally, we show that anchored reverse process yields a higher probability of decoding the important token.

**Standard Reverse Process.** As described in §2, the reverse process is parameterized as:

$$
\begin{aligned}
p_\theta(\mathbf{z}_{1/3}^l|\mathbf{z}_{2/3}) &= q(\mathbf{z}_{1/3}^l|\mathbf{z}_{2/3}^l, \mathbf{x}_\theta^l(\mathbf{z}_{2/3})) \\
&= \begin{cases} \mathrm{Cat}(\mathbf{z}_{1/3}^l; \mathbf{z}_{2/3}^l), & \mathbf{z}_{2/3}^l \neq \mathbf{m} \\ \mathrm{Cat}\left(\mathbf{z}_{1/3}^l; \frac{1}{2}\mathbf{x}_\theta^l(\mathbf{z}_{2/3}) + \frac{1}{2}\mathbf{m}\right), & \mathbf{z}_{2/3}^l = \mathbf{m}. \end{cases}
\end{aligned} \tag{14}
$$

In this case, all tokens are masked at $\mathbf{z}_{2/3}$. To proceed, we estimate $\mathbf{x}_\theta^l(\mathbf{z}_{2/3})$ using samples from the data distribution $q$. Recall that $q$ is supported on:

$$x_1 = (1, 0, 0) \text{ with probability } 0.9, \quad x_2 = (0, 1, 1) \text{ with probability } 0.1.$$

Thus, we compute the predicted token mixture as:

$$\mathbf{x}_\theta^1(\mathbf{z}_{2/3}) = [0.1, 0.9, 0]^\top, \quad \mathbf{x}_\theta^2(\mathbf{z}_{2/3}) = [0.9, 0.1, 0]^\top, \quad \mathbf{x}_\theta^3(\mathbf{z}_{2/3}) = [0.9, 0.1, 0]^\top.$$

This concurs with the zero-masking parameterization discussed in §2. Substituting into the reverse conditional (14), we obtain:

$$
p_\theta(\mathbf{z}_{1/3}|\mathbf{z}_{2/3}) = \begin{bmatrix} 0.05 & 0.45 & 0.45 \\ 0.45 & 0.05 & 0.05 \\ 0.50 & 0.50 & 0.50 \end{bmatrix}.
$$

Each column represents the categorical distribution over the vocabulary $\{0, 1, m\}$ for $\mathbf{z}_{1/3}^1, \mathbf{z}_{1/3}^2, \mathbf{z}_{1/3}^3$ respectively. We now compute the likelihood of decoding the target partial sequence $(\mathbf{m}, \mathbf{0}, \mathbf{m})$, i.e., correctly unmasking the important token:

$$p_\theta(\mathbf{z}_{1/3} = (\mathbf{m}, \mathbf{0}, \mathbf{m})|\mathbf{z}_{2/3} = (\mathbf{m}, \mathbf{m}, \mathbf{m})) = (0.5) \cdot (0.45) \cdot (0.5) = 0.1125.$$

This probability reflects the chance of correctly decoding only the important token using the standard reverse process. In the following section, we contrast this with the likelihood achieved under the anchored reverse process used in ADLM.

**ADLM Reverse Process.** As discussed in §3, our anchored reverse process is parameterized as:

$$
\begin{aligned}
p_{[\psi,\varphi]}(\mathbf{z}_{1/3}^l|\mathbf{z}_{2/3}) &= q(\mathbf{z}_{1/3}^l|\mathbf{z}_{2/3}^l, \mathbf{x}_\psi^l(\mathbf{y}_\varphi(\mathbf{z}_{2/3}))) \\
&= \begin{cases} \mathrm{Cat}(\mathbf{z}_{1/3}^l; \mathbf{z}_{2/3}^l), & \text{if } \mathbf{z}_{2/3}^l \neq \mathbf{m} \\ \mathrm{Cat}\left(\mathbf{z}_{1/3}^l; \frac{1}{2}\mathbf{x}_\psi^l(\mathbf{y}_\varphi(\mathbf{z}_{2/3})) + \frac{1}{2}\mathbf{m}\right), & \text{if } \mathbf{z}_{2/3}^l = \mathbf{m} \end{cases}
\end{aligned} \tag{15}
$$

Given an input sequence $\mathbf{x}$, let the operator $\mathcal{A}(\cdot)$ construct the anchored sequence $\mathbf{y} = \mathcal{A}(\mathbf{x}) = (\mathbf{x}^1, \mathbf{0}, \mathbf{x}^3)$. This operator overwrites the second position with the important anchor token $[1, 0, 0]^\top$, while copying the first and third tokens from $\mathbf{x}$. Since the anchor loss (see Eq. (9)) is applied only to important tokens, the anchor network is trained to predict $\mathbf{y}_\varphi^2(\mathbf{z}_{2/3}) = [1, 0, 0]^\top$. In contrast, the first and third tokens are jointly parameterized and optimized with the denoiser network to maximize the overall sequence likelihood. The denoiser network $\mathbf{x}_\psi^l(\cdot)$ behaves as follows:

- If $\mathbf{y}_\varphi^l(\mathbf{z}_{2/3})$ is unmasked, then $\mathbf{x}_\psi^l(\mathbf{y}_\varphi(\mathbf{z}_{2/3}))$ simply copies it to the output due to the carry-over unmasking parameterization (§2).
- Otherwise, it defaults to a standard MLE estimate as in the vanilla reverse process.

Therefore, the predicted token mixture becomes:

$$\mathbf{x}_\psi^1(\mathbf{y}_\varphi(\mathbf{z}_{2/3})) = [0.1, 0.9, 0]^\top, \quad \mathbf{x}_\psi^2(\mathbf{y}_\varphi(\mathbf{z}_{2/3})) = [1, 0, 0]^\top, \quad \mathbf{x}_\psi^3(\mathbf{y}_\varphi(\mathbf{z}_{2/3})) = [0.9, 0.1, 0]^\top.$$

Substituting these predictions into Eq. (15), the conditional distribution under the ADLM reverse process becomes:

$$p_{[\psi,\varphi]}(\mathbf{z}_{1/3}|\mathbf{z}_{2/3}) = \begin{bmatrix} 0.05 & 0.50 & 0.45 \\ 0.45 & 0.0 & 0.05 \\ 0.50 & 0.50 & 0.50 \end{bmatrix}.$$

We now compute the likelihood of decoding the desired partial sequence $(\mathbf{m}, \mathbf{0}, \mathbf{m})$:

$$p_{[\psi,\varphi]}(\mathbf{z}_{1/3} = (\mathbf{m}, \mathbf{0}, \mathbf{m})|\mathbf{z}_{2/3} = (\mathbf{m}, \mathbf{m}, \mathbf{m})) = (0.5) \cdot (0.5) \cdot (0.5) = 0.125$$

This is higher than the standard reverse process likelihood computed earlier:

$$p_\theta(\mathbf{z}_{1/3} = (\mathbf{m}, \mathbf{0}, \mathbf{m})|\mathbf{z}_{2/3} = (\mathbf{m}, \mathbf{m}, \mathbf{m})) = 0.1125$$

This example illustrates how anchoring improves the recovery of important tokens during inference. Similarly, the likelihood of $\mathbf{z}_{1/3} = (\mathbf{1}, \mathbf{0}, \mathbf{m})$ given $\mathbf{z}_{2/3} = (\mathbf{m}, \mathbf{m}, \mathbf{m})$ is higher for ADLM compared standard DLM. By prioritizing important tokens like $\mathbf{x}^2$, ADLM improves masked likelihood modeling and reduces sample complexity of the denoiser.

**Practical Considerations.** While this example assumes sampling from the anchor distribution followed by denoising, such a two-step pipeline is not end-to-end differentiable due to the intermediate sampling operation. In practice, we implement anchoring by projecting the anchor logits through a linear layer into the embedding space of the denoiser. This allows gradients to flow from the denoiser output back to the anchor network though the linear projection, enabling efficient joint training of both the modules; refer to implementation details in §C.1.3.

**Summary.** Our anchored graphical model analysis demonstrates the dual advantage of anchoring in language models: (1) reduced sample complexity during training (§A.3.2) and (2) improved likelihood modeling during inference (§A.3.4). While this analysis was performed on a small DAG ($L = 3$) and coarse time discretizations ($T = 3$), the insights generalize to broader classes of DAGs and finer time discretizations. We believe this theoretical understanding complements our strong empirical results across generative modeling (§5.1) and complex reasoning benchmarks (§5.2) in the main draft (§5) and also in the Appendix C. We hope it offers a compelling justification for the use of anchoring as a general framework for language modeling.

# B  Additional Background and Related Works

In this section, we provide extended background and related work relevant to our proposed approach. We focus on two model families: diffusion language models (§B.1) and autoregressive models (§B.2), covering their recent developments.

## B.1  Diffusion Language Models

Diffusion models are built on two stochastic processes: a forward (noising) process that gradually corrupts a clean input $\mathbf{x}$ into a noisy latent representation $\mathbf{z}_t$ for $t \in [0, 1]$, and a reverse (denoising) process that reconstructs $\mathbf{x}$ from $\mathbf{z}_1$. The effectiveness of diffusion models is primarily attributed to two factors: *iterative refinement* through multiple steps, and a *simple regression-based training objective* (Sohl-Dickstein et al., 2015; Ho et al., 2020).

For continuous domains (e.g., image generation), the forward process is typically modeled as an Ornstein–Uhlenbeck (OU) process that adds Gaussian noise with increasing variance (Sohl-Dickstein et al., 2015; Ho et al., 2020). In the discrete setting, the forward process is defined by either: (a) *uniform noising*, where each token is replaced with another random token from the vocabulary (Austin

et al., 2021; Lou et al., 2024; Liu et al., 2025; Varma et al., 2025), or (b) *random masking*, where each token is independently replaced by a special mask token $\mathbf{m}$ (Austin et al., 2021; Lou et al., 2024; Sahoo et al., 2024; Arriola et al., 2025; Wang et al., 2025a; Ou et al., 2025; Nie et al., 2025a,b).

Recent studies (Austin et al., 2021; Sahoo et al., 2024; Lou et al., 2024; Shi et al., 2024; Ou et al., 2025) demonstrate that random masking provides improved training stability and sample quality compared to uniform noising. In these *masked* DLMs, the reverse process is trained to progressively reconstruct the sequence from a fully masked input $\mathbf{z}_1 = (\mathbf{m}, \mathbf{m}, \ldots, \mathbf{m})$ by unmasking tokens step-by-step. Training is done via a negative evidence lower bound objective (Austin et al., 2021), which admits a score-based interpretation (Lou et al., 2024) and supports time-independent parameterization (Ou et al., 2025), enabling simplified training and efficient scaling (Nie et al., 2025a,b).

The time-independent formulation is based on the insight that in masked DLMs, the number of masked tokens implicitly encodes the timestep. Therefore, the denoising network does not require an explicit time embedding as input. Our proposed ADLM follows this time-independent parameterization, simplifying the model training and improving scalability without sacrificing performance.

## B.2 Auto-Regressive Models

**Explicit CoT Fine-Tuning (GPT-2 baseline).** In this baseline, a GPT-2 model is fine-tuned to generate chain-of-thought (CoT) reasoning traces followed by the final answer (Wei et al., 2022). It serves as a strong supervised benchmark for comparison, and we include several variants in our experimental evaluation. We demonstrate how anchoring can be integrated on top of standard CoT to complement reasoning.

**COCONUT (Chain-of-Continuous-Thought).** COCONUT (Hao et al., 2024) extends CoT by replacing discrete reasoning tokens with continuous latent representations. A GPT-2 model is fine-tuned using a multi-stage procedure that gradually introduces continuous latent "thoughts" between the question and answer. When integrated with our anchoring mechanism, this approach also yields strong performance on symbolic reasoning benchmarks like ProsQA (see §C.2).

**CODI (Continuous Chain-of-Thought via Self-Distillation).** CODI (Shen et al., 2025) distills reasoning steps into a continuous latent space, achieving a final accuracy of 43.7% on GSM8K (Cobbe et al., 2021). This approach outperforms the previous best GPT-2 finetuned model by roughly 9.6% (from 34.1% to 43.7%) as shown in Table 5. CODI is orthogonal to our method: while it compresses reasoning via distillation, our model explicitly guides reasoning via anchor tokens without relying on distillation. Exploring combinations of both ideas is an interesting direction for future work.

Since CoT and COCONUT are closely related to our work, we provide a more detailed discussion of these baselines and their variants in §C.2.

## B.3 Token Unmasking Strategies in Diffusion and AR Language Models

A large body of work has explored strategies for adaptive unmasking tokens during the generative process in diffusion and AR language models. These strategies include:

- **Greedy decoding**, where the most confident (i.e., high-probability) tokens are unmasked first (Nie et al., 2025a,b).
- **Locked-in sampling**, where once a token is unmasked, it is fixed for the remainder of the generation (Sahoo et al., 2024).
- **Remasking sampling**, which allows previously unmasked tokens to be re-masked and resampled in future steps (Wang et al., 2025a).
- **Top-$p$ (nucleus) sampling**, where the sampling distribution is restricted to the smallest subset of tokens whose cumulative probability mass exceeds a threshold $p$ (Radford et al., 2019; Wang & Cho, 2019; Varma et al., 2025).
- **Top-$k$ sampling**, where only the top $k$ tokens with the highest logits are retained for sampling (Fan et al., 2018).
- **Beam search**, where a set of candidate sequences (beam) is expanded and pruned at each step until a termination condition is met.

While these approaches focus on selecting tokens based on confidence or likelihood, our work introduces a new approach: *token importance*. Rather than decoding the most likely tokens, which

often correspond to frequent but semantically shallow tokens such as articles or conjunctions, our anchoring mechanism prioritizes decoding *informative tokens*—typically content-bearing nouns, verbs, or entities that anchor a sentence.

By identifying (using a jointly trained anchor network) and decoding these anchor tokens first, our model improves contextual understanding and enables the denoiser to more accurately recover the remaining tokens. We demonstrate that this strategy is effective across two representative sampling strategies in DLMs: the locked-in sampler (Sahoo et al., 2024), and the remasking sampler (Wang et al., 2025a), which internally integrates top-$p$ (nucleus) sampling.

This shift from decoding based on likelihood to decoding based on semantic utility offers a new perspective on generative planning and opens the door to further improvements in interpretability, reasoning, and controllable generation. This also opens up an interesting research direction—look ahead planning and reasoning in AR models—as we demonstrate in §C.2.

## C Additional Experiments

This section provides additional experimental details on our proposed anchoring mechanism, applied to both masked diffusion language models in §C.1 and autoregressive models in §C.2. We present benchmark datasets, training procedure, ablation studies, and additional quantitative and qualitative results to support the findings discussed in the main paper.

**Broader Impact.** This work introduces an anchoring framework that improves likelihood modeling and generation quality in DLMs, while also enhancing complex reasoning in AR models. On the positive side, ADLM has the potential to increase the accuracy, interpretability, and efficiency of language models applied to critical domains such as education, healthcare, and scientific research. Its ability to prioritize semantically important tokens may contribute to the development of more transparent and explainable AI systems.

However, these same capabilities also carry risks. Enhanced reasoning and generation fidelity may increase the potential for misuse, such as generating persuasive misinformation, reinforcing biases, or enabling manipulation. As with other generative models, there is a possibility of producing deceptive or harmful content if deployed irresponsibly. We emphasize the importance of safeguards and responsible deployment to mitigate these risks.

**Reproducibility.** To support reproducibility, we provide complete pseudo-code and all hyperparameter settings used in our experiments. Our training and evaluation protocols are aligned with prior work to ensure fair and transparent comparisons.

**Safeguards.** To mitigate risks of misuse, we recommend applying standard safeguards such as dataset filtering, usage auditing, and model alignment techniques, including Reinforcement Learning from Human Feedback (RLHF). We also advocate for releasing models and code under responsible use guidelines with access controls and documentation to promote ethical deployment.

### C.1 Diffusion Language Models

This subsection provides extended experimental details for diffusion language models. We describe the compared baselines in §C.1.1, outline the training and evaluation benchmarks in §C.1.2, and present implementation details along with additional results in §C.1.3. Finally, we include qualitative examples and analysis of samples generated by ADLM in §C.1.4.

### C.1.1 Compared Baselines

We compare ADLM against a broad range of autoregressive, diffusion, and hybrid (autregressive+diffusion) language models. Each baseline is evaluated under the same training and evaluation protocol as ADLM, using identical tokenizers, data splits, and sampling steps. Hyperparameters are adopted from official implementations or tuned for fairness when not explicitly provided. Below, we summarize each method and provide links to source code where available:

- **Autoregressive Transformer (AR):** A standard GPT-style transformer trained using next-token prediction. We use the architecture from the MDLM repository.[2]

---

[2] https://github.com/kuleshov-group/mdlm/blob/master/models/autoregressive.py

- **SEDD** (Lou et al., 2024): A discrete diffusion language model that denoises using a neural network trained to approximate score entropy. Source: `https://github.com/louaaron/Score-Entropy-Discrete-Diffusion`
- **MDLM** (Sahoo et al., 2024): A masked diffusion language model trained with the NELBO objective (4). MDLM uses a locked-in sampler where tokens, once unmasked, remain fixed. Source: `https://github.com/kuleshov-group/mdlm`
- **MD4** (Shi et al., 2024): A masked diffusion model for discrete data trained with the NELBO objective (4). Source: `https://github.com/google-deepmind/md4`
- **BD3LM** (Arriola et al., 2025): A hybrid model that combines autoregressive generation with block-wise diffusion-based refinement. This enables parallel sampling in each block and sequential across blocks. Source: `https://github.com/kuleshov-group/bd3lms`
- **ReMDM** (Wang et al., 2025a): Extends MDLM by incorporating a re-masking sampler, which allows the model to re-mask and re-predict tokens during inference. This re-masking sampler improves generated text quality. Source: `https://github.com/kuleshov-group/remdm`
- **GIDD** (von Rütte et al., 2025): GIDD introduces a general interpolation between masking and uniform noising in discrete diffusion models. As a concurrent work to ReMDM, it addresses a key limitation of MDLM (i.e., the inability to revise tokens once unmasked) by allowing previously unmasked tokens to be updated during inference. This flexibility enables the model to iteratively correct its own mistakes and refine its outputs more effectively. Source: `https://github.com/dvruette/gidd/`
- **DFM (Discrete Flow Matching)** (Gat et al., 2024): DFM is a flow-based approach for discrete data that replaces the diffusion loss with a flow matching objective (Gat et al., 2024). At the time of writing, the official implementation was unavailable. We instead reference a simplified public version capturing the core ideas[3]. For evaluation, we adopt the DFM sampler built on top of the MDLM base model, available in the ReMDM repository:[4].
- **Forward-Backward (FB)** (Campbell et al., 2022): FB is a corrective sampler that combines forward and backward transitions to improve reconstruction accuracy (Campbell et al., 2022). It is applied as a post-hoc sampling method on top of pretrained models such as MDLM. We use the publicly available implementation from the ReMDM repository:[5].

All models are evaluated using the same number of neural function evaluations (NFEs), training tokens, and sampling steps where applicable. This ensures a consistent and fair comparison across all baselines, allowing us to isolate the effect of anchoring on model performance.

### C.1.2 Training and Evaluation Benchmarks

We evaluate diffusion language models on two fronts: (1) generative modeling quality and (2) zero-shot generalization to downstream tasks.

For generative modeling, we use two widely adopted masked language modeling benchmarks: **One Billion Words (LM1B)** (Chelba et al., 2013) and **OpenWebText (OWT)** (Gokaslan & Cohen, 2019). These datasets provide large-scale and diverse natural language corpora for evaluating the ability of models to understand and generate natural language.

For downstream evaluation, we assess zero-shot likelihoods on seven standard benchmarks spanning commonsense reasoning, scientific language, and formal language domains. These include Lambada, PTB, WikiText, LM1B, AG News, PubMed, and ArXiv. Below, we briefly describe these benchmarks.

**One Billion Words (LM1B).** We use the LM1B dataset (Chelba et al., 2013)[6] which consists of news crawl data collected by Chelba et al. (2013). The dataset is released under the Apache 2.0 license. Following prior work (Sahoo et al., 2024; Wang et al., 2025a), we train ADLM for 1M steps using a batch size of 512 and a context length of 128, which corresponds to approximately 33B tokens. A larger variant trained for 2M steps sees approximately 65B tokens. The autoregressive baseline is trained for 0.5M and 1M steps respectively to match the total number of tokens processed. For evaluation, we use the standard LM1B test split.

---

[3]`https://github.com/facebookresearch/flow_matching`
[4]`https://github.com/kuleshov-group/remdm/blob/main/scripts/dfm.sh`
[5]`https://github.com/kuleshov-group/remdm/blob/main/scripts/fb.sh`
[6]`https://code.google.com/archive/p/1-billion-word-language-modeling-benchmark/`

**OpenWebText (OWT).** We use the OpenWebText dataset (Gokaslan & Cohen, 2019)[7], which is a public reproduction of the WebText dataset originally used in GPT-2 (Radford et al., 2019). It consists of web content extracted from high-quality Reddit URLs. The dataset is licensed under Creative Commons CC0 license ("no rights reserved").

**Zero-Shot Evaluation Benchmarks.** To assess generalization, we perform zero-shot evaluation on seven diverse benchmarks covering language understanding, scientific articles, and long-context reasoning. We measure the perplexity on the validation sets of the following benchmarks:

- **Lambada** (Paperno et al., 2016): This benchmark evaluates the ability of language models to predict a target word based on a broad context. Each example consists of a short narrative (*context*) followed by a target sentence with its final word omitted. Unlike typical language modeling tasks, solving Lambada requires understanding of longer-range dependencies beyond just the last sentence.
- **Penn Treebank (PTB)** (Marcus et al., 1993): A classical benchmark for language modeling, which is used to evaluate syntactic fluency and local coherence in generated outputs.
- **WikiText** (Merity et al., 2017): A long-form language modeling benchmark based on Wikipedia articles. This evaluation emphasizes factual correctness and world knowledge.
- **AG News** (Zhang et al., 2015): A text classification dataset with four categories, which is designed to predict the topic label given a news headline or excerpt.
- **PubMed** (Cohan et al., 2018): A benchmark based on biomedical research articles, curated for studying text summarization. Each example consists of a document body and a corresponding summary, typically derived from the abstract or conclusion section.
- **ArXiv** (Cohan et al., 2018): A benchmark similar to PubMed, but based on research articles from the ArXiv repository across diverse scientific domains. Summaries are typically derived from the abstract or conclusion sections of the papers.

**Licenses and Usage.** All datasets used in this work are publicly available and licensed for research use. OWT is distributed under the Creative Commons CC0 license, and LM1B is under the Apache 2.0 license. All baseline models (e.g., SEDD, MDLM, BD3LM, ReMDM, GPT-2) and implementations are sourced from repositories released under MIT or Apache 2.0 licenses.

All reported perplexity values for diffusion language models in this paper are *upper bounds*, rather than exact likelihoods. This is because diffusion models do not compute normalized likelihoods in closed form due to the intractability of the reverse process. Instead, we follow prior work (Lou et al., 2024; Sahoo et al., 2024; Arriola et al., 2025; Wang et al., 2025a) and evaluate diffusion models by estimating a negative log-likelihood upper bound via importance sampling or ELBO-based objectives. In contrast, perplexities for autoregressive models are computed exactly via standard token-wise cross-entropy. As such, direct PPL comparisons should be interpreted with this distinction in mind.

### C.1.3 Implementation Details & Additional Results

**Architecture Details.** We apply a learnable linear projection within the denoiser network $\mathbf{x}_\psi(\cdot)$ that maps the anchor logits $\mathbf{y}_\varphi(\mathbf{z}_t)$ directly into the embedding space of the $\psi$-transformer. This modular design avoids explicit decoding and re-embedding of anchor predictions, enabling fully end-to-end differentiability between the anchor and denoising networks.

Although the ANELBO loss (7) is defined over all tokens $\{\mathbf{x}^l\}_{l=1}^L$ in the sequence $\mathbf{x}$, only the *masked tokens* contribute to the denoiser loss via $\mathbf{x}_\psi(\mathbf{y}_\varphi(\mathbf{z}_t))$, and only the *important tokens* contribute to the anchor loss via $\mathbf{y}_\varphi(\mathbf{z}_t)$. In practice, this is implemented using carry-over unmasking (Sahoo et al., 2024) or by multiplying indicator functions in (7). Specifically, we use $\mathbf{1}_{\{\mathbf{z}_t^l = \mathbf{m}\}}$ to mask the denoising loss, and $\mathbf{1}_{\{\mathbf{y}^l = (\mathcal{A}(\mathbf{x}))^l\}}$ to mask the anchor loss.

Both networks follow the DiT-Base architecture (Peebles & Xie, 2023), which is consistent with SEDD (Lou et al., 2024) and MDLM (Sahoo et al., 2024). The anchor transformer network uses 12 DiT blocks. The denoising network uses 6 DiT blocks. Each block has hidden dimension = 768 and 12 attention heads. The input sequence length is 1024 for OWT and 128 for LM1B. We use the same AdamW optimizer for both anchor and denoising transformers with learning rate = 3e-4 and and no weight decay.

---

[7] http://Skylion007.github.io/OpenWebTextCorpus

**Algorithm 1:** Anchored Diffusion Language Model (ADLM)

---

**Input:** Anchor network $\mathbf{y}_\varphi(\cdot)$, denoising network $\mathbf{x}_\psi(\cdot)$, number of steps $T$, noise schedule $\alpha_t$, remasking schedule $\sigma_t$

**Output:** Generated sequence $\mathbf{z}_0$

1 Initialize $\mathbf{z}_T \leftarrow (\mathbf{m}, \mathbf{m}, \dots, \mathbf{m})$                      ▷ Fully masked sequence

2 **for** $i = T$ **to** $1$ **do**

3      $t = i/T, \quad s = (i-1)/T$

4      Compute noise schedule: $\alpha_t, \alpha_s$

5      Compute remasking schedule: $\sigma_t \in [0, \sigma_t^{\max}]$      ▷ Follows Eq. (9) (Wang et al., 2025a)

6      Compute anchor transition using noisy sequence:             ▷ Eq. (5) in §3

$$r(\mathbf{y}_s^l | \mathbf{z}_t^l, \mathbf{y}_\varphi(\mathbf{z}_t)) = \begin{cases} \mathrm{Cat}(\mathbf{z}_s^l; (1-\sigma_t)\mathbf{y}^l + \sigma_t\mathbf{m}), & \mathbf{z}_t^l \neq \mathbf{m} \\ \mathrm{Cat}(\mathbf{z}_s^l; \frac{\alpha_s - (1-\sigma_t)\alpha_t}{1-\alpha_t}\mathbf{y}_\varphi^l(\mathbf{z}_t) + \frac{1-\alpha_s-\alpha_t\sigma_t}{1-\alpha_t}\mathbf{m}), & \mathbf{z}_t^l = \mathbf{m} \end{cases}$$

7      Compute inference posterior using anchored prediction $\mathbf{y}_\varphi(\mathbf{z}_t)$:      ▷ Eq. (6) in §3

$$q(\mathbf{z}_s^l | \mathbf{z}_t^l, \mathbf{x}_\psi^l(\mathbf{y}_\varphi(\mathbf{z}_t))) = \begin{cases} \mathrm{Cat}(\mathbf{z}_s^l; (1-\sigma_t)\mathbf{x}^l + \sigma_t\mathbf{m}), & \mathbf{z}_t^l \neq \mathbf{m} \\ \mathrm{Cat}(\mathbf{z}_s^l; \frac{\alpha_s - (1-\sigma_t)\alpha_t}{1-\alpha_t}\mathbf{x}_\psi^l(\mathbf{y}_\varphi(\mathbf{z}_t)) + \frac{1-\alpha_s-\alpha_t\sigma_t}{1-\alpha_t}\mathbf{m}), & \mathbf{z}_t^l = \mathbf{m} \end{cases}$$

8      Sample $\mathbf{z}_s^l \sim q(\mathbf{z}_s^l | \mathbf{z}_t^l, \mathbf{x}_\psi^l(\mathbf{y}_\varphi(\mathbf{z}_t)))$ for all $l \in \{1, \dots, L\}$

9      Update $\mathbf{z}_t \leftarrow \mathbf{z}_s$

10 **end**

11 **return** $\mathbf{z}_0$

---

Table 6: Evaluation metrics on OWT (78B tokens) for our ADLM algorithm across different (a) anchoring loss coefficients ($\gamma$) and (b) anchoring thresholds ($\tau$).

| (a) $\gamma$ | 3e-2 | 3e-3 | 3e-5 | | (b) $\tau$ | 1 | 5 | 10 | 20 |
|---|---|---|---|---|---|---|---|---|---|
| NLL ($\downarrow$) | 3.1084 | **3.1055** | 3.1663 | | NLL ($\downarrow$) | 3.1429 | **3.1055** | 3.1358 | 3.1770 |
| PPL ($\downarrow$) | 22.386 | **22.321** | 23.719 | | PPL ($\downarrow$) | 23.171 | **22.321** | 23.008 | 23.976 |
| BPD ($\downarrow$) | 4.4878 | **4.4803** | 4.5680 | | BPD ($\downarrow$) | 4.5342 | **4.4803** | 4.5241 | 4.5835 |

**Sampling Implementation.** We provide the pseudo-code for ADLM sampling in **Algorithm 1**, which employs the standard locked-in sampler (Sahoo et al., 2024) with a remasking schedule $\sigma_t$ (Wang et al., 2025a). The locked-in sampler is a special case obtained by setting $\sigma_t = 0$.

**Implementation Details for OWT.** As OWT dataset does not provide an official train/test split, we use the splits used in prior work (Sahoo et al., 2024) and train ADLM for 1M and 2M steps with a GPT-2 tokenizer, batch size of 512, sequence length of 1024, and a log-linear diffusion schedule. This results in approximately 262B (1M steps) and 524B (2M steps) tokens. The AR baseline is trained for half as many steps under the same configuration to ensure a comparable number of tokens seen. We use the last 100K documents (held out during training) for evaluation.

**Ablation Study for OWT.** In Table 1(b) (262B tokens) of the main paper, we show that simply using our two-stage architecture without the anchor loss already improves test perplexity from 23.17 to 21.79, validating our architectural choice. Adding the anchor loss further reduces perplexity to 20.62, highlighting the effectiveness of anchoring in likelihood modeling.

To better understand the role of anchoring, we conduct an ablation study on OWT using 78B tokens over 300K training iterations. We evaluate the impact of two hyperparameters: the anchoring loss coefficient ($\gamma$) and the anchor threshold ($\tau$), as discussed in §3. Table 6 reports results across different values. In Table 6(a), we observe that $\gamma = $ 3e-3 yields the best trade-off across negative log-likelihood (NLL), perplexity (PPL), and bits-per-dimension (BPD), outperforming both higher (3e-2) and lower (3e-5) values. Similarly, Table 6(b) shows that $\tau = 5$ achieves the best results, outperforming thresholds such as 1, 10, and 20. Based on this study, we use $\gamma = $ 3e-3 and $\tau = 5$ as our default configuration across all experiments. With this default configuration, the training loss and validation PPL per iteration is shown in Figure 3.

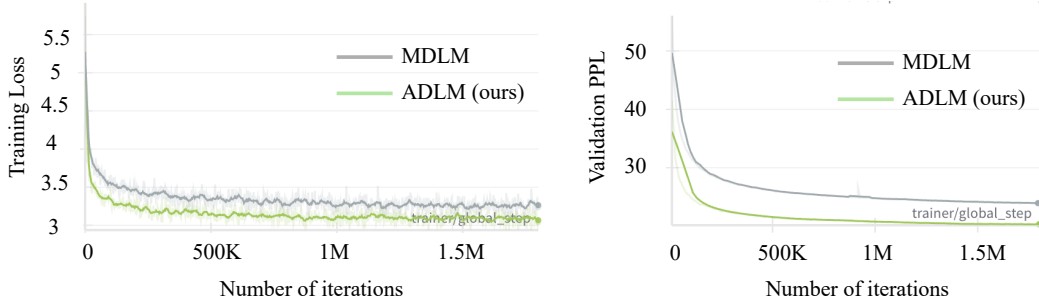

Figure 3: **Training loss and validation PPL versus number of iterations on OWT.** We train both MDLM (Sahoo et al., 2024) and our ADLM model for 2M iterations (524B tokens). As discussed in §4, anchoring improves the sample complexity during training, resulting in faster convergence and lower validation perplexity. While the anchor loss is part of the training objective, we only visualize the NELBO here for a direct comparison with MDLM.

**Implementation Details for LM1B.** The experimental setup follows prior works (Sahoo et al., 2024; Wang et al., 2025a). For LM1B, we follow the same setup as OWT, except a shorter sequence length of 128 tokens and the BERT-base-uncased tokenizer. Since LM1B includes an official test split, we use it for evaluation. We use the same anchoring configuration ($\gamma$ = 3e-3, $\tau$ = 5) as in OWT.

**Remasking Evaluation.** For the results reported in Table 3, we generate 5,000 samples using sampling steps ranging from 128 to 4096. We evaluate each set of samples using MAUVE score, GPT-2 perplexity, and entropy. All hyperparameters follow the settings recommended in the remasking sampler developed in ReMDM (Wang et al., 2025b). For clarity, we report the exact values used in our experiments:

- sampling steps: Number of sampling steps used per generation; values range over {128, 256, 512, 1024, 2048, 4096}.
- $p$: 0.9 Top-$p$ value used in nucleus sampling.
- $\eta$: 0.02 Parameter used in ReMDM strategies.
- $t_{\text{on}}$: 0.55 Activation time for remasking in the ReMDM loop.
- $t_{\text{off}}$: 0.05 Deactivation time for remasking in the ReMDM loop.
- $\alpha_{\text{on}}$: 0.9 Fixed masking schedule $\alpha(t_{\text{on}})$ used in the ReMDM loop.

**Discussion on Anchoring vs. Attention.** Anchoring is completely different from the traditional attention mechanism. In the example shown in Figure 1, attention layers operate only over the tokens available in $Z_t$. A key limitation arises when important tokens are masked—standard attention mechanisms are unable to access or reason about these missing tokens. In contrast, our anchoring mechanism explicitly predicts important tokens in its output, which can then be attended to by the downstream attention layers of the denoiser network. As such, our method is orthogonal to attention mechanisms. In fact, anchoring provides an efficient way to reduce the sample complexity of these layers and improve training as discussed in §A.3.

**Discussion on Size vs. Generative Perplexity.** In Table 7, we compare the generative perplexities of ADLM against a range of autoregressive, masked language, and diffusion-based models, all evaluated using GPT2-Large over 1024 unconditional generations. A key advantage of ADLM lies in its ability to produce high-quality generations with significantly fewer parameters and fewer sampling steps than prior diffusion language models.

Specifically, ADLM achieves a perplexity of 15.7 using 4096 sampling steps, outperforming Plaid (Gulrajani & Hashimoto, 2023), which reaches a perplexity of 19.7 despite using nearly *five times* more parameters (1.3B vs. 293M). Even at lower sampling budgets (e.g., $T$=2048), ADLM maintains strong performance (20.1), reducing the performance gap with GPT2-medium (12.4) and outperforming BERT-Large+Gibbs while using at least 40M fewer parameters. This indicates that anchoring enables more efficient denoising compared to prior diffusion language models.

Moreover, ADLM demonstrates a consistent reduction in perplexity as the number of sampling steps increases, validating the role of iterative refinement in enhancing sample quality. These results

Table 7: **Generative perplexities evaluated over 1024 unconditional generations using GPT2-Large (774M params) as the evaluation model.** ADLM significantly outperforms prior masked and diffusion language models under comparable sampling configurations. Notably, it surpasses Plaid (Gulrajani & Hashimoto, 2023) while using only ∼20% of the parameters (5x smaller size), and nearly matches the performance of GPT2-medium despite having ∼50M fewer parameters.

| Model Type | Evaluated Model | Params | Gen. PPL ($\downarrow$) |
|---|---|---|---|
| *Autoregressive* | GPT2-medium (Radford et al., 2019) | 345M | 12.4 |
| *Masked Language* | BERT-Large + Gibbs (Wang & Cho, 2019) ($T$=2048) | 334M | 487.0 |
| | BERT-Large + Gibbs (Wang & Cho, 2019) ($T$=65536) | 334M | 28.7 |
| *Diffusion* | Plaid (Gulrajani & Hashimoto, 2023) ($T$=4096) | 1.3B | 19.7 |
| | SEDD-medium (Lou et al., 2024) ($T$=2048) | 424M | 27.3 |
| | SEDD-medium (Lou et al., 2024) ($T$=1000) | 424M | 31.95 |
| | MD4-medium (Shi et al., 2024) ($T$=1000) | 416M | 27.51 |
| | SEDD-small (Lou et al., 2024) ($T$=1000) | 165M | 42.94 |
| | MD4-small (Shi et al., 2024) ($T$=1000) | 165M | 33.16 |
| | MDLM (Sahoo et al., 2024) (locked-in, $T$=1000) | 170M | 44.2 |
| | ReMDM (Wang et al., 2025a) (remasking, $T$=1024) | 170M | 28.6 |
| | GGM (Varma et al., 2025) ($T$=4096) | 387M | 19.5 |
| | ADLM (ours) (locked-in, $T$=1000) | 293M | 32.9 |
| | ADLM (ours) (remasking, $T$=1000) | 293M | 26.8 |
| | ADLM (ours) (remasking, $T$=1024) | 293M | 25.1 |
| | ADLM (ours) (remasking, $T$=2048) | 293M | 20.1 |
| | ADLM (ours) (remasking, $T$=4096) | 293M | **15.7** |

highlight the sample efficiency and scalability of our approach, making it a practical alternative to large autoregressive or parameter-heavy diffusion models for high-quality text generation.

**Discussion on MAUVE Evaluation with BERT Embeddings.** We recompute MAUVE scores using `bert-base-uncased` embeddings (maximum sequence length = 512, $T = 4096$) to verify the robustness of our evaluation. Compared to the best-performing baseline ReMDM (Table 3), our method achieves a 7% higher MAUVE score, confirming that ADLM generates more coherent and human-like text under both GPT-2 and BERT embedding evaluations.

Table 8: **Evaluation on OWT.** MAUVE scores computed using `bert-base-uncased` embeddings with maximum sequence length = 512 and $T = 4096$.

| Method | MAUVE ($\uparrow$) | Gen PPL ($\downarrow$) | Entropy ($\uparrow$) |
|---|---|---|---|
| ReMDM | 0.702 | 17.6 | 5.20 |
| **ADLM (ours)** | **0.754** | **15.9** | **5.19** |

### C.1.4 Samples Generated using ADLM

In this section, we present qualitative samples generated by our method ADLM and compare them with MDLM (Sahoo et al., 2024). We illustrate outputs under two sampling strategies: the *locked-in sampler* (Examples I and IV) and the *remasking sampler* (Examples II and III). As shown, increasing the number of sampling steps and enabling remasking leads to more coherent, paragraph-structured generations. These qualitative observations align with the quantitative results reported in Table 3, where ADLM achieves significantly higher MAUVE scores and lower generative perplexities than the existing diffusion language models.

**Example-I.** This example shows a sample generated using ADLM with 1000 sampling steps and the locked-in sampler from MDLM. While the generated content is largely coherent, it lacks structured paragraph breaks and clear discourse delimiters. The result reads as a continuous stream of thoughts rather than a segmented essay. Notably, even this partially structured output is both qualitatively and quantitatively (Table 3) superior to MDLM (Example-IV), which fails to organize ideas into meaningful paragraphs without breaking logical continuity. In subsequent examples (Example-II and Example-III), we show that increasing the number of sampling steps and incorporating remasking in ADLM significantly improves human-like text generation quality.

## Example-I: Output from Anchored Diffusion Language Model using locked-in sampler

<|endoftext|> knows they didn't have the answers to answer this.  It was a
solid piece of early Survivor and people glossed over it.  Sadly guess that's
where the Tribe go.  It wasn't a family.  It was based on a communitynt, and
that was everyone else.'' Probst Steve Watson.
Never as good as our pundit, we'll repair the damage we've done!
Tell a Random Story
A large facet of the new season will be to tell a random story, not sure
if it's how the players are portrayed.  But it's fairly obvious that this
specificity will be replaced by community.  The players will be given history,
and the people we've seen seemed less and more surprising as the episodes
went.
The host will highlight Tukuku Tshivu as a captain that's always tried to
respect the senior group and ditches the Amazonians in their barbaric ways.
After that we will see Toatia; some've described Tasha as someone who listens
to the players despite scheduling theirs, and she's a pretty Open Tribal so
far.
''And there's Andrea the Polarnesian,'' Probst continues, remembering the wood
thinner swimmer named Andrea whose entire life she's been challenging both
Rossheimerk and the predecessor to make IHOPE more racially inclusive.  Paris,
so much for standing up to her at the beginning, was willing to take no down.
This showed that the guild wouldn't just be her.  Andrea had been waiting
for free play, few months and months, and she stayed.  But boy did she know
that players with such wide access to free play at the beginning of her life
did require a fierce lane for attention.  There's a reason she was ripped
from her alliance for a reason we'll likely see players end up in separate
struggles with.  Paris.
For those players who can't wait to see the transition into the beginning,
the big reward before the game will be in reality.  If you waited for three
weeks for example, you'll still need fully footed shoes to get your reward,
you won't get old shoes.  The urgency of you signing up long enough to
get this reward makes as big of a social sense because that's the way the
society goes, as the rewards are physical.  The maximum rewards have yet to
be determined but we'll all keep tabs on the Breck King for progress.
Go Jaeger, the Trienge!  It's some fun!  This a system will work well but not
for an individual tribe in particular.  With two challenges means a lot of
people can only earn a full one day of fun and rewards.  And if you haven't
yet seen a single challenge, sure not the one played by the KaZinaks.  Julie,
Caleb, and Shirishio Polyhrine have chosen creative challenge themes for them,
albeit stoic winter difficult.
Winter's captains might get less access to raw gameplay, but it doesn't
have to be a shallow session where everyone goes over to hash out rewards
and conflicts.  It won't take place mid-season like winter, and it won't go
organically.  Book dramas are inevitable, but reboots have less incentive to
fail if the show does, and network commitments can to put everyone peoples
hustle in a sorech we don't.  I'm not sure this makes up for these two
things.
From What Built
Yes for the record, Jason and Miss were from a school group on the
controversial side of history.  Now Jay and the photographer will have
an unique way of representing the tribe while keeping the story relevant.
Though the cast will have something to do with it (gohan, skinnyfoots),
there will be some mismatched players who generally won't fill up the ample,
talented roster.  Which they clearly will.  Anyone stuck trying to play too
heavy on the other sides of the story?  Time out is Rachel Probst, who only
worked one Survivor Bannett final with EMA, '97:  Sandra's power was barely
that of Bailey Devevaorda, and Candice Burke was Pats to name a few.
Like Raj wound Katie's name, her tribe will have an extraordinary impact on
all of the stories and alliances.
Thai's Olympic tribe is another portfolio for the series.  Last season,
Zinke Group managed to represent 13 Pacific nations on the world stage
by visiting the indigenous peoples of them in various languages (Lahonese

```
    for example) and including First Peoples Tribal Councils; half of Taiwan's
    ''transformations'' were so successful that they typically received their
    return for their contributions.  It was a creative action, and even if
    one<|endoftext|>
```

**Example-II.** The output generated by ADLM exhibits coherent structure and articulates thoughts in paragraphs like human-written text. In this example, we use our pre-trained ADLM base model and apply the remasking sampler (Wang et al., 2025a) with 2048 sampling steps during inference. The model is able to maintain logical progression across long sequences, keeping together facts, statistics, and historical context into a coherent narrative.

This qualitative output complements the quantitative results reported in Table 3, where ADLM achieves the highest MAUVE score among all diffusion-based language models. MAUVE is known to correlate well with human evaluation of fluency and coherence, and this example provides evidence supporting ADLM's superior generation quality.

---

**Example-II: Output from Anchored Diffusion Language Model using remasking sampler**

```
<|endoftext|> health and demographics compared to the EU [ edit ]

Historical European birth rate map

for EU and European men and immigrants[4] [5]

B.R.E.O. [6] Birth rate by country Birth Rate co- Interval Interval 1910
50.4% (3.1%) 4.8% (12.6) 1910 61.5% (4%) 1920 51.1% (3.0%) 4.8% (12.6) 1920
63.5% (2.5%) 3.4% (1925) 59.4% (2.2%) 1930 62.7% (2.1%) 3.4% (1930) 60.2%
(2.0%) Other caregiving* 1910 49.1% (3.0%) 4.8% (12.6) 1910 63.1% (2.8%) 3.4%
(other) 1920 62.7% (2.5%) 1930 60.5% (2.1%) (b) First and (c) Birth:  average
(for girls, for boys, both to sexes.  d) Sex only and for all is for life and
people can't be combined, have to be under 15 years of age to have a means to
measure motherhood) Oldest age.  In the mid-19th century, at least half the
population had abortions before sterilization had taken.  Abortion increased
more rapidly in the twentieth century (nearly half in that time, 1.6 million
births, infant mortality over 2 million during the period up to 1960).  In
a study of Austrian prisons, 50% of the prisoners under the age of about 20
had been conscripted or maltreated for purposes before 1882.  Even in cases
where there was a thorough investigation, the participant had to remain in
cruel conditions, including medical experimentation, electric lashings, etc.,
while pregnancy rates were very low[ [ ].  The sterilized prisoners were not
only much more likely to have babies than the prisoners with no pregnancy;
they were less likely to have abortions.  Researchers found that the number
of births to sterilized prisoners between 1882 and 1930 was historically the
highest in the Czech Republic.

In Egypt, the penal code threatened local women and men with death if they
didn't marry foreigners.  Jihadists were slaughtered and people were forced
to flee out of Egypt's borders.  Likewise, foreigners women were forced to
marry men from Egyptians as well.  The Roma continued to rise, creating an
informal climate and assimilated life in the country.  Soon Egypt teemed with
lesbian women abandoned by migrants shortly after departing for the country.
This ethnic seeds played a role in the decline of the Ottoman Empire.  It is
widely agreed that this prejudice towards foreigners, and other groups, arose
among Orientals and had little influence at the time, probably because of the
security of limited exchanges between the two peoples.

In England, it was a family tradition to ferment wine up until the middle of
the 19th century.  After defeating the crown in 1791, the Shelburne brothers
flattened the cladding of the Osgoode Hill to form a pot for setting in their
wine cellar; in the process, they extracted quintense from wine.  Thus in the
first half of the nineteenth century, the use of preglycerine only lasted
three thousand years, when any other taxes were levied.  In Poland, torus
```

```
helped prevent tuberculosis before it became widespread, as the pilgrim
resisted wine, showing evidence of the disease.  The use of torus accounted
for the extinction of European flu.

Prevention measures [ edit ]

Pumps and band vein pumping apparatus

Because of increased RME and MMB in the central 20s researchers set about
examining how physical physiological factors might affect pregnancy and
its prevention.  As had been done in the epidemics, there were two types
of PNEA delivery devices used:  cardiac pumps and ECG band pumps.  The
former was placed in the blood vessel, through which 20 mL/mm flow of gold
tubing at 70,744 feet (21,131 m) was rated for blood pressure and had a
continuous output of mL (100 yps) was produced and blood pressure measured.
Approximately 7,000 km (3,200 mi) of gold tubing was used for each of these
two devices.  The hg pump that was placed in the patent vein could go from 5
cm (1.8 in) to 5 cm (1.5 in) of tubing to 3,000 feet/1,800 m (2,100 m/720 ft).
The band pump had the same potential but the added disadvantage of releasing
gas too hard.  More than 100 million births were prevented by these pumps
every year.[8]

Figure 2 Heart signs in a fetus during conception of the mother.

The gauges measure blood vessel pressure and blood pressure for 24 h during
the first month (320137 weeks) after conception, then a sample cylinder was
placed in<|endoftext|>
```

**Example-III.** In this example, we use 4096 sampling steps for ADLM with remasking sampler. The generated text demonstrates strong discourse-level coherence, well-structured paragraphs, and natural transitions between topics. For instance, the model begins with a detailed sports commentary and transitions smoothly into a socio-political news report. These shifts are logically segmented with paragraph-level breaks and consistent timelines (e.g., 18:00 PST, 30:00 PST), giving the impression of human-authored journals. This qualitative improvement aligns with the quantitative results reported in Table 3, where ADLM achieves the highest MAUVE score among existing DLMs.

**Example-III: Output from Anchored Diffusion Language Model using remasking sampler**

```
<|endoftext|> were pretty good in that regard.  Dwight Howard chipped in 15
points, and who's to argue that defense doesn't deserve a stat of the award?

Stephen Curry, who was on the floor as soon as I got there, finished with
27 points.  Wesley Matthews put in a great effort as well.

Washington (Curry 30-31, Wall 80-78, Wall 82-7)

I listened to more of this game than originally planned.  The Wizards took
control of the game when Otto Porter scored 22 points.

Takeaways:

The Wizards took control of the game during the first quarter, mostly courtesy
of Dennis Schroder and Shaun Livingston.  In the second minute, Stephen Curry
hit a perfect layup that bounced to the hoop for a dunk, Durant scored two
quick points to make it 15-7 with about seven minutes left, and in the third
minute Jameer Nelson hit a layup in the corner from range to make it 18-9.

18:00 PST

Wall and Marcus Thornton each dribbled their way through traffic and made
two threes, the first by Andre Drummond with three minutes left to record his
```

```
10th career triple-double, and the second by Kevin Durant with five minutes
left.  However, both shots were blocked at the basket.  Kevin Durant missed
most of the rest of it with an ankle injury, and Austin Rivers, getting a
last-second shot off on him pretty badly, tried to knock the ball down, but
couldn't as the ref just waved it back.  Then Rivers shot, then Thornton
drove his way in front of Durant, and Wall had to do a reverse dribble move
to knock Curry's first shot free.  This was probably the highlight of the
game; the defense wasn't matched with the offense very well against the
Thunder from here on out.

Halftime:

Oklahoma City gave up 10 points for coming up in the final minute, and Zach
LaVine came off the bench.  In the fourth minute Russell Westbrook ran in
a low drive to post up screen while the ball was on the floor, but with 10
seconds left in the game, Zach LaVine stepped onto the floor and slammed the
ball in the basket for the bucket.

30:00 PST

Tony Allen was brought in for Monta Ellis, and he just could not start.  He
tried his way to the corner for a 3-pointer, but he was shot.  Nerlens Noel
quickly got up and knocked it back down low, and Allen ran back into the
stands of the arena, knocking the ball down high and out of bounds.

Oklahoma City's bench never stopped suiting as the game went on.  This is
a team that is making a leap in the NBA, with DeMarcus Cousins and Chauncey
Billups leading the way, and it's just the right thing to do in such a crazy
situation.<|endoftext|>(CN) Hundreds of Seattle residents Tuesday marched to
Columbia City to protest the construction of Tesla's new headquarters in one
of the largest nationwide demonstrations in years, as Seattle police clashed
with student protesters at a park near Washington State University.

Thousands of the student marchers gathered at around 2:30 p.m.  in Seattle,
capital of Western Washington as they confronted riot police carrying rocks,
rocks bottles, and gas masks.  Large demonstrations of this kind are nearly
every weekly in the sprawling city.  Tesla CEO Elon Musk said he didn't know
how much the new headquarters could cost him but that it would be enough for
the car company to complete its plans.  However, the cost estimate is unknown.
In the debating point, the city of Seattle, a state infrastructure agency,
the Metropolitan Transportation Authority and the thousand-plus protesters
wait for a judge's approval to assess possible damages.

Tesla's headquarters is the second-tallest office building proposed in
Greater Snohomish County.  It is planned on the Seattle Pacific University
campus.  Tesla also has a 14-story supertall building on Mercer Street that
would also include mixed-use office space.  There is also construction of
a 50,000 square foot garage that is expected to open next year.  There is
also a parking garage on the 8.7 acre property.  No residences are listed in
Tesla's property plan approved by federal agencies.

Tesla and company officials are expected to have the newest office building
at 240,000 square feet made for public access public, once it's slated for
completion.  The company is already in court for a waiver that allows it to
maintain the building because it is in a park.  A lawyer for the company said
on April 27 that a judge in Seattle is expected to rule on the waiver, which
would allow the building to be enabled.  This is one of several lawsuits that
have emerged in recent weeks since Tesla's new headquarters was announced in
March.  Across the U.S<|endoftext|>
```

**Example-IV.** For completeness, we also include samples generated using MDLM (Sahoo et al., 2024). In contrast to ADLM, the generated text lacks coherent paragraph structure and clear discourse delimiters.  The output tends to drift semantically and fails to maintain logical continuity across

segments. This qualitative degradation is consistent with the lower MAUVE scores and high Gen PPL reported for MDLM in Table 3, reflecting lower human-like text generation quality.

---

**Example-IV: Output from Masked Diffusion Language Model using locked-in sampler**

```
<|endoftext|> told local reporters. ''Any interlocutor was useful for formal
consulting.''UCU basketball coach, classmate of three, David Irving, is a
teacher at Memorial High and is among the same players who recently won
his credentials in PE Illustrated for the category.  Four longtime players
make such notable contacts:  two career elons (not Columbus native champ
Bobby Pulfres; Erie it's Duane Wright than Vin Diesel) and he's the current
fourth student at the entire campus, as far as entrance per pupil is well.
According to Wright, Shaw was about to thirsty with traders celebrating
with an invented flavoral beer and the extra muggers when he approached the
team with a shot on Irving, something he was doing while across on Thursday
afternoon on an acquired Colt rifle.  He even spoke as he tried to reach out
with his fists and pose to look good and enhance his visual presentation.
During a photo shirt Shaw took on the stool, his head fell like the head of
fungus straight for a hanging liquid.  He could not believe the team were
able to get through.''There were around six or seven guys and somebody takes
shots,'' Aggaeed says, "and once they got ready they sat beside them laughing
for a bit.  And everyone was super friendly.  They usually get been late
and then it would be 104 o'clock while we went...''It surprising people a
lot for the guys here.  UCU quarterback Jeffrey Zifits calls him a group
of hives stirring spirits.  ''This group is meant for the senior players and
one of the best young performers,''the team or the officials or coaches or
whatever, but it's not meant for Gasta-Grader," Zifit says."You could see
it first hand each time we went fishing with fans...  It wasn't just people
throwing stuff.''It's the biggest thing they did after third windows and they
put it out there,'' Zifits, "we didn't let them but it brought more energy
here.  When we came showled them it was like a CFL game.  The fun played with
energy and it's great.''It is definitely in the vein of the spirit of cabo,
kind of a bar that lets people in two buckets and for Wright, if you can or
can't squeeze your lip while golfing, go bowling and play.''You expect to be
at the conference in this room that vacillums with everybody else,'' Wright
says.  ''So, just remember, and our motto in a group is 'Reason,' when it
comes to material Value, we're a group and all our resources are all being
used by the future to promote the future.  It works out.  They do it by
laughing.''Pulkies'' students really do love coming back," says Shaw, then a
member of the PE group.  "Jefferson will still come back and brilliant and
all that.  'We can't let them shut them down.  If it goes their way, they are
not letting us down.'  Working title:  Mentors, I need to get out here we're
all together with students to make it through this."PS<|endoftext|>By Colin
Collins Piper for UCMP
PRINT CLOSE British university governors unveiled the cancellation of the
student-held protest in defiance of the declaration it government has gone
into a pitch to the Liberal Democrats to sack it for several hours for its
register of support for atheists.
The university's slaughter of Scottish-encompassing religious figures and
included tacit approval by both houses of parliament before the Niche
government was in administration.
It is understood that the load followed the closure of the university's
reserve vice chancellor subsequently who died of Parkinson's syndrome as
his wife's tombstone was on fire.
Catherine Baird of Parliamentary Coalition for Government (Scrabble) leader
Lassie Mann joined further the condemnation of the stalls.
Ms Baird said:  ''Good thing the students approached the university and they
were confident that you found much solace.
''It's a sad day in British life and I think it would be tragic if we ever had
this again but this is a very simple decision, to be clear.''
Scalric-chancellor Jack Major who attended the protest told a press
conference:  ''It has been very challenging but I would like to later
apologise personally in the language of protesters of what happened.
```

```
‘‘The Department of Students and Service at the University of California,
Berkeley was placed following people's request and it is no surprise that the
university is now in administration but those schools needed to be confirmed
afterwards that they would face disciplinary action.
Directors Major said: ‘‘I'm a minority Instructors were very disappointed
about the protest within the university and the university'<|endoftext|>
```

## C.2    Auto-Regressive Models

This section provides an extended discussion of our anchoring mechanism as applied to autoregressive language models. We begin by contrasting standard autoregressive training with our proposed anchored autoregressive training framework. In §C.2.1, we outline the AR baselines used for comparison. In §C.2.2, we describe the training and evaluation benchmarks. §C.2.3 provides implementation details necessary for reproducibility. In §C.2.4, we demonstrate the benefits of anchoring for likelihood modeling on the OWT benchmark. Finally, §C.2.5 presents results on math and logical reasoning tasks, showing improved reasoning capabilities due to Anchored Chain-of-Thought (ACoT) fine-tuning.

**Discussion on standard AR and anchored AR models.** Figure 4 illustrates the standard training setup for autoregressive large language models (LLMs), where each token is predicted based on its left context. This sequential decoding lacks structural guidance to prioritize important tokens.

In contrast, as shown in Figure 5, we decompose LLM training into a two-stage process. In the first stage, an anchor network is used to predict the likelihoods of important tokens (e.g., 'cat' and 'dog' highlighted in blue)—referred to as *anchor tokens* or [ANT]—conditioned on the preceding context. An anchor token is not necessarily the next token in a sequence. Its prediction is supervised using a cross-entropy loss applied only at the anchor position.

In the second stage, a *lightweight* autoregressive LLM (half the number of transformer layers used in Figure 4) is trained using the standard next-token prediction objective, but conditioned on the output logits from the anchor network. Rather than sampling tokens and re-embedding them, we project the anchor logits through a linear layer and feed the resulting representations directly into the LLM transformer, similar to the ADLM setup discussed in §C.1.3. This allows gradients from the LLM's output to backpropagate through the projection layer to the anchor network, enabling joint training. Importantly, this anchoring mechanism allows the model to "look ahead" by leveraging important tokens in a sequence, which improves reasoning without significantly altering the decoding process.

Formally, our anchored autoregressive training loss becomes:

$$\mathcal{L}_{\text{A2R}}(\psi, \varphi) = -\mathbb{E}_{X \sim q}\left[\sum_{l=2}^{L} \log p_\psi(X^l | Y_\varphi^{1:l-1}(X^{1:l-1}))\right] - \gamma \mathbb{E}_{X \sim q}\left[\sum_{l=2}^{L} \log r_\varphi(Y^l | X^{1:l-1})\right],$$

where $Y = \mathcal{A}(X)$ is the sequence of anchors obtained through the operator $\mathcal{A}$ as in ADLM (§3).

### C.2.1    Compared Baselines

We compare our method against a diverse range of latent reasoning and chain-of-thought (CoT) approaches. We also include baselines that do not use reasoning traces during training. To ensure a fair comparison, we use the same base implementation from prior work (Hao et al., 2024) and integrate our anchoring mechanism into the identical multi-stage training pipeline[8].

- **CoT** (Wei et al., 2022): The base model is fine-tuned using the question as context and the concatenated (reasoning, answer) as tokens contributing to training loss. During inference, the model first generates a reasoning trace and then the final answer.
- **No-CoT**: A standard supervised fine-tuning baseline that does not use reasoning traces. The model is trained to predict the answer directly given the question as context.
- **Pause Token** (Goyal et al., 2024): A pause token is inserted between the question and answer without using reasoning traces. The number of pause tokens is set to match the training stages of COCONUT, to ensure fair comparison.

---

[8]https://github.com/facebookresearch/coconut

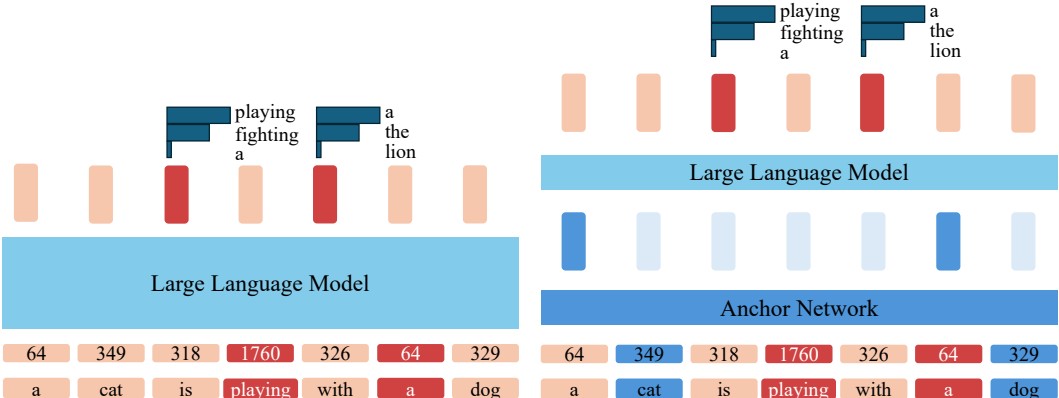

Figure 4: **Training of standard autoregressive (AR) models.** A neural network is trained to predict the next token using causal attention (left-to-right context). All tokens contribute equally to the training loss, and the model treats the sequence uniformly without structural guidance.

Figure 5: **Training of anchored autoregressive (A2R) models.** An anchor network first identifies important tokens (e.g., 'cat', 'dog' shown in blue), which are supervised via an auxiliary anchor loss. A lightweight LLM is then trained to predict the next token based on anchored predictions.

- **Pause-as-Thought in COCONUT**: This variant replaces the continuous latent thoughts in COCONUT with pause tokens while following the same multi-stage training schedule.
- **iCoT** (Deng et al., 2024): Internalizes the reasoning trace into intermediate transformer layers, allowing the model to reason implicitly without generating explicit steps for reasoning.
- **COCONUT** (Hao et al., 2024): This method uses continuous latent representations (referred to as continuous thoughts) instead of discrete reasoning tokens, and inserts these continuous thoughts directly in the embedding layers before processing through the transformer block. This is motivated by the idea that reasoning can often be more intuitively encoded in a continuous latent space, especially when explanation using words is difficult.
- **BoLT** (Ruan et al., 2025): Bootstraps its "reasoning to learn" ability using latent thoughts and self-distillation. This is useful in data-constrained environments, such as GSM8K.

One key distinction between methods that introduce new tokens between the question and (reasoning, answer)—such as Pause, COCONUT+Pause-as-Thought, and our method (ACoT)—is that ACoT explicitly applies an *anchor loss* on the inserted [ANT] tokens. This external supervision provides semantic guidance on which tokens are important, resulting in better performance on both language modeling and complex reasoning tasks.

### C.2.2 Training and Evaluation Benchmarks

**Text (OWT).** We conduct next-token prediction experiments on the OWT (Gokaslan & Cohen, 2019) to evaluate likelihood modeling in the AR setting. In this setting, the model does not have access to structured reasoning traces as in the Math and Logic benchmarks described below. The training and validation splits follow the same setup used in the diffusion experiments (§C.1.2).

**Math (GSM8K).** The GSM8K dataset (Cobbe et al., 2021) contains grade-school math word problems requiring multi-step arithmetic reasoning. Each problem is presented as a natural language prompt followed by intermediate reasoning steps and a final answer. Successful modeling on GSM8K requires accurate understanding of the question through natural language and execute multi-step arithmetic reasoning to derive the final answer.

**Logic (ProntoQA, ProsQA).** For logical reasoning, we evaluate on ProntoQA (Saparov & He, 2023) and ProsQA (Hao et al., 2024). ProntoQA consists of deductive reasoning tasks, where the model must verify or falsify a hypothesis using a given set of symbolic rules. We use the 5-hop variant, which serves as a controlled benchmark for logical reasoning.

ProsQA (Hao et al., 2024) is a more challenging planning task that requires navigating through complex reasoning graphs to arrive at the correct answer. The model must identify and follow valid

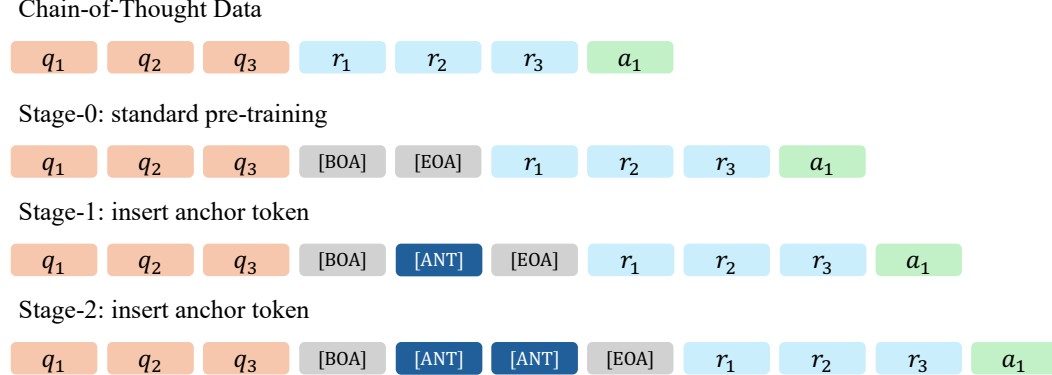

Figure 6: **Multi-stage training pipeline for Anchored Chain-of-Thought (ACoT).** Here, [BOA] and [EOA] denote the beginning and end of anchors, respectively. Many reasoning traces contain redundant information, increasing entropy and making the reasoning process harder to learn. By supervising the model through a small set of important tokens extracted from the reasoning trace, ACoT encourages more structured intermediate computations, guiding the model to reason in a more targeted and interpretable way. To reduce the number of additional tokens produced, we drop reasoning tokens for every [ANT] insertion. For example, we drop $r_1$ in Stage-1 and $r_1, r_2$ in Stage-2 to demonstrate this phenomenon.

inference paths among distractors. We follow the multi-stage fine-tuning procedure described in (Hao et al., 2024) to enable direct comparison with previous baselines.

**Evaluation Metric.** For all reasoning benchmarks, we use accuracy as the primary evaluation metric, measuring whether the final answer produced by the model matches the ground truth.

### C.2.3 Implementation Details

**GSM8K.** To ensure a fair comparison with prior baselines, we closely follow the training protocol used in COCONUT (Hao et al., 2024). Below, we highlight the key distinctions in our ACoT training setup and recall relevant aspects of the original training procedure for completeness.

The primary distinction in our approach is the use of [ANT] tokens to guide the generation of reasoning traces and final answers. Unlike COCONUT, we do not remove intermediate reasoning steps when inserting [ANT] tokens. Instead, we treat anchors as auxiliary indicators that help the model focus on important context of the reasoning trace.

For instance, in Question 1 of Table 10, we identify the tokens 16, 3, 4, 9, 2, and 18 as important due to their role in the arithmetic operations leading to the final answer. When using two [ANT] tokens, we select the initial subset of these important tokens—in this case, 16 and 3—to serve as anchors. These anchors act as planning cues that guide the model's reasoning trajectory.

Following COCONUT, we adopt a multi-stage training strategy shown in Figure 6 and train all models for a total of 25 epochs. We report results from the best checkpoint among these 25 epochs. The first stage is equivalent to standard CoT pre-training, where no [ANT] tokens are inserted between the question and the (reason,answer) tokens. In the second stage (epochs 1–3), we introduce one [ANT] token and continue training for 3 more epochs. In the third stage (epochs 4–6), we add one more [ANT] token and train for another 3 epochs. After a maximum of 2 [ANT] tokens have been introduced in the span of 6 epochs, we continue training for the remaining epochs (up to 25) using supervised fine-tuning.

**ProntoQA.** We follow a similar multi-stage training procedure for the ProntoQA benchmark, with a total of 50 training epochs. Given the longer context and deeper reasoning chains in this dataset, we introduce [ANT] tokens progressively over six stages, following an initial stage of CoT fine-tuning.

On this benchmark, we consider the valid nodes in the reasoning trace as important tokens, since they anchor the generation of the ground truth reasoning steps that ultimately lead to the correct answer.

Table 9: **Anchoring improves autoregressive modeling on OWT.** Test perplexities (PPL; ↓) for standard AR models and our anchored variant (A2R) at various training scales. † Results from (Sahoo et al., 2024). A2R consistently improves perplexity by introducing a two-stage prediction process: anchor tokens are first predicted, then used to guide next-token prediction.

| Model | PPL ($\downarrow$) | Tokens |
|---|---|---|
| AR (retrained) | 17.94 | 110B |
| AR$^\dagger$ | 17.54 | 262B |
| AR (retrained) | 17.53 | 262B |
| AR (retrained) | 17.26 | 524B |
| **A2R (ours)** | 17.29 | 110B |
| **A2R (ours)** | 16.23 | 262B |
| **A2R (ours)** | **15.86** | 524B |

For example, in Question 1 of Table 11, we treat the tokens `Alex`, `Tumpus`, `Gorpus`, `Wompus`, `Sterpus`, `Brimpus`, and `Happy` as anchor tokens. When training with six `[ANT]` tokens, we select the first six tokens from this list as supervised labels.

The first stage consists of CoT training without any anchor tokens. In each subsequent stage, we insert one additional `[ANT]` token into the input and train for 5 epochs. After completing six such stages (i.e., 30 epochs in total), each involving an additional anchor token (up to 6), we continue standard SFT (without further changes to the number of `[ANT]` tokens) for the remaining 20 epochs.

**ProsQA.** For the ProsQA benchmark, we follow the same multi-stage training procedure as used for ProntoQA, with one key distinction: at each stage where a new `[ANT]` token is introduced, we remove one reasoning step from the groundtruth reasoning steps used in SFT. This encourages the model to learn missing reasoning steps based on the guidance provided by `[ANT]`. As in ProntoQA, we progressively insert six `[ANT]` tokens over six stages, training each stage for 5 epochs, totalling up to 50 epochs. In Question 1 of Table 13, we identify `Tom`, `Terpus`, `Brimpus`, and `Lempus` as anchoring tokens. When training with six `[ANT]`, we use the first six valid nodes[9] identified in the reasoning trace as supervised labels.

### C.2.4 Generative Modeling using Anchored Auto-Regressive Models

**Results on OWT.** We evaluate the impact of anchoring on autoregressive language models using the OWT benchmark. Similar to our approach in diffusion language models, we decompose the standard next-token prediction task into two stages. In the first stage, an anchor network predicts a distribution over semantically important tokens (e.g., low-frequency or structurally informative tokens), referred to as anchor tokens or `[ANT]` for short. These anchor predictions help the model identify key parts of the input that provide better context for effectively predicting the next token.

As shown in Table 9, our anchored autoregressive model (A2R) outperforms standard AR models at 3 training scales. When trained on 262B tokens, A2R achieves a perplexity of 16.23, surpassing the corresponding AR baseline (17.53). At the largest scale (524B tokens), A2R further reduces perplexity to 15.86, surpassing the best AR baseline (17.26) by a margin of 1.4 PPL. Interestingly, while performance gains in standard AR pretraining tend to saturate around a perplexity of 17, our A2R pretraining continues to improve, reducing perplexity from 17.29 to 15.86 as the number of training tokens increases from 110B to 524B. This consistent improvement across scales demonstrates that anchoring provides a more informative latent representation, helping the model better estimate token likelihoods.

These results confirm that anchoring is not limited to diffusion models and is broadly applicable to autoregressive architectures. It enables improved context modeling through an anchor network without requiring many architectural changes to the base model.

---

[9] If the number of `[ANT]`s is larger than the available anchors, then we ignore the loss on extra `[ANT]`s.

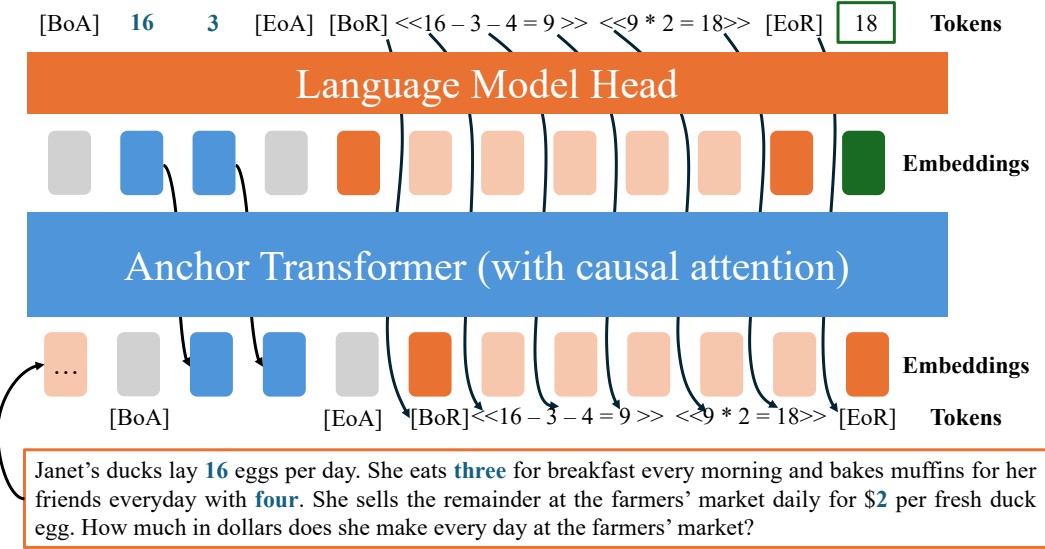

Figure 7: **Anchored Chain-of-Thought (ACoT)** training pipeline. In this example with two anchor tokens, we preprocess the original reasoning trace «16–3–4=9» «9*2=18», and extract the numbers '16' and '3' as the anchor tokens (the first two numbers in the reasoning trace). We then create a modified training trace that is augmented with these anchor tokens (top line in figure). Example trace from GSM8K dataset.

### C.2.5 Improved Reasoning using Anchored Chain-of-Thought

Decoding order is known to affect inference quality, as established in classical structured prediction literature (e.g., via topological or lexical ordering) (Jordan et al., 1999; Bengio et al., 2003; Koller & Friedman, 2009). However, most LLMs (especially autoregressive ones) follow a strictly left-to-right generation scheme, which often leads to shallow, sequential reasoning. This bias can hinder tasks where intermediate computations depend on later context or global structure.

Our proposed Anchored Chain-of-Thought (ACoT) addresses this issue by leveraging important tokens (such as root nodes in the underlying reasoning graph) prior to decoding. These tokens serve as intermediate supervision points, enabling the model to focus on the most critical sub-computations early in the reasoning trace. We then apply the standard next-token prediction loss conditioned on these anchor predictions, effectively guiding the model toward more structured and globally coherent solutions. Thus, anchors allow the model to *look ahead* and think *non-sequentially* while retaining compatibility with standard autoregressive training and decoding pipelines.

During training, as illustrated in Figure 7, the loss function for the Anchor Network is supervised using future important tokens. As a consequence, during inference, this results in anchor predictions having logits with higher likelihood on future important tokens. Thus, the auto-regressively generated tokens depend not only previous tokens (causal attention), but also implicitly on potential future important tokens that are encoded within the anchor predictions (see Figure 2 in the main paper). This results in out-of-order reasoning as discussed below.

**Results on GSM8K.** To evaluate the effectiveness of anchoring in complex reasoning tasks, we apply our method ACoT on GSM8K (Cobbe et al., 2021). Table 10 presents qualitative examples comparing standard CoT reasoning traces with those produced by our ACoT model. Each example includes the input question, ground truth CoT trace, the model's full output (including anchor tokens), and the extracted final answer. Table 5 contains the quantitative results.

A key insight from Question 1 is that standard CoT processes the question in a purely left-to-right manner. It computes 16 - 3 - 4 = 9, and then 9 * 2 = 18, *following the order in which quantities appear in the question*. In contrast, ACoT introduces [ANT] to capture important tokens, which allows it to reason more globally. Specifically, ACoT first computes 3 + 4 = 7 to aggregate all consumption before subtracting from the total (16 - 7 = 9), a pattern more aligned with human

intuition. This demonstrates how anchoring enables a "look-ahead" planning behavior, in contrast to the left-to-right decoding bias of standard CoT.

In Question 3, ACoT diverges from the gold trace and produces an incorrect final answer. This example highlights a limitation: while anchoring allows more flexible planning, it still depends on correctly identifying and utilizing important tokens.

CoT and our method ACoT share a similar experimental setup, with the only exception of anchoring via multi-strage training. While standard multi-stage training as in COCONUT reduces the performance from 42.9% to 34.1%, our anchoring mechanism prvoides a substantial gain of 11.1% over these latent reasoning methods (such as COCONUT and iCoT), and a notable gain of 2.3% over CoT.

Overall, these examples demonstrate that anchoring provides a more flexible and semantically guided framework for reasoning, enabling the model to break free from strictly left-to-right token prediction. This leads to improved planning and more human-like reasoning behaviors, especially in problems requiring multi-step arithmetic reasoning as in GSM8K.

**ProntoQA.** As shown in Table 11, our ACoT model generates valid reasoning traces and correctly predicts the final answer across symbolic reasoning tasks in ProntoQA. In the first example, the question asks whether "Alex is happy" based on a complex chain of relational facts. The standard CoT trace progresses step-by-step by chaining multiple class membership relations (e.g., "Alex is a tumpus", "Tumpuses are gorpuses", ..., "Brimpuses are happy"). Our ACoT model, guided by six [ANT] tokens, learns to identify and attend to these anchors in the reasoning process.

Recall from Table 5 that CoT, COCONUT, and ACoT share nearly identical training setups, with the only distinction being the use of anchor tokens in ACoT and continuous thoughts in COCONUT. While standard multi-stage training in COCONUT leads to a marginal improvement from 98.8% (CoT) to 99.8%, our anchoring mechanism further enhances performance. By incorporating explicit supervision through anchor tokens, ACoT achieves a perfect accuracy of 100% on this relatively easier benchmark, demonstrating the effectiveness of anchoring in logical reasoning.

**ProsQA.** This experiment illustrates that ACoT can internalize and reconstruct reasoning traces through anchor tokens, similar to latent reasoning methods such as iCoT and COCONUT. As in those prior works, our model avoids explicitly generating intermediate reasoning steps while still maintaining logical coherence in the final answer.

We evaluate ACoT under a variant of the training setup where we do not remove any reasoning steps (i.e., following the setup used in ProntoQA). In this setting, ACoT achieves 81% accuracy—outperforming standard CoT (77.5%) but still falling short of the 97.0% achieved by COCONUT. Prior work (Hao et al., 2024) has shown that gradually removing reasoning steps during multi-stage training provides a significant performance boost on ProsQA. Following this recommendation, we integrate reasoning step removal into our ACoT fine-tuning. This modification improves our model's accuracy from 81% to 97.3%. Importantly, ACoT generates much less tokens compared to CoT and COCONUT while surpassing their accuracy, as shown in Table 12.

In the first example from Table 13, the question asks whether "Tom is a lempus or scrompus", requiring a multi-hop reasoning chain through symbolic class hierarchies. The ground truth reasoning trace proceeds via three steps: (1) Tom is a terpus, (2) Every terpus is a brimpus, and (3) Every brimpus is a lempus. Our ACoT model accurately identifies and anchors this chain using six [ANT] tokens, and directly generates the correct final inference: Tom is a lempus. Notably, this is done without producing intermediate reasoning steps, demonstrating that the model has internalized the underlying inference structure via the anchor tokens. These results highlight that anchoring provides complementary benefits to existing fine-tuning strategies (such as CoT and COCONUT), and further boosts the model's ability to solve complex reasoning problems.

Table 10: Examples of math word problems with reasoning traces from GSM8K (Cobbe et al., 2021). Each row shows the input question, groundtruth reasoning trace (CoT), answer, our model's full output sequence, and the final extracted answer. Our model implicitly reasons through anchoring tokens ([ANT]) and produces reasonable traces before computing the final answer.

| | |
|---|---|
| **Question 1** | *Janet's ducks lay 16 eggs per day. She eats three for breakfast every morning and bakes muffins for her friends every day with four. She sells the remainder at the farmers' market daily for \$2 per fresh duck egg. How much in dollars does she make every day at the farmers' market?* |
| **CoT** | «16-3-4=9» «9*2=18» |
| **Answer** | **18** |
| **Full Output** | [ANT] [ANT] «3+4=7» «16-7=9» «9*2=18» |
| **Extracted Output** | 18 |
| **Question 2** | *A robe takes 2 bolts of blue fiber and half that much white. How many bolts in total?* |
| **CoT** | «2/2=1» «2+1=3» |
| **Answer** | **3** |
| **Full Output** | [ANT] [ANT] «2/2=1» «2+1=3» |
| **Extracted Output** | 3 |
| **Question 3** | *Josh decides to try flipping a house. He buys a house for \$80,000 and then puts in \$50,000 in repairs. This increased the value of the house by 150%. How much profit did he make?* |
| **CoT** | «80000+50000=130000» «80000*1.5=120000» «120000+80000=200000» «200000-130000=70000» |
| **Answer** | **70000** |
| **Full Output** | [ANT] [ANT] «50000*.15=7500» «7500+50000=57500» «57500-80000=-32500» |
| **Extracted Output** | -32500 |
| **Question 4** | *James decides to run 3 sprints 3 times a week. He runs 60 meters each sprint. How many total meters does he run a week?* |
| **CoT** | «3*3=9» «9*60=540» |
| **Answer** | **540** |
| **Full Output** | [ANT] [ANT] «3*60=180» |
| **Extracted Output** | 180 |
| **Question 5** | *Every day, Wendi feeds each of her chickens three cups of mixed chicken feed, containing seeds, mealworms and vegetables to help keep them healthy. She gives the chickens their feed in three separate meals. In the morning, she gives her flock of chickens 15 cups of feed. In the afternoon, she gives her chickens another 25 cups of feed. How many cups of feed does she need to give her chickens in the final meal of the day if the size of Wendi's flock is 20 chickens?* |
| **CoT** | «3*20=60» «60-15-25=20» |
| **Answer** | **20** |
| **Full Output** | [ANT] [ANT] «15+25=40» «40-20=20» |
| **Extracted Output** | 20 |

Table 11: Examples of logical reasoning tasks with symbolic reasoning traces from Pronto-toQA (Saparov & He, 2023). Each row shows the input question, groundtruth reasoning trace (CoT), answer, the model's generated output sequence, and the extracted final answer. Our model implicitly reasons through anchoring tokens ([ANT]) to infer logical relationships.

| | |
|---|---|
| **Question 1** | *Tumpuses are floral. Each jompus is not melodic. Tumpuses are numpuses. Gorpuses are opaque. Each grimpus is small. Sterpuses are dumpuses. Tumpuses are gorpuses. Every lorpus is not happy. Every wumpus is a shumpus. Each gorpus is a grimpus. Shumpuses are slow. Every dumpus is overcast. Gorpuses are wumpuses. Vumpuses are dull. Sterpuses are brimpuses. Numpuses are not metallic. Jompuses are impuses. Brimpuses are happy. Sterpuses are hot. Brimpuses are vumpuses. Wumpuses are sterpuses. Brimpuses are zumpuses. Wumpuses are sour. Alex is a tumpus. Alex is a jompus. True or false: Alex is happy.* |
| **CoT** | Alex is a tumpus.  Tumpuses are gorpuses.  Alex is a gorpus.  Gorpuses are wumpuses.  Alex is a wumpus. Wumpuses are sterpuses.  Alex is a sterpus.  Sterpuses are brimpuses.  Alex is a brimpus.  Brimpuses are happy. Alex is happy. |
| **Answer** | **True** |
| **Full Output** | [ANT][ANT][ANT][ANT][ANT][ANT] Alex is a tumpus. Tumpuses are gorpuses.  Alex is a gorpus.  Gorpuses are wumpuses.  Alex is a wumpus.  Wumpuses are sterpuses. Alex is a sterpus.  Sterpuses are brimpuses.  Alex is a brimpus.  Brimpuses are happy.  Alex is happy. |
| **Extracted Output** | **True** |
| **Question 2** | *Brimpuses are sterpuses. Each grimpus is a numpus. Lorpuses are angry. Grimpuses are moderate. Each dumpus is a lempus. Each lempus is cold. Gorpuses are yumpuses. Every rompus is not sunny. Each dumpus is a gorpus. Yumpuses are bitter. Each grimpus is a vumpus. Gorpuses are not large. Brimpuses are lorpuses. Every vumpus is discordant. Every numpus is shy. Brimpuses are not brown. Dumpuses are floral. Gorpuses are grimpuses. Each shumpus is a wumpus. Vumpuses are rompuses. Tumpuses are brown. Every shumpus is not metallic. Vumpuses are brimpuses. Polly is a dumpus. Polly is a shumpus. True or false: Polly is brown.* |
| **CoT** | Polly is a dumpus.  Each dumpus is a gorpus.  Polly is a gorpus.  Gorpuses are grimpuses.  Polly is a grimpus. Each grimpus is a vumpus.  Polly is a vumpus.  Vumpuses are brimpuses.  Polly is a brimpus.  Brimpuses are not brown.  Polly is not brown. |
| **Answer** | **False** |
| **Full Output** | [ANT][ANT][ANT][ANT][ANT][ANT] Polly is a dumpus.  Each dumpus is a gorpus.  Polly is a gorpus.  Gorpuses are grimpuses.  Polly is a grimpus.  Each grimpus is a vumpus.  Polly is a vumpus.  Vumpuses are brimpuses. Polly is a brimpus.  Brimpuses are not brown.  Polly is not brown. |
| **Extracted Output** | **False** |

Table 12: **Accuracy and Token Usage on ProsQA.** ACoT outperforms prior latent reasoning methods, including COCONUT, while using fewer or comparable reasoning tokens during inference. † reported in COCONUT.

| Method | ProsQA | |
|---|---|---|
| | Accuracy (%) | # Tokens |
| No-CoT[†] | 76.7±1.0 | 8.2 |
| Pause Token[†] (Goyal et al., 2024) | 75.9±0.7 | 8.2 |
| CoT[†] (Wei et al., 2022) | 77.5±1.9 | 49.4 |
| iCoT (Deng et al., 2024) | **98.2±0.3** | 8.2 |
| COCONUT[†] (Hao et al., 2024) | 97.0±0.3 | 14.2 |
|   - *Pause*[†] | 96.6±0.8 | 8.2 |
| **ACoT (ours)** | 97.3±0.2 | 8.2 |

Table 13: Examples of logical reasoning tasks with symbolic reasoning traces from ProsQA (Hao et al., 2024). Each row shows the input question, groundtruth reasoning trace (CoT), answer, our model's generated output sequence, and the extracted final answer. Our model implicitly reasons through anchoring tokens ([ANT]) to infer logical relationships. In this experiment, we employ the gradual CoT removal scheme used in prior works (Deng et al., 2024; Hao et al., 2024) to demonstrate the reasoning ability in the anchored latent space without producing word tokens.

| | |
|---|---|
| **Question 1** | *Every shumpus is a rempus. Every shumpus is a yimpus. Every terpus is a fompus. Every terpus is a gerpus. Every gerpus is a brimpus. Alex is a rempus. Every rorpus is a scrompus. Every rorpus is a yimpus. Every terpus is a brimpus. Every brimpus is a lempus. Tom is a terpus. Every shumpus is a timpus. Every yimpus is a boompus. Davis is a shumpus. Every gerpus is a lorpus. Davis is a fompus. Every shumpus is a boompus. Every shumpus is a rorpus. Every terpus is a lorpus. Every boompus is a timpus. Every fompus is a yerpus. Tom is a dumpus. Every rempus is a rorpus. Is Tom a lempus or scrompus?* |
| **CoT** | `Tom is a terpus.  Every terpus is a brimpus.  Every brimpus is a lempus.` |
| **Answer** | **Tom is a lempus.** |
| **Full Output** | `[ANT][ANT][ANT][ANT][ANT][ANT] Tom is a lempus.` |
| **Extracted Output** | **Tom is a lempus.** |
| **Question 2** | *Sally is a zhorpus. Every yumpus is a fompus. Every zhorpus is a rempus. Every rompus is a sterpus. Every kerpus is a timpus. Stella is a yumpus. Every zhorpus is a zumpus. Every wumpus is a yumpus. Sally is a rempus. Stella is a wumpus. Every zumpus is a rorpus. Sally is a rompus. Every numpus is a bompus. Every zumpus is a scrompus. Every rempus is a kerpus. Every zumpus is a vumpus. Every timpus is a yerpus. Every rempus is a numpus. Every vumpus is a worpus. Every rompus is a felpus. Every wumpus is a sterpus. Every rompus is a kerpus. Every zumpus is a rempus. Every rempus is a chorpus. Bob is a rorpus. Every wumpus is a fompus. Sally is a kerpus. Every zhorpus is a rompus. Is Sally a fompus or worpus?* |
| **CoT** | `Sally is a zhorpus.  Every zhorpus is a zumpus.  Every zumpus is a vumpus.  Every vumpus is a worpus.` |
| **Answer** | **Sally is a worpus.** |
| **Full Output** | `[ANT][ANT][ANT][ANT][ANT][ANT] Sally is a worpus.` |
| **Extracted Output** | **Sally is a worpus.** |
| **Question 3** | *Every shumpus is a yumpus. Every worpus is a yimpus. Every shumpus is a gwompus. Every tumpus is a boompus. Every worpus is a shumpus. Every storpus is a terpus. Max is a yimpus. Every shumpus is a rompus. Every wumpus is a jelpus. Every boompus is a terpus. Fae is a tumpus. Every tumpus is a worpus. Every rompus is a gorpus. Every timpus is a impus. Every jompus is a gerpus. Every boompus is a rompus. Fae is a boompus. Every boompus is a kerpus. Every zumpus is a bompus. Max is a rempus. Every rompus is a kerpus. Max is a impus. Every rempus is a impus. Every wumpus is a yumpus. Every grimpus is a terpus. Every tumpus is a jompus. Every yumpus is a felpus. Every jelpus is a felpus. Every shumpus is a felpus. Every rempus is a timpus. Every storpus is a jompus. Every rompus is a storpus. Every tumpus is a wumpus. Every wumpus is a jompus. Every boompus is a worpus. Fae is a storpus. Every worpus is a jelpus. Every grimpus is a felpus. Every worpus is a yumpus. Every rempus is a zumpus. Every kerpus is a grimpus. Is Fae a gwompus or bompus?* |
| **CoT** | `Fae is a tumpus.  Every tumpus is a worpus.  Every worpus is a shumpus.  Every shumpus is a gwompus.` |
| **Answer** | **Fae is a gwompus.** |
| **Full Output** | `[ANT][ANT][ANT][ANT][ANT][ANT] Fae is a bompus.` |
| **Extracted Output** | **Fae is a bompus.** |

