# OpenReview forum: "Anchored Diffusion Language Model"
_NeurIPS.cc/2025/Conference — NeurIPS 2025 poster_

### Official Review · Reviewer_zAqx · 2025-06-13

**Clarity:** 4
**Significance:** 3
**Originality:** 4
**Rating:** 5
**Confidence:** 4

**Summary:**

The paper "Anchored Diffusion Language Model" introduces a novel framework, the Anchored Diffusion Language Model (ADLM), to address the performance gap between diffusion language models (DLMs) and autoregressive (AR) models in likelihood modeling and text generation quality. ADLM employs a two-stage approach: an anchor network predicts distributions over important tokens (e.g., low-frequency or semantically key words), which then guide a denoising network to reconstruct missing tokens more accurately. The method is trained end-to-end using a new Anchored Negative Evidence Lower Bound (ANELBO) objective, which enhances likelihood modeling by leveraging structured guidance from anchor predictions.
ADLM significantly improves test perplexity on benchmarks like LM1B (9.54–25.4% over prior DLMs) and OpenWebText (12.3% over MDLM), narrowing the gap with AR models. It achieves state-of-the-art zero-shot generalization on six out of seven language modeling benchmarks, excelling in long-context and domain-specific datasets (e.g., PubMed, ArXiv). Using the remasking sampler, ADLM surpasses AR models in MAUVE score, indicating superior human-like text generation. The anchoring mechanism also enhances AR models, improving reasoning in math (GSM8K) and logic tasks (ProntoQA, ProsQA) beyond chain-of-thought approaches. Theoretically, the paper demonstrates that anchoring reduces sample complexity exponentially in directed graphical models.
Contributions:
- Introduction of ADLM, a two-stage DLM framework with anchor-guided denoising.
- Derivation of the ANELBO objective for joint training of anchor and denoising networks.
- Significant perplexity improvements on LM1B and OpenWebText, with strong zero-shot generalization.
- First DLM to outperform AR models in MAUVE score for human-like text generation.
- Application of anchoring to AR models, enhancing reasoning in math and logic tasks.

**Questions:**

- Why were comparisons limited to MDLM, SEDD, and BD3LM, and can you evaluate ADLM against other diffusion models (e.g., DiffusionLM, DiffusionBERT) or token prioritization methods (e.g., attention-based salience)?

- How does the frequency-based anchor selection ($\mu(\mathbf{x}^l) \leq \tau$) perform compared to semantic or syntactic criteria, and can you provide an ablation study?

- Can you quantify the computational cost (e.g., memory, runtime) of ADLM’s training and inference, particularly for the anchor and denoising networks?

- What are the quantitative improvements for AR models with anchoring on reasoning tasks (e.g., GSM8K, ProntoQA), and can you provide specific metrics?

- How does ADLM perform on larger models (e.g., GPT-3 scale) or diverse datasets (e.g., multilingual), and can you provide such evaluations?

**Ethical Concerns:**

["NO or VERY MINOR ethics concerns only"]

**Final Justification:**

Thank you for the clarification. This is one of the strongest paper I've come across in this NeurIPS. I stand by the scores given the rebuttal by the authors.

**Limitations:**

yes

**Paper Formatting Concerns:**

No.

**Quality:**

4

**Strengths And Weaknesses:**

*Quality*

Theoretical Rigor: The paper introduces the Anchored Negative Evidence Lower Bound (ANELBO) objective (Theorem 4.1, Page 5), which is rigorously derived to integrate anchor and denoising networks:

$\mathcal{L}_{\text{ANELBO}}(\mathbf{x}, \mathbf{y}; \varphi, \psi) = \mathbb{E}_{Z_0 \sim q(\cdot|\mathbf{x})} \left[-\log p_\psi(\mathbf{x} | \mathbf{y}_\varphi(Z_0))\right] + \sum_{i=1}^T \mathbb{E}_{Z_{t(i)} \sim q(\cdot|\mathbf{x})} \left[\lambda_{t(i)} \sum_{l=1}^L \left( \log \langle \mathbf{x}_\varphi^l(\mathbf{y}_\varphi(Z_{t(i)})), \mathbf{x}^l \rangle + \gamma \log \langle \mathbf{y}_\varphi^l(Z_{t(i)}), \mathbf{y}^l \rangle \right)\right]$

The proof, supported by Assumption 4.3 and Proposition 4.4, demonstrates reduced sample complexity ($\mathcal{O}(K^L) \to \mathcal{O}(L K^{d+1})$) in anchored models, grounding the method in graphical model theory.
Comprehensive Experiments: ADLM is evaluated on LM1B and OpenWebText, achieving significant perplexity improvements (9.54–25.4% over MDLM and SEDD on LM1B, 12.3% on OpenWebText). Zero-shot generalization is tested on seven benchmarks, with ADLM leading in six (e.g., PubMed, ArXiv). The paper also assesses text quality via MAUVE scores, where ADLM outperforms AR models, and reasoning tasks (GSM8K, ProntoQA, ProsQA) with AR integration.
Statistical Robustness: The paper reports results from two independent runs with fixed seeds, matching up to the first decimal point, indicating low variability. Additional statistical details in §C (not provided in the excerpt) further support robustness.
Limited Baseline Comparisons: While ADLM is compared to AR, MDLM, SEDD, and BD3LM, other relevant diffusion-based models (e.g., DiffusionBERT [He et al., 2023], DiffusionLM [Li et al., 2022b]) or recent discrete diffusion advancements (e.g., Liu et al., 2025) are not included. This narrows the context for assessing ADLM’s superiority.
Anchor Selection Simplification: The frequency-based anchor token selection ($\mu(\mathbf{x}^l) \leq \tau$) is simplistic and may not capture semantic importance in all contexts. The paper acknowledges alternatives (e.g., syntactic importance) but lacks ablation studies to justify the choice.

*Clarity*

Clear Problem Statement: The introduction articulates the limitations of DLMs, particularly the loss of context when important tokens are masked early, and positions ADLM as a solution to narrow the gap with AR models. The motivation is compelling and grounded in prior work.
Well-Structured Methodology: Section 3 (Page 3) clearly outlines ADLM’s two-stage framework: anchor network prediction and denoising conditioned on anchors. Figure 1 (Page 2) visually illustrates the process, enhancing understanding of token pathways (e.g., ‘cat’, ‘dog’).
Accessible Theory: The ANELBO derivation (Page 5) is detailed, with clear explanations of terms like the anchor loss ($\mathcal{L}_{\text{Anchor}}$) and weighting ($\lambda_{t(i)}$). The graphical model analysis (Page 6) connects anchoring to sample complexity, making theoretical implications accessible:

$\mathcal{L}_{\text{Anchor}}(\mathbf{x}; \varphi) = \mathbb{E}_{q(Z_{0:1}|\mathbf{x})} \left[ \sum_{i=0}^T D_{\text{KL}}(r(Y_{s(i)} | Z_{t(i)}, \mathbf{y}) \| r_\varphi(Y_{s(i)} | Z_{t(i)})) \right]$

Under-explained Anchor Selection: The frequency-based anchor selection (Page 5) is described briefly without justifying the threshold $\tau$ or discussing its impact on performance. A short example or ablation would clarify its practical effect.

*Significance*

Substantial Performance Gains: ADLM’s perplexity improvements (up to 25.4% on LM1B, 12.3% on OpenWebText) and state-of-the-art zero-shot results on six benchmarks demonstrate its potential to advance DLM pre-training, a critical area in NLP.

Breakthrough in Text Quality: Achieving the first DLM to surpass AR models in MAUVE score is a significant milestone, suggesting ADLM generates more human-like text, with implications for applications like chatbots and content generation.

Broad Applicability: The anchoring mechanism’s success in both DLMs and AR models (improving reasoning on GSM8K, ProntoQA, ProsQA) indicates versatility, potentially influencing diverse tasks like reasoning and controllable generation.

Theoretical Impact: The exponential reduction in sample complexity ($\mathcal{O}(L K^{d+1})$) via anchoring offers a new lens for analyzing generative models, likely inspiring further research in diffusion and graphical models.

Scalability Concerns: The lack of evaluation on larger models (e.g., GPT-3 scale) or computational cost details raises questions about ADLM’s scalability, potentially limiting its adoption in industry settings.

*Originality*

Novel Framework: ADLM’s two-stage approach—anchor network predicting important tokens followed by guided denoising—is a new paradigm for DLMs. The ANELBO objective is a novel training strategy, distinct from standard NELBO.

Theoretical Novelty: The graphical model analysis and sample complexity reduction via anchoring are original contributions, offering a new theoretical lens for diffusion models:

$$\text{Sample complexity: } \mathcal{O}(K^L) \to \mathcal{O}(L K^{d+1})$$

Cross-Model Application: Extending anchoring to AR models for improved reasoning is a creative application, distinguishing ADLM from prior DLM-focused work like MDLM [Sahoo et al., 2024] or SEDD [Lou et al., 2024].

Frequency-Based Anchoring: The anchor selection method (Page 5) is straightforward and not entirely novel, as frequency-based criteria have been used in NLP (e.g., TF-IDF). More sophisticated selection would enhance originality.

---

> ### Author Rebuttal · Authors · 2025-07-31
>
> >**W1. Missing comparison with DiffusionBERT [He et al., 2023], DiffusionLM [Li et al., 2022b] and DDPD (Liu et al., 2025). This narrows the context for assessing ADLM’s superiority.**
>
>
> A1. We thank the reviewer for the helpful suggestion.
>
> We respectfully note that we already compare with **DiffusionBERT** (He et al., 2022) and **DiffusionLM** (Li et al., 2022) in **Table 1(a)**.  For completeness, we now also include **DDPD** (Liu et al., ICLR 2025), **DUO** (Sahoo et al., ICML 2025) and **UDLM** (Schiff et al., ICLR 2025), although these baselines use uniform noising, while our method focuses on masked diffusion.
>
> ---
>
> ### Generative Perplexity using GPT2-Large
>
> Repeating results from Table 7 for clarity. DDPD-medium is newly added.
>
> | **Model**                | **Params** | **Sampling Steps** | **Gen. PPL ↓** |
> |--------------------------|------------|--------------------|----------------|
> | Plaid                    | 1.3B       | 4096               | 19.7           |
> | SEDD-medium              | 424M       | 2048               | 27.3           |
> | `DDPD-medium (Liu et al., 2025)` | 514M       | 2048               | 24.5           |
> | **ADLM (Ours)**          | 293M       | 2048               | **20.1**       |
> | **ADLM (Ours)**          | 293M       | 4096               | **15.7**       |
>
> ---
>
> ### Test Perplexity
>
> Repeating our results from Table 1. UDLM and DUO are newly added.
>
> | **Method**                  | **LM1B** | **OWT** |
> |-----------------------------|---------:|--------:|
> | `UDLM (Schiff et al., 2025)`  |31.3     |27.4    |
> | `DUO (Sahoo et al., 2025)`    | 29.9     |25.2    |
> | **ADLM (Ours)**             | **26.4** | **20.6** |
>
> ---
>
> ### Zero-Shot Validation Perplexity
>
> | **Model**               | **Lambada** | **PTB**  | **WikiText** | **LM1B** | **AG News** | **PubMed** | **ArXiv** |
> |-------------------------|-------------|----------|--------------|----------|-------------|------------|-----------|
> |`UDLM (Schiff et al.)`    | 53.57       | 112.82   | 39.42        | 77.59    | 80.96       | 50.98      | 44.08|
> | `DUO (Sahoo et al.)`      | 49.78       | **89.35**| 33.57        | 73.86    | 67.81       | 44.48      | 40.39     |
> | **ADLM (Ours)** (262B)  | 44.93       | 98.16    | 32.45        | 65.59    | 57.10       | 38.29      | 35.08     |
> | **ADLM (Ours)** (524B)  | **44.32**   | 95.37    | **31.94**    | **64.43**| **55.72**   | **37.56**  | **33.69** |
>
> ---
>
>
> >**W2. Anchor Selection Simplification: The frequency-based … justify the choice.**
>
>
> A2. Thank you for the thoughtful question. We actually have experiments with three different anchoring mechanisms: (i) relative frequency for LM1B and OWT (generative modeling), (ii) digit extraction for GSM8K (math reasoning), and (iii) verbs and nouns (exclude articles) in ProntoQA and ProsQA (logical reasoning). We will bring these results from the appendix to the main draft to better highlight these contributions.
>
> ---
>
> **(Relative frequency based anchoring)**  As noted in **Lines 378–380**, we acknowledge that using token frequency as a proxy for importance is a simplification and may not capture semantic importance in all contexts. However, our primary goal in this paper is to demonstrate that **even this simple strategy**, when integrated into DLM training via our **ANELBO** objective, yields substantial improvements (e.g., up to 25\% in PPL and outperforms AR in MAUVE score).
>
> ---
>
> **(Digit extraction anchoring)** In Sec 5.2.1 (and Appendix C.2.5), we study the use of anchoring in math reasoning using the GSM8K dataset. Here, we use the numerical digits as the anchors, and this enabled out-of-order reasoning (as shown in Lines 1740 - 1746 in Appendix C.2.5). Please see for example Question 1 from Table 9).
>
> Interestingly, ACoT first computes 3 + 4 = 7 to aggregate all consumption before subtracting from the total (16 - 7 = 9), a pattern that seems more aligned with human intuition. Anchoring thus enables a **look-ahead planning behavior**, which is in contrast to the commonly observed left-to-right decoding bias of standard chain-of-thought (CoT) reasoning.
>
> ---
>
> **(Verbs and nouns (exclude articles) anchoring)** For logical reasoning tasks on ProntoQA and ProsQA, we extract verbs and nouns (excluding articles such as ‘a’, “an’, ‘the’, etc.) as the anchors. By incorporating explicit supervision through [ANT], our method ACoT achieves a perfect accuracy of 100% on the relatively easier ProntoQA and 16.3\% improvement over CoT on ProsQA, demonstrating the effectiveness of anchoring in logical reasoning. We refer to Appendix C.2.5 for further details.
>
> This particular study also opens up other intriguing questions and potential directions. Words corresponding to verbs and nouns relate to the concepts of **terminals and syntactic categories** in the **Chomsky Hierarchy** (formal language theory). With this preface, anchoring can be thought of as another means of bringing classical NLP back into statistical language models. This view opens up other grammar-based mechanisms for anchor network design (e.g., multi-level hierarchical anchor network), which is an exciting direction for future study.
>
> ---
>
>
>
> >**W3. Ablation study on anchor threshold.**
>
>
> A3. We appreciate the reviewer’s suggestion to clarify the anchor selection threshold and its effect on performance. As noted in **Line 266**, we defer the ablation study of the threshold $\tau$ to **Appendix C.1.3 (Lines 1489–1501)**. There, we present a detailed empirical analysis showing how different values of $\tau$ influence performance (measured by NLL, PPL and BPD metrics). This justifies our default choice of $\tau = 5$ used across experiments.
>
> ---
>
> >**W4. Frequency-Based Anchoring: The anchor … originality.**
>
>
> A4. As acknowledged in **Lines 378–380**, the frequency-based anchoring mechanism is indeed a simplification and not claimed as a contribution. Our **primary focus is to demonstrate the impact of anchoring in DLMs**, rather than to propose methods for anchoring. Indeed, more sophisticated anchor selection would enhance performance as we have demonstrated in math and logical reasoning tasks. Please refer to Sec 5.2.1, Appendix C.2.5 and also our detailed response in A2.
>
> ---
>
>
> >** W5(Q1). Why were comparisons limited to MDLM, SEDD, and BD3LM, and can you evaluate ADLM against other diffusion models (e.g., DiffusionLM, DiffusionBERT) or token prioritization methods (e.g., attention-based salience)?**
>
> A5. We already include comparisons with **DiffusionLM** and **DiffusionBERT** in **Table 1(a)**. For completeness, we now include **DUO** (Sahoo et al., ICML 2025), **UDLM** (Schiff et al., ICLR 2025), and **DDPD** (Liu et al., ICLR 2025), as detailed in our **response A1** above.
>
> ---
>
> >**W6(Q2). How does… provide an ablation study?**
>
> A6. Please refer to our response in A4.
>
> ---
>
> >**W7(Q3). Can you quantify the computational cost (e.g., memory, runtime) …?**
>
> A7. As already reported in **Table 7 (Appendix C.1.3)**,  ADLM has **293M** parameters for both the **anchor** and **denoising** networks combined.
>   - For comparison:
>     - **Plaid**: 1.3B parameters
>     - **SEDD-Medium**: 424M parameters
>     - **GGM**: 387M parameters
>     - **MD4-Medium**: 400M parameters
>     - **DDPD-Medium**: 514M parameters
>
>
> Despite having fewer parameters, ADLM achieves **significantly better perplexity** and **zero-shot performance** than these models.
>
> ---
>
> #### Runtime Analysis (1024-token generation)
>
> | **Model** | **Sampling Steps** | **Time per Sample (s)** | **PPL ↓** | **Gen. PPL ↓** |
> |-----------|--------------------|--------------------------|-----------|----------------|
> | ReMDM      | 1024               | 19 sec                   | 23.17     | 28.6           |
> | ADLM (ours)      | 1024               | 28 sec                   | 20.62     | 25.4           |
>
> With just **9 additional seconds per sequence**, ADLM delivers:
> - 11.0% improvement in **test perplexity**
> - 11.1% improvement in **generative perplexity**
> - Improved performance across 6 of 7 **zero-shot benchmarks**
>
> ---
>
> >**W8(Q4). What are the quantitative improvements for AR models with anchoring on reasoning tasks (e.g., GSM8K, ProntoQA), and can you provide specific metrics?**
>
>
>
> A8. **Table 5** already shows quantitative improvements for AR models with anchoring on reasoning tasks:
>
> - **GSM8K** (math reasoning),
>
> - **ProntoQA** (logical reasoning), and
>
> - **ProsQA** (long-chain logical reasoning).
>
> Following prior works, we use **accuracy** as the standard evaluation metric. For clarity, we repeat our observation from Table 5 below:
>
> Anchored Chain-of-Thought (ACoT) consistently outperforms standard CoT and latent reasoning methods (e.g., COCONUT, iCoT). For example:
>
> - On **GSM8K**, ACoT improves over standard CoT by **+2.3%**.
>
> - On **ProsQA**, ACoT achieves **97.3%**, surpassing \textsc{Coconut} (97.0%) and CoT (77.5%).
>
> Please refer to **Appendix C.2** for more results.
>
> ---
>
>
> >**W9(Q5). How does ADLM perform on larger models (e.g., GPT-3 scale) or diverse datasets (e.g., multilingual), and can you provide such evaluations?**
>
> A9. To ensure fair comparison with prior diffusion baselines, we train and evaluate our diffusion model on the same benchmarks: **LM1B** and **OpenWebText** using comparable model sizes. Remarkably, **ADLM (293M)** outperforms much larger models like **Plaid (1.3B)**, achieving better perplexity with **5X fewer parameters** (please refer to Table 7 in Appendix C.1.3).
>
> Training models at **GPT-3 scale (175B)** or on **multilingual corpora** requires substantial compute and financial resources, which are currently beyond our academic scope. However, we believe that our **theoretical results**,  extensive evaluations on **standard pretraining**,  **zero-shot validation**, **complex planning/reasoning**, and robust performance across **varying inference steps** collectively demonstrate the **scalability and generality** of our proposed **anchored diffusion language model (ADLM)**.

---

> > ### Author Response · Authors · 2025-08-02
> > **Request Feedback on Rebuttal**
> >
> > Dear Reviewer zAqx,
> >
> > As we are in the middle of the discussion period, we kindly request you to let us know if the above clarifications and the newly added experiments have addressed your questions. We would be happy to address any additional points during the remaining time of the discussion phase. We thank you in advance for your time and efforts in engaging with us.
> >
> > Best,
> >
> > Authors of Paper # 963

---

### Official Review · Reviewer_gd1Y · 2025-07-02

**Clarity:** 3
**Significance:** 3
**Originality:** 3
**Rating:** 4
**Confidence:** 3

**Summary:**

The paper introduces the Anchored Diffusion Language Model (ADLM), which is a two stage framework that first predicts anchor tokens, and then predicts the likelihoods of missing tokens conditioned on the anchored predictions. This paper aims to tackle the issue of masking important tokens early in the forward process through use of anchoring. The paper compares to prior Diffusion Language Models (DLMs) on LM1B and OWT datasets and shows lower test perplexities. The paper also benchmarks against autoregressive models and shows better perplexity and MAUVE scores. The paper also demonstrates the utility of anchoring for AR models and compares to chain-of-thought approaches.

**Questions:**

The main questions are highlighted in the weakness section and are related to:
1. The cost of the additional anchor network.
2. The anchor token selection.

Additionally:

* Could the authors provide more details on the approach to labelling the anchor tokens for the training. There is no algorithm containing training details or how the online anchor labelling is done within the training loop. Algo. 1 seems to be pseudo-code for the sampling implementation and not the training.

**Ethical Concerns:**

["NO or VERY MINOR ethics concerns only"]

**Final Justification:**

Repeat of response below: "Thank you for your response. I will keep my score as I my concern over reproducibility is still present without a code release."

**Limitations:**

Yes.

**Quality:**

3

**Strengths And Weaknesses:**

### Strengths
* Experimentation: The paper includes significant experimentation where they compare to the existing DLMs and show improved performance. The paper includes experiments that vary the time parameter which is also helpful for demonstrating how all DLMs perform, with different time schedules.
* Results: The results of the experiments show that ADLMs tend to outperform the baselines/previous approaches.
* Theory: The theory seems to be correct.
* Novelty: The idea to build an additional anchoring network seems novel.
### Weaknesses
* Anchor token selection: The requirement to pre-label tokens (supervision) via a frequency-based analysis seems a weakness of the approach. The relative frequency of a token in a sequence might not be directly related to importance in some scenarios and it seems like it might require significant domain expertise. For example, in A.3.4, how would the anchor token selection operate within the sequence (0, 1, 1) and (1, 0, 0), since the main argument of that section is on the middle token $x^2$ is the anchor token. However, in both cases the middle variable is represented by a token that is more frequent (and potentially labelled as the less important token for the supervision).
* Requirement of large anchor network transformer: The paper writes that the anchor network adopts the transformer architecture from SEDD, with the denoising architecture using the same base architecture, but with half the number of layers. How easy is it to distinguish between the potential performance gain associated with adding significantly more parameters to the overall model and the introduction of the anchoring approach. For example, if you half the number of layers of the anchor network, (or get a similar total number of parameters to the SEDD network for ADLM), then does it still outperform the previous approaches. Overall, the question is whether the associated computational cost of the additional anchor network is work the performance gain?
* Reproducibility: The paper asserts that it “provides open access to the data and code, with sufficient instructions to reproduce the main experimental results”.  However, given the complexity of the work, the pseudo-code would not be enough to reproduce the results of the paper. The supplementary materials did not include code.

---

> ### Author Rebuttal · Authors · 2025-07-30
>
> >**W1.1.  Anchor token selection: The requirement to pre-label tokens (supervision) via a frequency-based analysis seems a weakness of the approach. The relative frequency of a token in a sequence might not be directly related to importance in some scenarios and it seems like it might require significant domain expertise.**
>
> A1.1. Thank you for the thoughtful question. We actually have experiments with three different anchor token selection mechanisms: (i) relative frequency for LM1B and OWT (generative modeling), (ii) digit extraction for GSM8K (math reasoning), and (iii) verbs and nouns (exclude articles) in ProntoQA and ProsQA (logical reasoning). These were not well highlighted in the main draft before; in the revision we will bring all three up front.
>
> **Relative frequency based anchoring)**  As noted in our **limitations section (Lines 378–380)**, we acknowledge that using token frequency as a proxy for importance is a simplification and may not always align with semantic relevance. However, our primary goal in this paper is to demonstrate that **even this simple strategy**, when integrated into diffusion training via our **ANELBO** objective, yields substantial performance gains (e.g., up to 25\% in PPL and outperform AR in MAUVE score).
>
> ---
>
> **(Digit extraction anchoring)** In Sec 5.2.1 (and Appendix C.2.5), we have studied the use of anchoring in math reasoning using the GSM8K dataset. Here, we used the numerical digits as the anchors, and this enabled out-of-order reasoning (as shown in Lines 1740 - 1746 in Appendix C.2.5; pasted an example question below for quick access).
>
> ---
>
> ### Question 1 from Table 9
>
> **Prompt**
> Janet's ducks lay 16 eggs per day. She eats three for breakfast every morning and bakes muffins for her friends every day with four. She sells the remainder at the farmers' market daily for \$2 per fresh duck egg.
>
> **Question:** How much in dollars does she make every day at the farmers' market?
>
> ---
>
> **Chain-of-Thought (CoT)**
> `«16 - 3 - 4 = 9»  «9 * 2 = 18»`
> **Answer:** `18`
>
> ---
>
> **Anchored Chain-of-Thought (ACoT) -- ours**
> `[ANT] [ANT] «3 + 4 = 7» «16 - 7 = 9» «9 * 2 = 18»`
> **Extracted Answer:** `18`
>
> Interestingly, ACoT first computes 3 + 4 = 7 to aggregate all consumption before subtracting from the total (16 - 7 = 9), a pattern that seems more aligned with human intuition. Moreover, anchoring enables a **look-ahead planning behavior**, which is in contrast to the commonly observed left-to-right decoding bias of standard chain-of-thought (CoT) reasoning.
>
> ---
>
> **(Verbs and nouns (exclude articles) anchoring)** For logical reasoning tasks with symbolic reasoning traces from ProntoQA, we extracted verbs and nouns (excluded articles such as ‘a’, “an’, ‘the’, etc.) as the anchors. While standard multi-stage training such as in COCONUT leads to a marginal improvement from 98.8% (CoT) to 99.8%, our anchoring mechanism further enhances performance. By incorporating explicit supervision through [ANT], ACoT achieves a perfect accuracy of 100% on this relatively easier benchmark, demonstrating the effectiveness of anchoring in logical reasoning. We refer to Appendix C.2.5 for further discussion and results from ProsQA.
>
> This particular study also opens up other intriguing questions and potential directions. Words corresponding to verbs and nouns relate to the concepts of **terminals and syntactic categories** in the **Chomsky Hierarchy** (formal language theory). With this preface, anchoring can be thought of as another means of bringing classical NLP back into statistical language models. This view opens up other grammar-based mechanisms to explore for anchor network design (e.g., multi-level hierarchical anchor network), which is an exciting direction for future study.
>
> ---
>
> >**W1.2. Regarding the Example in A.3.4…**
>
> A1.2. This example is designed as a controlled synthetic setting to **theoretically demonstrate the benefit of anchoring** when the important token is known. It is not meant to rely on frequency-based selection. Note that our main experiments are conducted on real-world datasets, where **token frequency provides a practical and scalable proxy** for importance, especially when combined with pretrained tokenizers from GPT-2 or BERT that capture semantic priors. While this proxy is admittedly simple, it consistently yields gains across diverse benchmarks. Learning anchor tokens through attention, gradients, or saliency would further enhance performance of our model.
>
> ---
>
> >**W2. Requirement of large anchor network transformer: The paper writes … half the number of layers. How easy is it to distinguish between the potential performance gain associated with adding significantly more parameters to the overall model and the introduction of the anchoring approach.**
>
>
> A2. Thank you for raising this important point. We add clarification below:
>
> 1. **ADLM does **not** significantly increase parameter count.** As detailed in **Lines 1526–1540** and Table 7 in Appendix C.1.3:
>    - ADLM uses **293M total parameters** (Anchor: 170M, Denoiser: 123M),
>    - This is **~5X smaller** than **Plaid (1.3B)** and **~130M smaller** than **SEDD-medium (424M)**,
>    - Yet ADLM outperforms both in Gen PPL by **4–11 points**, demonstrating high performance with lower cost.
>
> ---
> 2. **Ablation on architectural gains vs. anchoring.** In **Lines 276–278**, we introduce **ADLM***, which uses the same two-stage architecture as ADLM but disables the anchor loss (i.e., $\gamma=0$). This allows a fair comparison to **MDLM** under **equal training compute**.
>    - MDLM: 23.17 PPL
>    - ADLM*: 21.79 PPL
>    - ADLM (with ANELBO): **20.62 PPL**
>
>    This shows that:
>    - The **two-stage design alone** provides improvement (MDLM → ADLM*),
>    - But **anchoring via ANELBO** is essential for achieving state-of-the-art.
>
> ---
>
> 3. **Summary of compute-vs-performance tradeoff:**
>
> Repeating the results from Table 7 for clarity.
>
> | **Model**         | **Params** | **Sampling Steps** | **Gen. PPL ↓** |
> |------------------|------------|--------------------|----------------|
> | Plaid            | 1.3B       | 4096               | 19.7           |
> | SEDD-medium      | 424M       | 2048               | 27.3           |
> | GGM              | 387M       | 4096               | 19.5           |
> | **ADLM (ours)**  | 293M       | 2048               | **20.1**       |
> | **ADLM (ours)**  | 293M       | 4096               | **15.7**       |
>
> We will make this discussion more explicit in the revision.
>
> ---
>
> >**W3. Reproducibility: The paper asserts that it “provides open access to the data and code, with sufficient instructions to reproduce the main experimental results”. However, given the complexity of the work, the pseudo-code would not be enough to reproduce the results of the paper. The supplementary materials did not include code.**
>
>
> A3. Our implementation builds on open-source codebases from SEDD (https://github.com/louaaron/Score-Entropy-Discrete-Diffusion) and MDLM (https://github.com/kuleshov-group/mdlm), with training done using our ANELBO objective (Eq. 7). All training hyperparameters are detailed, and ablations are included to justify their choice in Appendix C.
>
> For evaluation, we use the source code of MDLM to compute the test perplexity. We use ReMDM (https://github.com/kuleshov-group/remdm) to compute MAUVE, Gen PPL, and entropy. Appendix C.1.1 provides links for all baselines, and Appendix C.1.2 describes our training/evaluation datasets.
>
> We believe these details are sufficient for reproducibility and will release our full code after incorporating the reviewers’ comments to ensure clarity and accessibility.
>
> ---
>
> >**Q1. The cost of the additional anchor network.**
>
> Please refer to W2 above.
>
> ---
>
> > **Q2. The anchor token selection.**
>
> Please refer to W1 above.
>
> ---
>
> > **Q3. Could the authors provide more details on the approach to labelling the anchor tokens for the training. There is no algorithm containing training details or how the online anchor labelling is done within the training loop. Algo. 1 seems to be pseudo-code for the sampling implementation and not the training.**
>
> Thank you for the helpful suggestion. We clarify the anchor labelling and training process below.
>
> - **Anchor Token Selection**: As described in Lines **183–190** of the main paper, we adopt a frequency-based strategy to identify important (anchor) tokens during training. These are tokens that occur less frequently within a sequence and are likely to carry more semantic content.
>
> ---
>
> - **Additional Implementation Details**: Appendix **C.1.3** elaborates on the implementation of anchor supervision and how these labels are integrated into the training loss (ANELBO). During each training step, anchor tokens are dynamically selected for each input sequence based on their token frequency, and only the tokens with frequency lower than a threshold contribute to the anchor loss term. We ablate on this threshold in Lines 1489 - 1501.
>
> ---
>
> - **Code Release**: We will release the full source code—including anchor token selection, model training, and evaluation—after incorporating reviewers’ feedback.
>
> We hope this addresses the reviewer’s concern, and we appreciate the opportunity to clarify.
>
> ---

---

> > ### Author Response · Authors · 2025-08-02
> > **Request Feedback on Rebuttal**
> >
> > Dear Reviewer gd1Y,
> >
> > As we are in the middle of the discussion period, we kindly request you to let us know if the above clarifications and the newly added experiments have addressed your questions. We would be happy to address any additional points during the remaining time of the discussion phase. We thank you in advance for your time and efforts in engaging with us.
> >
> > Best,
> >
> > Authors of Paper # 963

---

### Official Review · Reviewer_oJvZ · 2025-07-02

**Clarity:** 3
**Significance:** 2
**Originality:** 3
**Rating:** 4
**Confidence:** 3

**Summary:**

This paper identifies a key limitation of current diffusion language models: important tokens are often masked early in the forward process, which prevents the model from accessing sufficient context to accurately reconstruct the original sequence. To address this, the authors propose the Anchored Diffusion Language Model (ADLM), which introduces an anchor network alongside the standard denoising network to predict the likelihoods of important tokens. They derive an Anchored Negative Evidence Lower Bound (ANELBO) to jointly train the two networks. ADLM achieves strong performance on language modeling perplexities and MAUVE. Additionally, the proposed anchoring mechanism can be integrated into autoregressive models to enhance reasoning ability.

**Questions:**

- Have you explored applying the proposed method to stronger language models such as LLaMA or Qwen? What are the results in those settings?
- What optimizer settings did you use during training? Do different optimizer hyperparameters for the anchor network and the denoising network lead to better performance?
- I find Eq. (5) somewhat confusing: the right-hand side appears to describe a distribution over $z_s$ ​, but the left-hand side uses $r(y_s)$. Could you provide further clarification on this point?

**Ethical Concerns:**

["NO or VERY MINOR ethics concerns only"]

**Final Justification:**

I increased my score from 3 to 4 after the rebuttal because most of my concerns have been addressed. The authors provided detailed architectural descriptions, clarified compute comparisons via controlled ablations, and demonstrated the benefits of their two-stage anchoring mechanism. However, I did not raise my score higher because I still have reservations regarding the significance of the method. In particular, there is a lack of evaluations on stronger language models. Overall, the paper represents a technically sound and well-motivated contribution to the emerging area of diffusion language modeling, and I tend to accept it now.

**Limitations:**

yes

**Quality:**

2

**Strengths And Weaknesses:**

Strengths
- The motivation is clear and novel. It addresses an interesting question that may help explain why diffusion language models underperform autoregressive models in likelihood.
- The paper is well-structured.
- The experimental results are strong: ADLM outperforms previous diffusion language models on both perplexity and MAUVE.

Weaknesses
- The paper does not report architectural details of the anchor and denoising networks, such as layer counts or dimensions. This makes it unclear whether ADLM requires more compute than MDLM under the same training data. It also leaves open how ADLM compares to baselines under equal pre-training or inference cost.
- As noted in [1], using GPT-2 perplexity to measure sentence quality is questionable. The authors are encouraged to report entropy in Table 2 alongside GPT-2-Large perplexity.
- In autoregressive experiments, the authors use GPT-2 as the base model. GPT-2 is outdated and performs poorly on reasoning tasks, making it hard to validate the claimed improvements.
- The motivation is to use an anchor network to predict important tokens, as illustrated in Figure 1. However, the paper does not visualize the learned anchor outputs after training to verify that the model captures such tokens. This weakens the connection between the method and its original motivation.
[1] Zheng et al. Masked Diffusion Models are Secretly Time-Agnostic Masked Models and Exploit Inaccurate Categorical Sampling. ICLR2025.

---

> ### Author Rebuttal · Authors · 2025-07-30
>
> >**W1.1. The paper does not report architectural details of the anchor and denoising networks, such as layer counts or dimensions.**
>
> A1.1. We appreciate the reviewer’s suggestion to improve clarity. We will expand the discussion in Lines **247–250** and Appendix **C.1.3 (Lines 1471–1479)** to incorporate the following architecture details:
>
> - Both networks follow the DiT-Base architecture [Peebles & Xie, ICCV 2023], consistent with SEDD (Lou et al., ICML 2024) and MDLM [Sahoo et al., NeurIPS 2024].
> - The anchor network uses 12 DiT blocks.
> - The denoising network uses 6 DiT blocks.
> - Each block has hidden dimension = 768 and 12 attention heads.
> - The input sequence length is 1024 for OWT and 128 for LM1B.
>
> ---
>
>
> >**W1.2 This makes it unclear whether ADLM requires more compute than MDLM under the same training data. It also leaves open how ADLM compares to baselines under equal pre-training or inference cost.**
>
> A1.2. Thank you for highlighting this important point. As discussed in **Lines 276–278**, we perform a controlled ablation to compare **ADLM** against **MDLM** under equal compute and data budgets.
>
> We introduce **ADLM***, a variant of our model that uses the same two-stage architecture as ADLM but **disables the anchor loss** (i.e., sets $\gamma = 0$). This allows a fair comparison with **MDLM**, using the same training compute.
>
> - As shown in **Table 1(b)** of the main paper:
>   - **MDLM** achieves a test perplexity of **23.17**.
>   - **ADLM*** improves this to **21.79**, validating the benefit of our two-stage design.
>   - Full **ADLM**, trained with our proposed **ANELBO** loss, further reduces perplexity to **20.62**
>
> In addition, **Table 7** compares ADLM with other baselines under comparable pre-training or inference cost.
>
> - **ADLM** achieves **better generation quality** than several larger models:
> - Outperforms Plaid (1.3B), SEDD-medium (424M), and GGM (387M),
> - While using only 293M parameters and comparable sampling steps—thus requiring significantly **less pre-training and inference cost**.
>
> ---
>
> ### Model Size vs Performance
> Repeating our results from Table 7 for clarity.
> Newly added results: SEDD-medium (T=1000), MD4-medium, SEDD-small, MD4-small.
>
> | **Model Type**       | **Evaluated Model**                                            | **Params** | **Gen. PPL (↓)** |
> |----------------------|----------------------------------------------------------------|------------|------------------|
> | *Diffusion*    | Plaid (Gulrajani & Hashimoto, 2023) (*T=4096*)          | 1.3B     | 19.7    |
> |                      | SEDD-medium (Lou et al., 2024) (*T=2048*)                      | 424M       | 27.3 |
> |                      | `SEDD-medium` (Lou et al., 2024) (*T=1000*)                      | 424M       | 31.95 |
> |                      | `MD4-medium` (Shi et al., 2024)  (*T=1000*)                    | 416M       |27.51 |
> |                      | `SEDD-small` (Lou et al., 2024) (*T=1000*)                      | 165M      | 42.94 |
> |                      | `MD4-small` (Shi et al., 2024) (*T=1000*)                    | 165M      |33.16|
> |                      | MDLM (Sahoo et al., 2024) (*locked-in, T=1000*)                | 170M  | 44.2  |
> |                      | ReMDM (Wang et al., 2025a) (*remasking, T=1024*)  | 170M  | 28.6   |
> |                      | GGM (Varma et al., 2025) (*T=4096*)                            | 387M | 19.5  |
> |                      | **ADLM (ours)** (*locked-in, T=1000*)                          | 293M | 32.9  |
> |                      | **ADLM (ours)** (*remasking, T=1000*)             | 293M       | 26.8             |
> |                      | **ADLM (ours)** (*remasking, T=1024*)             | 293M       | 25.1             |
> |                      | **ADLM (ours)** (*remasking, T=2048*)             | 293M       | 20.1             |
> |                      | **ADLM (ours)** (*remasking, T=4096*)             | 293M       | **15.7**         |
>
>
> ---
>
>
>
> >**W2. As noted in [1], using GPT-2 perplexity to measure sentence quality is questionable. The authors are encouraged to report entropy in Table 2 alongside GPT-2-Large perplexity.**
>
>
> A2. Thank you for the suggestion. We agree that GPT-2 perplexity alone may not fully capture generation quality. For this reason, we already report both **GPT-2-Large perplexity and entropy** in **Table 3**, along with a range of sampling steps. We kindly refer the reviewer to Table 3 and Lines 313–318 for detailed discussion. Table 3 results indicate that the proposed method ADLM not only generates high quality text but also preserves diversity.
>
> ---
>
>
> >**W3. In autoregressive experiments, the authors use GPT-2 as the base model. GPT-2 is outdated and performs poorly on reasoning tasks, making it hard to validate the claimed improvements.**
>
>
> A3. We thank the reviewer for pointing this out. As noted in Lines **320–322**, our primary focus in this paper is on **diffusion language models (DLMs)**. However, we find it noteworthy that the benefits of our **anchoring mechanism** extend beyond diffusion, unlike the prior DLMs that are restricted to only diffusion settings.
>
> To support this claim, we include experiments on **autoregressive models using GPT-2** (a widely adopted baseline), ensuring **compatibility with recent works** such as **BoLT, iCoT, and Coconut**.  Our goal is not to compete with modern LLMs, but to show **consistent gains from anchoring** under comparable compute. All baselines use the same architecture and training setup for a fair analysis, which otherwise becomes difficult given the cost associated with training large language models.
>
> We hope the reviewer will consider our anchored AR experiments as a proof-of-concept, highlighting the generality of our approach across diffusion and AR model families.
>
> ---
>
>
> >**W4. The motivation is to use an anchor network to predict important tokens, as illustrated in Figure 1. However, the paper does not visualize the learned anchor outputs after training to verify that the model captures such tokens. This weakens the connection between the method and its original motivation.**
>
> A4. Thank you for the insightful question. We note that anchor tokens are not explicitly decoded to minimize the number of extra tokens generated  during inference. This offers 2 crucial benefits during training: (1) ANELBO remains end-to-end differentiable with respect to the parameters of the anchor and the denoiser network (discussed in Lines 1471-1479); and (2) This further enables implicit reasoning via latent distribution over anchors as opposed to a single sampled token.
>
> Below, we recall **qualitative and visual evidence** from the paper showing that the model learns to focus on important tokens. **Appendix C.1.4** shows that anchoring leads to the generation of **more human-like, structured text**, supporting the **qualitative claims** in Figure 1 and the **quantitative gains** reported in **Table 3** (e.g., improved MAUVE, Gen PPL and entropy scores).
>
> Importantly, we **visualize intermediate tokens** generated during inference in **Appendix C.2.5**, specifically in the section *“Results on GSM8K”* (Lines **1740–1746**) where anchoring can be better interpreted. This example demonstrates how the anchor network **selects semantically important tokens**, enabling **out-of-order reasoning** that is different from the conventional left-to-right decoding of standard CoT, and thus, anchoring reflects better planning behavior.
>
> Furthermore, we provide theoretical results justifying the benefits of anchoring in improved sample complexity (Sec 4.2) and likelihood modeling (Appendix A3.4).  These findings support our core motivation: that explicitly modeling token importance via anchor transition enhances both **latent reasoning** and **generation quality**.
>
> ---
>
> >**W5 (Q1). Have you explored applying the proposed method to stronger language models such as LLaMA or Qwen?**
>
>
> A5. We appreciate the reviewer’s interest in scaling our method to stronger LLMs. As noted in Lines 320 - 322, the **primary focus** of this paper is on **diffusion language models**, a line of research that is still in its early stages compared to LLMs. Our goal is to improve the modeling, training, inference, and zero-shot validation capabilities of diffusion language models through our proposed **anchoring mechanism**.
>
> To demonstrate the generality of our approach, we apply anchoring to GPT-2 scale models in a controlled autoregressive setting. While GPT-2 is not state-of-the-art, it provides a fair and transparent testbed to isolate the effects of our algorithm under similar compute budgets. Our method is modular and can be integrated into any transformer-based diffusion or AR backbone.
>
> We thank the reviewer for showing this promising direction for our future study.
>
> ---
>
> >**W6.1 (Q2). What optimizer settings did you use during training?**
>
> A6.1. We used the same **AdamW optimizer** for both anchor and denoiser with learning rate = 3e-4, $\beta_1=0.9$ and $\beta_2=0.999$ and no weight decay. We will expand the implementation details in Appendix C.1.3 to include these details.
>
> >**W6.2. Do different optimizer hyperparameters for the anchor network and the denoising network lead to better performance?**
>
> A6.2. Our goal is to **isolate the impact of anchoring** without extreme hyper-parameter tuning, so we use the same optimizer and hyperparameters for both the anchor and denoiser networks. Using **different optimizers or hyperparameters** may lead to better performance. Our current results already show strong gains without much hyper-parameter tuning.
>
> ---
>
> >**W7 (Q3). I find Eq. (5) somewhat confusing.... Could you provide further clarification on this point?**
>
> W7. Thank you for catching this typo. In **Eq (5)**, the **right-hand side describes a categorical distribution over $y_s$**, not $z_s$. The left-hand side correctly denotes the distribution over $y_s$. We appreciate the reviewer’s attention to detail.
>
> ---

---

> > ### Author Response · Authors · 2025-08-02
> > **Request Feedback on Rebuttal**
> >
> > Dear Reviewer oJvZ,
> >
> > As we are in the middle of the discussion period, we kindly request you to let us know if the above clarifications and the newly added experiments have addressed your questions. We would be happy to address any additional points during the remaining time of the discussion phase. We thank you in advance for your time and efforts in engaging with us.
> >
> > Best,
> >
> > Authors of Paper # 963

---

> > > ### Comment · Reviewer_oJvZ · 2025-08-03
> > > **score updated**
> > >
> > > I increased my score from 3 to 4 after the rebuttal because most of my concerns have been addressed. The authors provided detailed architectural descriptions, clarified compute comparisons via controlled ablations, and demonstrated the benefits of their two-stage anchoring mechanism.

---

> > > > ### Author Response · Authors · 2025-08-03
> > > > **Thank You**
> > > >
> > > > Dear Reviewer oJvZ,
> > > >
> > > > Thank you for increasing your score. We are glad to hear that our rebuttal has addressed your concerns.
> > > >
> > > > We appreciate the reviewer’s thoughtful comments and suggestions to improve the clarity of our paper.
> > > >
> > > > Best,
> > > >
> > > > Authors of Paper #963

---

### Official Review · Reviewer_mP11 · 2025-07-03

**Clarity:** 2
**Significance:** 3
**Originality:** 2
**Rating:** 4
**Confidence:** 4

**Summary:**

This paper observes that the performance of masked diffusion models can be improved by generating certain “special” tokens—called anchor tokens—early in the sampling process. (This isn't a novelty as a similar observation was made in Shi et al., 2024 and He et al., 2022)

Modified forward process:  To enforce such behavior, they train their model where these anchor tokens are masked later in the forward diffusion process.

Modified reverse process: The reverse sampling process is redesigned so that anchor tokens are predicted first. Additionally, the parameterization of the denoising model is adjusted: instead of mapping noisy sequences directly to clean data, part of the network predicts a distribution over the anchor tokens, which is then used by another network to map to the clean data distribution.

Modified NELBO: To support this training strategy, the authors propose a modified NELBO objective tailored for the anchoring mechanism.

The authors claim that this approach improves the likelihood performance of masked diffusion models. Moreover, anchoring also boosts the performance of autoregressive models and enhances reasoning capabilities in mathematical and logical tasks, outperforming existing chain-of-thought methods.


### References
He et al., 2022 "DiffusionBERT: Improving Generative Masked Language Models with Diffusion Models"
Sahoo et al., 2025 "The diffusion duality", ICML 2025

**Questions:**

See above

**Ethical Concerns:**

["NO or VERY MINOR ethics concerns only"]

**Final Justification:**

After the rebuttal, I choose to retain my rating

**Limitations:**

yes

**Quality:**

3

**Strengths And Weaknesses:**

### Strengths
The overall results, particularly on the reasoning tasks, are very promising.

### Weaknesses

1.	The observation that prioritizing important words during the diffusion process can improve likelihood was previously proposed in Shi et al. (2024) and He et al. (2022). The authors must compare their method against Gen-MD4 from Shi et al. (2024). Notably, Gen-MD4 does not use a specialized network to identify important tokens; it simply introduces a learnable weight for each token in the vocabulary. Since the proposed method involves two networks, I suspect it may require significantly more parameters (see my concern below).

2.	Model size clarity: It is unclear how large the denoising model is and what transformer sizes are used for the anchor network and the denoising network. What is the total parameter count for both networks combined? Given that the embedding and final projection layers in transformers are typically large, I would not be surprised if the proposed algorithm uses substantially more parameters than a standard autoregressive transformer.

3.	MAUVE score: The MAUVE scores reported using GPT-2 embeddings are often unreliable. Could the authors recompute the MAUVE scores using modern BERT embeddings?


### Minor comments
* For completeness, the authors should consider including additional baselines such as DUO and UDLM (Uniform-State Diffusion) in Tables 1 and 3.

### References
He et al., 2022. "DiffusionBERT: Improving Generative Masked Language Models with Diffusion Models"

Sahoo et al., 2025. "The diffusion duality", ICML 2025

---

> ### Author Rebuttal · Authors · 2025-07-31
>
> > **Q1.1 The observation that prioritizing important words during the diffusion process can improve likelihood was previously proposed in Shi et al. (2024) and He et al. (2022).**
>
> A1.1. The importance of specific tokens has been well-explored in classic NLP literature —e.g., via TF-IDF weighting (Salton & Buckley, 1987), conditional random fields (Lafferty et al., 2001), and neural attention mechanisms (Bahdanau et al., 2014; Vaswani et al., 2017). While recent works such as GenMD4 (Shi et al., 2024) and DiffusionBERT (He et al., 2022) focus on static global importance, our proposed ADLM introduces a fundamentally different approach based on **sequence-level contextual anchoring** and a novel, **fully differentiable training objective (ANELBO)**. We highlight the key differences below.
>
> ---
>
> **GenMD4 vs. ADLM**
>
> - **Local vs. Global Importance.**
>   GenMD4 assigns global, static importance to tokens via learnable weights across the vocabulary. In contrast, ADLM selects important tokens dynamically **per input sequence** via anchored transition function (Eq. 5). For instance, "cat" and "dog" may be important in a given context but overlooked by GenMD4 due to low global weights. ADLM adaptively anchors these tokens via our anchor transition function.
>
> - **Objective Function.**
>   GenMD4 requires REINFORCE-style gradient estimators because it cannot backpropagate through discrete samples (which introduces high variance and instability). ADLM introduces an **Anchored NELBO** that supports end-to-end differentiable training using an adaptive anchor transition (see Sec 3), leading to more stable optimization.
>
> - **Generalization vs. Overfitting.**
>   As acknowledged in Shi et al. (2024, Sec 7.1), GenMD4 can overfit to training data, achieving low perplexity but poor generalization. ADLM achieves both strong **generative perplexity** and **zero-shot generalization**, as shown in Sec 5.1.1 in the main draft and Appendix C.1.3-C.1.4 in the supplementary material. .
>
> ---
>
> **DiffusionBERT vs. ADLM**
>
> DiffusionBERT also assigns static importance via frequency-based "spindle" noise schedules. This global assignment, similar to GenMD4, does not adapt to sequence-level context and suffers from the same **hard unmasking** problem.
> In contrast, ADLM introduces **anchoring as a latent reasoning process** to model contextual importance via anchor transition function and adaptively decode tokens important for each sequence.
>
> ---
>
> Our anchoring mechanism opens up other intriguing directions. Words corresponding to verbs and nouns relate to the concepts of **terminals and syntactic categories** in the **Chomsky Hierarchy** (formal language theory). With this preface, anchoring can be thought of as another means of bringing classical NLP back into statistical language models. This view opens up other grammar-based mechanisms for anchor network design (e.g., multi-level hierarchical anchor network), which is an exciting direction for future study.
>
>
> > **Q1.2. The authors must compare with GenMD4 from Shi et al. 2024.**
>
> A1.2. We now include GenMD4 and MD4 in Table 1 (b). Repeating other results for clarity.
>
> ###
>
> Test Perplexity on OWT
>
> | **Method**                | **PPL (↓)** |
> |---------------------------|--------------------|
> | SEDD Absorb    | 24.10              |
> | `MD4`        | 22.13 |
> | `GenMD4`      | 21.80|
> | MDLM                      | 22.98              |
> | **ADLM (ours)** | **20.14**         |
>
> ---
>
> ### Model Size vs Performance
> Repeating results from Table 7 for clarity.
> Newly added: SEDD-medium (T=1000), MD4-medium, SEDD-small and MD4-small.
>
> | **Model Type**       | **Evaluated Model**                                            | **Params** | **Gen. PPL (↓)** |
> |----------------------|----------------------------------------------------------------|------------|------------------|
> | *Diffusion*    | Plaid (Gulrajani & Hashimoto, 2023) (*T=4096*)          | 1.3B     | 19.7    |
> |                      | SEDD-medium (Lou et al., 2024) (*T=2048*)                      | 424M       | 27.3 |
> |                      | `SEDD-medium` (Lou et al., 2024) (*T=1000*)                      | 424M       | 31.95 |
> |                      | `MD4-medium` (Shi et al., 2024)  (*T=1000*)                    | 416M       |27.51 |
> |                      | `SEDD-small` (Lou et al., 2024) (*T=1000*)                      | 165M      | 42.94 |
> |                      | `MD4-small` (Shi et al., 2024) (*T=1000*)                    | 165M      |33.16|
> |                      | MDLM (Sahoo et al., 2024) (*locked-in, T=1000*)                | 170M  | 44.2  |
> |                      | ReMDM (Wang et al., 2025a) (*remasking, T=1024*)  | 170M  | 28.6   |
> |                      | GGM (Varma et al., 2025) (*T=4096*)                            | 387M | 19.5  |
> |                      | **ADLM (ours)** (*locked-in, T=1000*)                          | 293M | 32.9  |
> |                      | **ADLM (ours)** (*remasking, T=1000*)             | 293M       | 26.8             |
> |                      | **ADLM (ours)** (*remasking, T=1024*)             | 293M       | 25.1             |
> |                      | **ADLM (ours)** (*remasking, T=2048*)             | 293M       | 20.1             |
> |                      | **ADLM (ours)** (*remasking, T=4096*)             | 293M       | **15.7**         |
>
> ---
>
> ### Zero-shot Evaluation
>
> Repeating results from Table 4 for clarity. Newly added: MD4
>
> | **Model**                 | **Lambada** | **PTB**  | **WikiText** | **LM1B** | **AG News** | **PubMed** | **ArXiv** |
> |--------------------------|-------------|----------|--------------|----------|-------------|------------|-----------|
> | MDLM                     | 47.52       | 95.26    | 32.83        | 67.01    | 61.15       | 41.89      | 37.37     |
> | `MD4`                       | 48.43       |102.26|35.42| —        | —           | —          | —         |
> | **ADLM (ours)** (262B)   | 44.93       | 98.16    | 32.45        | 65.59    | 57.10       | 38.29      | 35.08     |
> | **ADLM (ours)** (524B)   | **44.32**   | **95.37**| **31.94**    | **64.43**| **55.72**   | **37.56**  | **33.69** |
>
> ---
>
> > **Q2.1. Model size clarity: It is unclear how large … network.**
>
> A2.1. We will expand architecture details in Lines **247–250** and Appendix **C.1.3 (Lines 1471–1479)** as follows:
>
> - Both networks follow the DiT-Base architecture [Peebles & Xie, 2023], consistent with SEDD and MDLM.
> - The anchor network uses 12 DiT blocks.
> - The denoising network uses 6 DiT blocks.
> - Each block has hidden dimension = 768 and 12 attention heads.
> - The input sequence length is 1024 for OWT and 128 for LM1B.
>
> ---
>
> >**Q2.2. What is the total parameter count for both networks combined?**
>
> A2.2. ADLM has **293M** total As given  in **Table 7 in Appendix C.1.3**, . For comparison:
>
> - **Plaid** has **1.3B** parameters
>
> - **SEDD-Medium** has **424M** parameters
>
> - **GGM** has **387M** parameters
>
> - **MD4-medium** has **424M** parameters
>
> Thus, ADLM achieves superior results while using **fewer parameters** (293M) than these strong baselines.
>
> ---
>
> >**Q2.3. Given … the proposed algorithm uses substantially more parameters than a standard AR transformer.**
>
> A2.3. Our theoretical results (Sec 4) show that anchoring requires less parameters. This translates into practice: the **denoiser uses ~123M parameters**, while the anchor network uses ~170M, totaling **293M**—still smaller than DLMs with comparable performance, please refer to Table 7 in Appendix C.1.3 (and new results in A1.2).
>
> ADLM is **parameter-efficient** because:
>
> - The denoiser has **half the transformer layers** than that of the anchor network.
>
> - We **share the tokenizer and vocabulary** across the anchor and the denoiser.
>
> - We use a **lightweight linear projection** to pass anchor logits to the denoiser (Appendix C.1.3), avoiding additional parameters due to re-embedding layers.
>
> ---
>
> >**Q3. MAUVE score … Could the authors recompute the MAUVE scores using modern BERT embeddings?**
>
> We have now recomputed the MAUVE scores using BERT embeddings (bert-base-uncased) with max sequence length=512 and T=4096. We compare with the best performing baseline ReMDM from Table 3 and obtain a **7% improvement** in MAUVE score. We will add these results in the revision.
>
> ### Evaluation on OWT.
>
> | **Method**      | **MAUVE (↑)** | **Gen PPL. (↓)** | **Entropy (↑)** |
> |----------------|---------------|------------------|-----------------|
> | ReMDM          |   0.702      | 17.6             | 5.20            |
> | **ADLM (ours)**| **0.754**     | **15.9**         | **5.19**        |
>
> ---
>
> >**Q4. For completeness, compare with DUO and UDLM (Uniform-State Diffusion).**
>
> A4. While our primary focus is on **masked diffusion**, we now include comparison with **uniform diffusion** methods: DUO and UDLM.
>
> ---
>
> ### Test PPL
>
> | **Method**                  | **LM1B** | **OWT** |
> |-----------------------------|---------:|--------:|
> | `UDLM (Schiff et al., 2025)`  | 31.3     | 27.4    |
> | `DUO (Sahoo et al., 2025)`    | 29.9     | 25.2    |
> | **ADLM (Ours)**             | **26.4** | **20.6** |
>
> ---
>
> ### Zero-Shot Validation PPL
>
> | **Model**               | **Lambada** | **PTB**  | **WikiText** | **LM1B** | **AG News** | **PubMed** | **ArXiv** |
> |-------------------------|-------------|----------|--------------|----------|-------------|------------|-----------|
> | `UDLM (Schiff et al., 2025)` | 53.57     | 112.82   | 39.42        | 77.59    | 80.96       | 50.98      | 44.08     |
> | `DUO (Sahoo et al., 2025)`   | 49.78     | **89.35**| 33.57        | 73.86    | 67.81       | 44.48      | 40.39     |
> | **ADLM (Ours)** (262B)     | 44.93     | 98.16    | 32.45        | 65.59    | 57.10       | 38.29      | 35.08     |
> | **ADLM (Ours)** (524B)     | **44.32** | 95.37    | **31.94**    | **64.43**| **55.72**   | **37.56**  | **33.69** |
>
> We hope this addresses the reviewer’s concern, and we appreciate the opportunity to clarify.

---

> > ### Author Response · Authors · 2025-08-02
> > **Request Feedback on Rebuttal**
> >
> > Dear Reviewer mP11,
> >
> > As we are in the middle of the discussion period, we kindly request you to let us know if the above clarifications and the newly added experiments have addressed your questions. We would be happy to address any additional points during the remaining time of the discussion phase. We thank you in advance for your time and efforts in engaging with us.
> >
> > Best,
> >
> > Authors of Paper # 963

---

> > > ### Author Response · Authors · 2025-08-07
> > > **Request Feedback on Rebuttal**
> > >
> > > Dear Reviewer mP11,
> > >
> > > As we are approaching the end of the discussion phase, we kindly request you to let us know if the above clarifications and the newly added experiments have addressed your questions. We thank you again for your thoughtful comments and for helping us improve the clarity of our paper.
> > >
> > > Best,
> > >
> > > Authors of Paper # 963

---

### Note · Authors · 2025-08-16

### Summary of Changes Made During Rebuttal

Dear Area Chair and Reviewers,

Thank you for overseeing the review process of our paper (#963). We are grateful to the reviewers for their constructive feedback and engaging with us during the discussion phase. Below,  we summarize the additions in response to reviewer feedback:

- **Architectural and implementation details**
  Added number of layers, model sizes, optimizer settings, and expanded Appendix C.1.3 (as requested by `mP11`, `oJvZ`, `gd1Y`, `zAqx`).

- **Additional baselines**
  - *Masked DLMs*: Compared against **SEDD-small**, **GenMD4**, and **MD4** (`mP11`, `oJvZ`).
  - *Uniform DLMs*: Added **DDPD**, **DUO**, and **UDLM** (`mP11`, `zAqx`).

- **Anchoring strategies**
  Clarified **three types of anchoring approaches** and ablations (requested by `gd1Y`, `zAqx`).

---

We hope these clarifications and additions assist in your assessment of our submission.


Best,

Authors of Paper #963

---

### Decision · Program_Chairs · 2025-09-17

**Decision:**

Accept (poster)

**Comment:**

I recommend acceptance of this paper, which presents the Anchored Diffusion Language Model (ADLM), a novel two-stage framework that significantly improves diffusion language models' performance. The approach addresses a key limitation of masked diffusion models by first predicting distributions over important tokens through an anchor network, then using these to guide missing token prediction. This results in substantial improvements in perplexity (up to 25.4% over prior DLMs) and MAUVE scores that surpass autoregressive models.

All reviewers acknowledge the paper's technical merits, with three recommending acceptance. The authors have addressed concerns regarding architectural details, baseline comparisons, and anchoring strategies during the rebuttal phase.

However, I would like to remind the authors that they should discuss recent developments in diffusion language models, particularly LLaDA (https://arxiv.org/abs/2502.09992) and Dream-7B (https://hkunlp.github.io/blog/2025/dream/), which have significantly closed the gap between autoregressive and diffusion language models. It would strengthen the paper to discuss how anchoring techniques could contribute to these newer models, and how ADLM fits into this rapidly evolving landscape.